# Optimal Uniform OPE and Model-based Offline Reinforcement Learning in Time-Homogeneous, Reward-Free and Task-Agnostic Settings

**Ming Yin** [1,2] and **Yu-Xiang Wang**[1]

[1]Department of Computer Science, UC Santa Barbara
[2]Department of Statistics and Applied Probability, UC Santa Barbara
ming_yin@ucsb.edu   yuxiangw@cs.ucsb.edu

## Abstract

This work studies the statistical limits of uniform convergence for offline policy evaluation (OPE) problems with model-based methods (for episodic MDP) and provides a unified framework towards optimal learning for several well-motivated offline tasks. Uniform OPE $\sup_\Pi |Q^\pi - \hat{Q}^\pi| < \epsilon$ is a stronger measure than the point-wise OPE and ensures offline learning when $\Pi$ contains all policies (the global class). In this paper, we establish an $\Omega(H^2 S/d_m \epsilon^2)$ lower bound (over model-based family) for the global uniform OPE and our main result establishes an upper bound of $\tilde{O}(H^2/d_m \epsilon^2)$ for the *local* uniform convergence that applies to all *near-empirically optimal* policies for the MDPs with *stationary* transition. Here $d_m$ is the minimal marginal state-action probability. Critically, the highlight in achieving the optimal rate $\tilde{O}(H^2/d_m \epsilon^2)$ is our design of *singleton absorbing MDP*, which is a new sharp analysis tool that works with the model-based approach. We generalize such a model-based framework to the new settings: offline task-agnostic and the offline reward-free with optimal complexity $\tilde{O}(H^2 \log(K)/d_m \epsilon^2)$ ($K$ is the number of tasks) and $\tilde{O}(H^2 S/d_m \epsilon^2)$ respectively. These results provide a unified solution for simultaneously solving different offline RL problems.

## 1 Introduction

Offline reinforcement learning (offline RL) targets at learning a reward-maximizing policy in an unknown *Markov Decision Process* (MDP) using a static data generated by running a behavior policy [Lange et al., 2012, Levine et al., 2020]. This framework is widely applicable in applications where online exploration is demanding but historical data are plentiful. Examples include medicine [Liu et al., 2017] (safety concerns limit the applicability of unproven treatments but electronic records are abundant) and autonomous driving [Codevilla et al., 2018] (building infrastructure for testing new policy is expensive while collecting data from current setting is almost free).

Parallel to its practical significance, recently there is a surge of theoretical investigations towards offline RL via two threads: *offline policy evaluation* (OPE), where the goal is to estimate the value of a target (fixed) policy $V^\pi$ [Jiang and Li, 2016, Liu et al., 2018, Kallus and Uehara, 2020, 2019, Uehara and Jiang, 2019, Nachum et al., 2019, Xie et al., 2019, Yin and Wang, 2020, Duan et al., 2020, Wang et al., 2021, Zhang et al., 2021a] and *offline (policy) learning* which intends to output a near-optimal policy [Chen and Jiang, 2019, Le et al., 2019, Xie and Jiang, 2021, 2020, Liu et al., 2020b, Hao et al., 2020, Zanette, 2021, Jin et al., 2020c, Hu et al., 2021, Yin et al., 2021b, Rashidinejad et al., 2021].

Yin et al. [2021a] initiates the studies for offline RL from the new perspective of *uniform convergence* in OPE (uniform OPE for short) which unifies OPE and offline learning tasks. Generally speaking,

35th Conference on Neural Information Processing Systems (NeurIPS 2021).

given a policy class $\Pi$ and offline data with $n$ episodes, uniform OPE seeks to coming up with OPE estimators $\widehat{V}_1^\pi$ and $\widehat{Q}_1^\pi$ satisfy $\sup_{\pi\in\Pi} ||\widehat{Q}_1^\pi - Q_1^\pi||_\infty < \epsilon$. The task is to achieve this with the optimal episode complexity: the "minimal" number of episodes $n$ needed as a function of $\epsilon$, failure probability $\delta$, the parameters of the MDP as well as the behavior policy $\mu$ in the minimax sense.

To further motivate the readers why uniform OPE should be considered, we state its relation to offline learning. Indeed, uniform OPE to RL is analogous of uniform convergence of empirical risk in statistical learning [Vapnik, 2013]. In supervised learning, it has been proven that almost all learnable problems are learned by an (asymptotic) *empirical risk minimizer* (ERM) [Shalev-Shwartz et al., 2010]. In offline RL, the natural counterpart is the *empirical optimal policy* $\widehat{\pi}^\star := \mathrm{argmax}_\pi \widehat{V}_1^\pi$ and with uniform OPE it further ensures $\widehat{\pi}^\star$ is a near-optimal policy for the offline learning via:

$$0 \leq Q_1^{\pi^\star} - Q_1^{\widehat{\pi}^\star} = Q_1^{\pi^\star} - \widehat{Q}_1^{\pi^\star} + \widehat{Q}_1^{\pi^\star} - \widehat{Q}_1^{\widehat{\pi}^\star} + \widehat{Q}_1^{\widehat{\pi}^\star} - Q_1^{\widehat{\pi}^\star} \leq 2\sup_\pi |Q_1^\pi - \widehat{Q}_1^\pi|. \tag{1}$$

On the *policy evaluation* side, there is often a need to evaluate the performance of a *data-dependent* policy. Uniform OPE suffices for this purpose since it will allow us to evaluate policies selected by safe-policy improvements, proximal policy optimization, UCB-style exploration-bonus as well as any heuristic exploration criteria (please refer to Yin et al. [2021a] and the references therein for further discussions). In this paper, we study the uniform OPE problem under the *finite horizon stationary MDPs* and focus on the model-based approaches. Specifically, we consider two representative class: global policy class $\Pi_g$ (contains all (deterministic) policies) and local policy class $\Pi_l$ (contains policies near the empirical optimal one, see Section 2.1). We ask the following question:

*What is the statistical limit for uniform OPE and what is its connection to optimal offline learning?*

We answer the first part by showing the global uniform OPE requires a lower bound of $\Omega(H^2 S/d_m\epsilon^2)$[1] for the family of model-based approach and the local uniform OPE can achieve $\tilde{O}(H^2/d_m\epsilon^2)$ minimax rate by the model-based plug-in estimator and this implies optimal offline learning. Importantly, the procedure of the model-based approach via learning $\widehat{\pi}^\star$ through planning over the empirical MDP has a wider range of use in offline RL as it naturally adapts to the challenging tasks like *offline task-agnostic learning* and *offline reward-free learning*. See Section 1.2.

## 1.1 Related works

**Offline reinforcement learning.**[2] Information-theoretical considerations for offline RL are first proposed for *infinite horizon discounted setting* via Fitted Q-Iteration (FQI) type function approximation algorithms [Chen and Jiang, 2019, Le et al., 2019, Xie and Jiang, 2021, 2020] which can be traced back to [Munos, 2003, Szepesvári and Munos, 2005, Antos et al., 2008a,b].

For the finite horizon case, Yin et al. [2021a] first achieves $\tilde{O}(H^3/d_m\epsilon^2)$ complexity under non-stationary transition but their results cannot further improve in the stationary setting. Recently, Yin et al. [2021b] designs the offline variance reduction algorithm for achieving the optimal $\tilde{O}(H^2/d_m\epsilon^2)$ rate. Their result is for a specific algorithm that uses data splitting while our results work for any algorithms that returns a nearly empirically optimal policy via uniform convergence. Our results on the offline task-agnostic and the reward-free settings are entirely new. Concurrently, Ren et al. [2021] considers the horizon-free setting but does not provide uniform convergence guarantee.

**Model-based approaches with minimaxity.** It is known model-based methods are minimax-optimal for online RL with regret $\tilde{O}(\sqrt{HSAT})$ (*e.g.* Azar et al. [2017], Efroni et al. [2019]). In the generative model setting, Agarwal et al. [2020] shows model-based approach is still minimax optimal $\tilde{O}((1-\gamma)^{-3}SA/\epsilon^2)$ by using a $s$-absorbing MDP construction and this model-based technique is later reused for other more general settings (*e.g.* Markov games [Zhang et al., 2020a] and linear MDPs [Cui and Yang, 2020]) and also for overcoming the sample size barrier [Li et al., 2020]. In offline RL, Yin et al. [2021a] uses the model-based methods to achieve $\tilde{O}(H^3/d_m\epsilon^2)$ complexity.

**Task-agnostic and Reward-free problems.** The reward-free problem is initiated in the online RL [Jin et al., 2020a] where the agent needs to efficiently explore an MDP environment *without* using any reward information. It requires high probability guarantee for learning optimal policy for *any*

---

[1]Here $d_m$ is the minimal marginal state-action occupancy, see Assumption 2.4.

[2]We only provide a short discussion of the most related works due to the space constraint. A detailed discussion can be found in Appendix A.

reward function. Later, Kaufmann et al. [2020], Menard et al. [2020] establish the $\tilde{O}(H^3S^2A/\epsilon^2)$ complexity and Zhang et al. [2020c] further tightens the dependence to $\tilde{O}(H^2S^2A/\epsilon^2)$. Recently, Zhang et al. [2020b] proposes the task-agnostic setting where one needs to use exploration data to simultaneously learn $K$ tasks and proves an upper bound $\tilde{O}(H^5SA\log(K)/\epsilon^2)$. However, although these settings remain critical in the offline regime, no statistical result has been derived so far.

## 1.2 Our contribution

**Optimal local uniform OPE**. First and foremost, we derive the $\tilde{O}(H^2/d_m\epsilon^2)$ optimal episode complexity for local uniform OPE (Theorem 4.1) via the model-based method and this implies optimal offline learning with the same rate (Corollary 4.2); this result strictly improves upon Yin et al. [2021a] ($\tilde{O}(H^3/d_m\epsilon^2)$) non-trivially through our new *singleton-absorbing MDP* technique.

**Information-theoretical characterization of the global uniform OPE.** We characterize the statistical limit for the global uniform convergence by proving a minimax lower bound $\Omega(H^2S/d_m\epsilon^2)$ (over all model-based approaches) (Theorem 3.1). This result answers the question left by Yin et al. [2021a] that the global uniform OPE is generically harder than the local uniform OPE / offline learning by a factor of $S$, such a difference will dominate when the state space is exponentially large.

**Generalize to the new offline settings.** Critically, our model-based frameworks naturally generalize to the more challenging settings like task-agnostic and reward-free settings. In particular, we establish the $\tilde{O}(H^2\log(K)/d_m\epsilon^2)$ (Theorem 5.3) and $\tilde{O}(H^2S/d_m\epsilon^2)$ (Theorem 5.4) complexities for *offline task-agnostic learning* and *offline reward-free learning*. Both results are new and optimal.

**Singleton-absorbing MDP: a sharp analysis tool for episodic stationary transition case.** On the technical end, our major contribution is the novel design of *singleton-absorbing MDP* which handles the data-dependence hurdle encountered in the stationary MDPs. To decouple the data-dependence between $\widehat{P}_{s,a}$ and $\widehat{V}$, Agarwal et al. [2020] uses a $s$-absorbing MDP $\widehat{V}_s$ (in lieu of $\widehat{V}$) of each state for the independence. To control the error propagation between $\widehat{V}_s$ and $\widehat{V}$, they use the $\epsilon$-net covering such that the value of $\widehat{V}_s$ traverse the evenly-spaced grids in $[0,(1-\gamma)^{-1}]$. However, when applied to finite horizon case, the complexity increases as there are $H$ different quantities $(V_1,...,V_H)$ and the $\epsilon$-nets need to cover the $H$-dimensional space $[0,H]^H$. This result in a exponential-$H$ covering number and the metric entropy blows up by a factor $H$, which yields suboptimal result. In contrast, the *singleton-absorbing MDP* technique designs a single absorbing MDP that can also control the error propagation sufficiently well. This sharp analysis tool negates the conjecture of Cui and Yang [2020] that absorbing MDP is not well suitable for finite horizon stationary MDP.

**Significance: Unifying different offline settings** Beyond the study of statistical limit in uniform OPE, this work solves the sample optimality problems for the local uniform OPE, offline task-agnostic and offline reward-free problems. If we take a deeper look, the algorithmic frameworks utilized are all based on the model-based empirical MDP construction and planning. Therefore, as long as we can analyze such framework sharply (*e.g.* via novel absorbing-MDP technique), then it is hopeful that our techniques can be generalized to tackle more sophisticated settings. On the other hand, things could be more tricky for online RL since the exploration phases need to be specifically designed for each settings and there may not be one general algorithmic pattern that dominates. Our findings reveal the model-based framework is fundamental for offline RL as it subsumes settings like local uniform OPE, offline task-agnostic and offline reward-free learning into the identical learning pattern. Considering these tasks were originally proposed in the online regime under different contexts, such a unified view from the model-based perspective offers a new angle for understanding offline RL.

## 2 Problem setup

**Episodic stationary reinforcement learning.** A finite-horizon *Markov Decision Process* (MDP) is denoted by a tuple $M = (\mathcal{S}, \mathcal{A}, P, r, H, d_1)$, where $\mathcal{S}$ and $\mathcal{A}$ are finite state action spaces with $S := |\mathcal{S}|, A := |\mathcal{A}|$. A stationary (time-invariant) transition kernel has the form $P : \mathcal{S} \times \mathcal{A} \times \mathcal{S} \mapsto [0,1]$ with $P(s'|s,a)$ representing the probability transition from state $s$, action $a$ to next state $s'$. Besides, $r : \mathcal{S} \times \mathcal{A} \mapsto \mathbb{R}$ is the expected reward function and given $(s,a)$ which satisfies $0 \le r \le 1$ and assumed known. $d_1$ is the initial state distribution and $H$ is the horizon. At time $t$, a policy $\pi = (\pi_1,...,\pi_H)$ assigns each state $s \in \mathcal{S}$ a probability distribution $\pi_t(s)$ over

Figure 1: Related comparisons of sample complexities for offline RL

| Result/Method | Setting | Type | Complexity | Uniform guarantee? |
|---|---|---|---|---|
| Le et al. [2019] | $\infty$-horizon | FQI variants | $\widetilde{O}((1-\gamma)^{-6}\beta_\mu/\epsilon^2)$ | No |
| FQI [Chen and Jiang, 2019] | $\infty$-horizon | FQI variants | $\widetilde{O}((1-\gamma)^{-6}C/\epsilon^2)$ | No |
| MSBO/MABO [Xie and Jiang, 2020] | $\infty$-horizon | FQI variants | $\widetilde{O}((1-\gamma)^{-4}C_\mu/\epsilon^2)$ | No |
| OPEMA [Yin et al., 2021a] | $H$-horizon | Non-splitting | $\widetilde{O}(H^3/d_m\epsilon^2)$ | $\sqrt{H}/S$-local uniform |
| OPDVR Yin et al. [2021b] | $H$-horizon | Data splitting | $\widetilde{O}(H^2/d_m\epsilon^2)$ | No |
| Model-based Plug-in (Corollary 4.2) | $H$-horizon | Non-splitting | $\widetilde{O}(H^2/d_m\epsilon^2)$ | $\sqrt{H/S}$-local uniform |
| Task-Agnostic (Theorem 5.3) | $H$-horizon | Non-splitting | $\widetilde{O}(H^2\log(K)/d_m\epsilon^2)$ | — |
| Reward-Free (Theorem 5.4) | $H$-horizon | Non-splitting | $\widetilde{O}(H^2S/d_m\epsilon^2)$ | — |

* $K$ is the number of tasks for Task-agnostic setting and $\beta_\mu$, $C$ and $1/d_m$ are data coverage parameters that measure the state-action dependence and are qualitative similar under their respective assumptions.

actions. For a policy $\pi$, a random trajectory $s_1, a_1, r_1, \ldots, s_H, a_H, r_H, s_{H+1}$ is generated as follows: $s_1 \sim d_1, a_t \sim \pi(\cdot|s_t), r_t = r(s_t, a_t), s_{t+1} \sim P(\cdot|s_t, a_t), \forall t \in [H]$.

For any policy $\pi$ and any $h \in [H]$, value function $V_h^\pi(\cdot) \in \mathbb{R}^S$ and Q-value function $Q_h^\pi(\cdot, \cdot) \in \mathbb{R}^{S \times A}$ are defined as: $V_h^\pi(s) = \mathbb{E}_\pi[\sum_{t=h}^H r_t|s_h = s]$, $Q_h^\pi(s, a) = \mathbb{E}_\pi[\sum_{t=h}^H r_t|s_h, a_h = s, a]$, $\forall s, a \in \mathcal{S}, \mathcal{A}$. The goal of RL is to find a policy $\pi^\star$ such that $v^\pi := \mathbb{E}_\pi\left[\sum_{t=1}^H r_t\right]$ is maximized, which is equivalent to simultaneously maximize $V_1^\pi(s)$ (or $Q_1^\pi(s, a)$) for all $s$ (or $s, a$) [Sutton and Barto, 2018]. Therefore, for a targeted accuracy $\epsilon > 0$ it suffices to find a policy $\pi_{\text{alg}}$ such that $\left\|Q_1^\star - Q_1^{\pi_{\text{alg}}}\right\|_\infty \leq \epsilon$. We denote $V_h^\pi, Q_h^\pi$ as column vectors and $P_{s,a}$ as the row vector. In particular, we denote the average marginal state-action occupancy $d^\pi(s, a)$ as: $d^\pi(s, a) := \frac{1}{H}\sum_{t=1}^H \mathbb{P}[s_t = s|s_1 \sim d_1, \pi] \cdot \pi_t(a|s)$.

**Offline setting.** The offline RL assumes that episodes $\mathcal{D} = \left\{\left(s_t^{(i)}, a_t^{(i)}, r_t^{(i)}, s_{t+1}^{(i)}\right)\right\}_{i \in [n]}^{t \in [H]}$ are rolling from some behavior policy $\mu$ a priori. In particular, we do not assume the knowledge of $\mu$.

**Model-based RL.** We focus our attention on the model-based methods, which has witnessed numerous successes and is one of the most critical components of theoretical RL as a whole (as reviewed in Section 1.1). To make the presentation precise, we define the following:

**Definition 2.1.** *Model-based RL: Solving RL problems (either learning or evaluation) through learning / modeling transition dynamic $P$.*

We emphasize that the model-based approaches in general (*e.g.* Jaksch et al. [2010], Ayoub et al. [2020], Kidambi et al. [2020]) follow the procedure of modeling the full MDP $M = (\mathcal{S}, \mathcal{A}, P, r, H, d_1)$ instead of only the transition $P$. Nevertheless, we (by convention) assume the mean reward function is known and the initial state distribution $d_1$ will not affect the choice of optimal policy $\pi^\star$. Thus, Definition 2.1 suffices for our purposes.

### 2.1 Uniform convergence in offline RL

We study offline RL from the uniform OPE perspective. Concretely, uniform OPE extends the point-wise (fixed target policy) OPE to a family of policies $\Pi$. The goal is to construct estimator $\widehat{Q}_1^\pi$ such that $\sup_{\pi \in \Pi}\left\|Q_1^\pi - \widehat{Q}_1^\pi\right\| < \epsilon$, which automatically ensures point-wise OPE for any $\pi \in \Pi$. More importantly, uniform OPE directly implies offline learning when $\Pi$ contains optimal policies. As explained in Section 1, let $\widehat{\pi}^\star := \arg\max_\pi \widehat{V}_1^\pi$ be the *empirical optimal policy* for some OPE estimator $\widehat{v}^\pi$, then by (1) $\widehat{\pi}^\star$ is a near-optimal policy given uniform OPE guarantee. We consider the following two policy classes that are of the interests.

**Definition 2.2** (The global (deterministic) policy class.). *The global policy class $\Pi_g$ consists of all the non-stationary (deterministic) policies.*

It is well-known [Sutton and Barto, 2018] there exists at least one (deterministic) optimal policy, therefore $\Pi_g$ is sufficiently rich for evaluating algorithms that aim at learning the optimal policy.

**Definition 2.3** (The local policy class). *Given empirical MDP $\widehat{M}$ and $\widehat{V}_h^\pi$ is the value under $\widehat{M}$. Let $\widehat{\pi}^\star := \arg\max_\pi \widehat{V}_1^\pi$ be the empirical optimal policy, then the local policy class $\Pi_l$ is defined as:*

$$\Pi_l := \left\{\pi : \text{s.t. } \left\|\widehat{V}_h^\pi - \widehat{V}_h^{\widehat{\pi}^\star}\right\|_\infty \leq \epsilon_{opt}, \forall h \in [H]\right\}$$

*where $\epsilon_{opt} \geq 0$ is a parameter.*

In above $\widehat{M}$ uses $\widehat{P}$ in lieu of $P$ where $\widehat{P}(s'|s,a) = \frac{n_{s',s,a}}{n_{s,a}}$ if $n_{s,a} > 0$ and $1/S$ otherwise.[3] This class characterizes policies in the neighborhood of empirical optimal policy. Given $\widehat{P}$, it is efficient to obtain $\widehat{\pi}^{\star}$ using Value / Policy Iteration, therefore it is more practical to consider the neighborhood of $\widehat{\pi}^{\star}$ (instead of $\pi^{\star}$) since practitioners can use data $\mathcal{D}$ to really check $\Pi_l$ whenever needed. Next we present the regularity assumption required for uniform convergence OPE problem.

**Assumption 2.4** (Exploration requirement). *Logging policy $\mu$ obeys that $\min_s d^{\mu}(s) > 0$, for any state $s$ that is "accessible". Moreover, we define the quantity $d_m := \min_{s,a}\{d^{\mu}(s,a) : d^{\mu}(s,a) > 0\}$ (recall $d^{\mu}(s,a)$ in Section 2) to be the minimal average marginal state-action probability.*

State $s$ is "accessible" means there exists a policy $\pi$ so that $d^{\pi}(s) > 0$. If for any policy $\pi$ we always have $d^{\pi}(s) = 0$, then state $s$ can never be visited in the given MDP. Note this is weaker than Yin et al. [2021a] since $d^{\mu}(s)$ is the average version of $d_t^{\mu}(s)$. Assumption 2.4 is the minimal assumption needed for the consistency of uniform OPE task and is qualitatively similar to the *concentrability* assumption [Munos, 2003]. This assumption can be potentially relaxed for pure offline learning problems, *e.g.* Liu et al. [2019], Rashidinejad et al. [2021], where they only require $d^{\mu}(s)(d^{\mu}(s,a)) > 0$ for any state $s$ $(s,a)$ satisfies $d^{\pi^*}(s)(d^{\pi^*}(s,a)) > 0$.

# 3   Statistical Hardness for Model-based Global Uniform OPE

From (1) and Definition 2.2, it is clear the global uniform OPE implies offline RL, therefore it is natural to wonder whether they just are *"the same task"* (their sample complexities have the same minimax rates). If this conjecture is true, then deriving sample efficient global OPE method is just as important as deriving efficient offline learning algorithm (plus the additional benefit of evaluating data-dependent algorithms)! Yin et al. [2021a] proves the $\tilde{O}(H^3 S/d_m \epsilon^2)$ upper bound and $\Omega(H^3/d_m \epsilon^2)$ lower bound for global uniform OPE, but it is unclear whether the additional $S$ is essential. We answer the question affirmatively by providing a tight lower bound result with a concise proof to show no model-based algorithm can surpass $\Omega(S/d_m \epsilon^2)$ information-theoretical limit.

**Theorem 3.1** (Minimax lower bound for global uniform OPE). *Let $d_m$ be a parameter such that $0 < d_m \leq \frac{1}{SA}$. Let the problem class be $\mathcal{M}_{d_m} := \{(\mu, M) \mid \min_{t,s_t,a_t} d_t^{\mu}(s_t,a_t) \geq d_m\}$. Then there exists universal constants $c, C, p > 0$ such that: for any $n \geq cS/d_m \cdot \log(SAp)$,*

$$\inf_{\widehat{Q}_{1,mb}} \sup_{\mathcal{M}_{d_m}} \mathbb{P}_{\mu,M} \left( \sup_{\pi \in \Pi_g} \left\| \widehat{Q}_{1,mb}^{\pi} - Q_1^{\pi} \right\|_{\infty} \geq C \sqrt{\frac{H^2 S}{n d_m}} \right) \geq p,$$

*where $\widehat{Q}_{1,mb}$ is the output of any model-based algorithm and $\Pi_g$ is defined in Definition 2.2.*

By setting $\epsilon := \sqrt{\frac{H^2 S}{n d_m}}$, Theorem 3.1 establishes the global uniform convergence lower bound of $\Omega(H^2 S/d_m \epsilon^2)$ over model-based methods, which builds the hard statistical threshold between the global uniform OPE and the local uniform OPE tasks by a factor of $S$ since the local case has achievable $\tilde{O}(1/d_m \epsilon^2)$ rate on the dependence for state-actions. This result also reveals the global uniform convergence bound in Yin et al. [2021a] ($\tilde{O}(H^3 S/d_m \epsilon^2)$) is essentially minimax rate-optimal for their *non-stationary setting*[4] and complements the story on the optimality behavior for global uniform OPE. Moreover, from the generative model view the lower bound degenerates to $S/d_m \epsilon^2 \approx \Theta(S^2 A/\epsilon^2)$ which is linear in the model size $S^2 A$. This means in order to achieve global uniform convergence any algorithm needs to estimate each coordinate of transition kernel $P(s'|s,a)$ accurately. We now provide the proof sketch and full proof is deferred to Appendix C.

*Proof Sketch.* We only explain the case where $H = 2$ in this proof sketch. Our proof relies on the following novel reduction to $l_1$ density estimation

$$\sup_{\pi \in \Pi_g} \left\| \widehat{Q}_1^{\pi} - Q_1^{\pi} \right\|_{\infty} \geq \sup_{s,a} \frac{1}{2} \left\| \widehat{P}(\cdot|s,a) - P(\cdot|s,a) \right\|_1$$

---

[3]Here $n_{s,a}$ is the number of pair $(s,a)$ being visited among $n$ episodes. $n_{s',s,a}$ is defined similarly.

[4]To be rigorous, we ramark that it is rate-optimal since for the non-stationary setting the dependence for horizon is higher by a factor $H$.

and leverages the Minimax rate for estimating discrete distribution under $l_1$ loss is $O(\sqrt{S/n_{s,a}})$ [Han et al., 2015]. Concretely, by Definition 2.1, let $\widehat{P}$ be the learned transition by any arbitrary model-based method. Since we assume $r$ is known and by convention $Q_{H+1}^\pi = 0$ for any $\pi$, then by Bellman equation $\widehat{Q}_h^\pi = r_h + \widehat{P}^{\pi_{h+1}}\widehat{Q}_{h+1}^\pi, \ \forall h \in [H]$. In particular, $\widehat{Q}_{H+1}^\pi = Q_{H+1}^\pi = 0$, and this implies $\widehat{Q}_H^\pi = Q_H^\pi = r_H$. Now, again by definition of Bellman equation $\widehat{Q}_{H-1}^\pi = r_{H-1} + \widehat{P}^{\pi_H}\widehat{Q}_H^\pi = r_{H-1} + \widehat{P}^{\pi_H}r_H$ and $Q_{H-1}^\pi = r_{H-1} + P^{\pi_H}r_H$, therefore (recall $H = 2$ and note $r_H \in \mathbb{R}^{S \cdot A}, r_H^{\pi_H} \in \mathbb{R}^S$ )

$$\sup_{\pi \in \Pi_g} \left\| \widehat{Q}_{H-1}^\pi - Q_{H-1}^\pi \right\|_\infty = \sup_{\pi \in \Pi_g} \left\| \left( \widehat{P}^{\pi_H} - P^{\pi_H} \right) r_H \right\|_\infty = \sup_{\pi \in \Pi_g} \left\| \left( \widehat{P} - P \right) r_H^{\pi_H} \right\|_\infty$$

$$\approx \sup_{r \in \{0,1\}^S} \left\| \left( \widehat{P} - P \right) r \right\|_\infty \geq \sup_{s,a} \frac{1}{2} \left\| \widehat{P}(\cdot|s,a) - P(\cdot|s,a) \right\|_1 \geq O(\sqrt{S/n_{s,a}});$$

Lastly, using exponential tail bound to obtain $O(\sqrt{S/n_{s,a}}) \gtrsim O(\sqrt{S/nd_m})$ with high probability. See Appendix C for how to prove the result for the general $H$. ∎

# 4 Optimal local uniform OPE via model-based plug-in method

Global uniform OPE is intrinsically harder than the offline learning problem due to the additional state-space dependence and such a gap will amplify when $S$ is (exponentially) large. This motivates us to switch to the local uniform convergence regime that enables optimal learning but also has sub-linear state-action size $\tilde{O}(1/d_m)$ in the policy evaluation. Yin et al. [2021a] Theorem 3.7 first obtains the $\tilde{O}(H^3/d_m\epsilon^2)$ local uniform convergence for $\Pi_l$ (recall Definition 2.3) and also obtains the same rate for the learning task. Unfortunately, their technique cannot further reduces the dependence of $H$ for stationary transition case. In this section we show the model-based plug-in approach ensures optimal local uniform OPE and further implies optimal offline learning with episode complexity $\tilde{O}(H^2/d_m\epsilon^2)$. To this end, we design the new *singleton-absorbing MDP* to handle the challenge in the stationary transition setting, which uses the absorbing MDP with one single $H$-dimensional reference point and is our major technical contribution. The *singleton-absorbing MDP* technique avoids the exponential $H$ cover used in Cui and Yang [2020] and answers their conjecture that absorbing MDP is not well suitable for finite horizon stationary MDP.[5]

## 4.1 Model-based Offline Plug-in Estimator

Recall $n_{s,a} := \sum_{i=1}^n \sum_{h=1}^H \mathbf{1}[s_h^{(i)}, a_h^{(i)} = s, a]$ be the total counts that visit $(s, a)$ pair, then the model-based offline plug-in estimator constructs estimator $\widehat{P}$ as: $\widehat{P}(s'|s,a) = \frac{\sum_{i=1}^n \sum_{h=1}^H \mathbf{1}[(s_{h+1}^{(i)}, a_h^{(i)}, s_h^{(i)}) = (s', s, a)]}{n_{s,a}}$, if $n_{s,a} > 0$ and $\widehat{P}(s'|s,a) = \frac{1}{S}$ if $n_{s,a} = 0$. As a consequence, the estimators $\widehat{Q}_h^\pi, \widehat{V}_h^\pi$ are computed as: $\widehat{Q}_h^\pi = r + \widehat{P}^{\pi_{h+1}}\widehat{Q}_{h+1}^\pi = r + \widehat{P}\widehat{V}_{h+1}^\pi$, with the initial distribution $\widehat{d}_1(s) = n_s/n$. Under the above setting, we can define the empirical Bellman optimality equations (as well as the population version for completeness) as $\forall s \in \mathcal{S}, h \in [H]$:

$$V_h^\star(s) = \max_a \left\{ r(s,a) + P(\cdot|s,a)V_{h+1}^\star \right\}, \quad \widehat{V}_h^\star(s) = \max_a \left\{ r(s,a) + \widehat{P}(\cdot|s,a)\widehat{V}_{h+1}^\star \right\}.$$

Now we can state our local uniform OPE result with this construction.

## 4.2 Main results for local uniform OPE and offline learning

Recall $\widehat{\pi}^\star := \arg\max_\pi \widehat{V}_1^\pi$ is the empirical optimal policy and the local policy class $\Pi_l := \{\pi : \text{s.t. } \left\| \widehat{V}_h^\pi - \widehat{V}_h^{\widehat{\pi}^\star} \right\|_\infty \leq \epsilon_{\text{opt}}, \forall h \in [H]\}$.

---

[5]See their Section 7, first bullet point for a discussion.

**Theorem 4.1** (optimal local uniform OPE). *Let $\epsilon_{opt} \leq \sqrt{H/S}$ and denote $\iota = \log(HSA/\delta)$. For any $\delta \in [0, 1]$, there exists universal constants $c, C$ such that when $n > cH \cdot \log(HSA/\delta)/d_m$, with probability $1 - \delta$,*

$$\sup_{\pi \in \Pi_l} \left\| \widehat{Q}_1^\pi - Q_1^\pi \right\|_\infty \leq C \left[ \sqrt{\frac{H^2 \iota}{nd_m}} + \frac{H^{2.5} S^{0.5} \iota}{nd_m} \right].$$

Theorem 4.1 establishes the $\tilde{O}(H^2/d_m \epsilon^2)$ complexity bound and directly implies the upper bound for $\sup_{\pi \in \Pi_l} ||\widehat{V}_1^\pi - V_1^\pi||_\infty$ with the same rate. This result improves the local uniform convergence rate $\tilde{O}(H^3/d_m \epsilon^2)$ in Yin et al. [2021a] (Theorem 3.7) by a factor of $H$ and is near-minimax optimal (up to the logarithmic factor). Such result is first achieved by our novel *singleton absorbing MDP* technique. We explain this technique in detail in the next section.

On the other hand, characterizing policy class through the distance in value (like $\Pi_l$) is more flexible than characterizing the distance between policies themselves (*e.g.* via total variation). This is because: if two policies are "close", then their values are also similar; but the reverse may not be true since two very different policies could possibly generate similar values. Therefore the consideration of $\Pi_l$ is generic and conceptually reflects the fundamental principle of RL: as long as two policies yield the same value, they are considered "equally good", no matter how different they are.[6]

Most importantly, Theorem 4.1 guarantees near-minimax optimal offline learning:

**Corollary 4.2** (optimal offline learning). *If $\epsilon_{opt} \leq \sqrt{H/S}$ and that $\sup_t ||\widehat{V}_t^{\widehat{\pi}} - \widehat{V}_t^{\widehat{\pi}^*}||_\infty \leq \epsilon_{opt}$, when $n > O(H \cdot \iota/d_m)$, then with probability $1 - \delta$, element-wisely,*

$$V_1^\star - V_1^{\widehat{\pi}} \leq C \left[ \sqrt{\frac{H^2 \iota}{nd_m}} + \frac{H^{2.5} S^{0.5} \iota}{nd_m} \right] \mathbf{1} + \epsilon_{opt} \mathbf{1}.$$

Corollary 4.2 first establishes the minimax rate for offline learning for any policy $\widehat{\pi}$ with the measurable gap $\epsilon_{\text{opt}} \leq \sqrt{H/S}$. This extends the standard concept of offline learning by allowing any empirical planning algorithm (*e.g.* VI/PI) to find an *inexact* $\widehat{\pi}$ as an $(\tilde{O}\sqrt{H^2/nd_m} + \epsilon_{\text{opt}})$-optimal policy (instead of finding exact $\widehat{\pi}^\star$). The use of *inexact* $\widehat{\pi}$ could encourage early stopping (*e.g.* for VI/PI) therefore saves computational iterations. Besides, we leverage full data to construct empirical MDP for planning and, on the contrary, Yin et al. [2021b] uses data-splitting (split data into mini-batches and only apply each mini-batch at each specific iteration) to enable Variance Reduction technique, which could cause inefficient data use for the practical purpose. By the following lower bound result from Yin et al. [2021b], our Corollary 4.2 is near minimax optimal.

**Theorem 4.3** (Theorem 4.2. Yin et al. [2021b]). *Let $\mathcal{M}_{d_m}$ be the same as Theorem 3.1. There exists universal constants $c_1, c_2, c, p$ (with $H, S, A \geq c_1$ and $0 < \epsilon < c_2$) such that when $n \leq cH^2/d_m \epsilon^2$,[7]*

$$\inf_{V_1^{\pi_{alg}}} \sup_{(\mu, M) \in \mathcal{M}_{d_m}} \mathbb{P}_{\mu, M} \left( ||V_1^\star - V_1^{\pi_{alg}}||_\infty \geq \epsilon \right) \geq p.$$

For the rest of the section, we explain the proving ideas by introducing the *singleton-absorbing MDP* technique and the full proofs of Theorem 4.1, Corollary 4.2 can be found in Appendix B, D.

## 4.3 Singleton absorbing MDP for finite horizon MDP

For the ease of illustration, we explain our idea via bounding $||\widehat{Q}_h^{\widehat{\pi}^*} - Q_h^{\widehat{\pi}^*}||_\infty$ (instead of $\sup_{\pi \in \Pi_l} ||\widehat{Q}_1^\pi - Q_1^\pi||_\infty$) and choose related quantity $\widehat{\pi}^\star$ (instead of $\widehat{\pi}$) and $\widehat{V}_h^\star$ (instead of $\widehat{V}_h^{\widehat{\pi}}$) to discuss. Essentially, the key challenge in obtaining the optimal dependence in stationary setting is the need to decouple the dependence between $P - \widehat{P}$ and $\widehat{V}_h^\star$ as we aggregate all data for constructing both $\widehat{P}$ and $\widehat{V}_h^\star$. This issue is not encountered in the non-stationary setting in general due to the flexibility to estimate different transition $P_t$ at each time [Yin et al., 2021a] and $\widehat{P}_t$ and $\widehat{V}_{t+1}^\star$ preserve conditional independence. However, when confined to stationary case, their complex $\tilde{O}(H^3/d_m \epsilon^2)$ becomes suboptimal. Moreover, the direct use of $s$-absorbing MDP in Agarwal et al. [2020] does

---

[6]We recognize that in the specific settings (*e.g.* safe policy improvement) some of the policies that yield high values are not feasible. These considerations are beyond the scope of this paper.

[7]The original Theorem uses $v^\star$ but we use $V_1^\star$ here. It does not matter since we can manually add a default state at the beginning of the MDP and obtain the result for our version.

not yield tight bounds for the finite horizon stationary setting, as it requires $s$-absorbing MDPs with $H$-dimensional fine-grid cover to make sure $\widehat{V}_h^\star$ is close to one of the elements in the cover (which has size $\approx H^H$ and it is not optimal Cui and Yang [2020]). We overcome this hurdle by choosing *only one* delicate absorbing MDP to approximate $\widehat{V}_h^\star$ which will not incur additional dependence on horizon $H$ caused by the union bound. We begin with the general definition of absorbing MDP initialized in Agarwal et al. [2020] and then introduce the *singleton absorbing MDP*.

**Standard $s$-absorbing MDP in the finite horizon setting.** The general $s$-absorbing MDP is defined as follows: for a fixed state $s$ and a sequence $\{u_t\}_{t=1}^H$, MDP $M_{s,\{u_t\}_{t=1}^H}$ is identical to $M$ for all states except $s$, and state $s$ is absorbing in the sense $P_{M_{s,\{u_t\}_{t=1}^H}}(s|s,a) = 1$ for all $a$, and the instantaneous reward at time $t$ is $r_t(s,a) = u_t$ for all $a \in \mathcal{A}, t \in [H]$. For convenience, we use the shorthand notation $V_{\{s,u_t\}}^\pi$ to denote $V_{s,M_{s,\{u_t\}_{t=1}^H}}^\pi$ and similarly for $Q_t, r$ and transition $P$. Also, $V_{\{s,u_t\}}^\star$ ($Q_{\{s,u_t\}}^\star$) is the optimal value under $M_{s,\{u_t\}_{t=1}^H}$.

Before defining singleton absorbing MDP, we first present the following Lemma 4.4 and Lemma 4.5 which support the our design.

**Lemma 4.4.** $V_t^\star(s) - V_{t+1}^\star(s) \geq 0, \forall s \in \mathcal{S}, t \in [H]$.

**Lemma 4.5.** *Fix a state $s$. If we choose $u_t^\star := V_t^\star(s) - V_{t+1}^\star(s)$, then we have the following vector form equation*

$$V_{h,\{s,u_t^\star\}}^\star = V_{h,M}^\star \quad \forall h \in [H].$$

*Similarly, if we choose $\hat{u}_t^\star := \widehat{V}_t^\star(s) - \widehat{V}_{t+1}^\star(s)$, then $\widehat{V}_{h,\{s,\hat{u}_t^\star\}}^\star = \widehat{V}_{h,M}^\star, \forall h \in [H]$.*

The proofs are deferred to Appendix B. Note by Lemma 4.4 the assignment of $u_t^\star(:= r_{t,\{s,u_t^\star\}})$ is well-defined. Lemma 4.5 is crucial since, under the specification of $u_t^\star$, the optimal value in $M_{s,\{u_t^\star\}_{t=1}^H}$ is identical to the optimal value in original $M$. Based on these, we can define the following:

**Definition 4.6** (Singleton-absorbing MDP). *For each state $s$, the singleton-absorbing MDP is chosen to be $M_{s,\{u_t^\star\}_{t=1}^H}$, where $u_t^\star := V_t^\star(s) - V_{t+1}^\star(s)$ for all $t \in [H]$.*

Using Definition 4.6, for each $(s,a)$ row the term $(\widehat{P}_{s,a} - P_{s,a})\widehat{V}_h^\star$ can be substituted by $(\widehat{P}_{s,a} - P_{s,a})\widehat{V}_{h,\{s,u_t^\star\}}^\star$, where $\widehat{P}_{s,a}$ and $\widehat{V}_{h,\{s,u_t^\star\}}^\star$ are independent by construction and Bernstein concentration applies. Furthermore, by the selection of $u_t^\star$, we can control the error of $||\widehat{V}_h^\star - \widehat{V}_{h,\{s,u_t^\star\}}^\star||_\infty$ to have rate $O(\sqrt{\frac{1}{n}})$ which forces the term $(\widehat{P}_{s,a} - P_{s,a})(\widehat{V}_h^\star - \widehat{V}_{h,\{s,u_t^\star\}}^\star)$ to have higher order error. These are the critical building blocks for bounding $||\widehat{Q}_h^{\widehat{\pi}^\star} - Q_h^{\widehat{\pi}^\star}||_\infty$.

Indeed, by Bellman equations we have the decomposition: $\widehat{Q}_h^{\widehat{\pi}^\star} - Q_h^{\widehat{\pi}^\star} = \ldots = \sum_{t=h}^H \Gamma_{h+1:t}^{\widehat{\pi}^\star}(\widehat{P} - P)\widehat{V}_{t+1}^\star$, where $\Gamma_{h+1:t}^\pi = \prod_{i=h+1}^t P^{\pi_i}$ is multi-step state-action transition and $\Gamma_{h+1:h} := I$. Then for each $(s,a)$ row

$$(\widehat{P}_{s,a} - P_{s,a})\widehat{V}_h^\star = (\widehat{P}_{s,a} - P_{s,a})(\widehat{V}_h^\star - \widehat{V}_{h,\{s,u_t^\star\}}^\star) + (\widehat{P}_{s,a} - P_{s,a})\widehat{V}_{h,\{s,u_t^\star\}}^\star$$

$$\lesssim ||\widehat{P}_{s,a} - P_{s,a}||_1 ||\widehat{V}_h^\star - \widehat{V}_{h,\{s,u_t^\star\}}^\star||_\infty + \sqrt{\frac{\mathrm{Var}_{s,a}(\widehat{V}_{h,\{s,u_t^\star\}}^\star)}{n_{s,a}}} \lesssim \sqrt{\frac{S}{n_{s,a}}} \left\|\widehat{V}_h^\star - \widehat{V}_{h,\{s,u_t^\star\}}^\star\right\|_\infty + \sqrt{\frac{\mathrm{Var}_{s,a}(\widehat{V}_h^\star)}{n_{s,a}}} \quad (\star)$$
$$(2)$$

where $(\star)$ is the place where the traditional technique uses the union bound over their *exponential large $\epsilon$-net* and we do not have it! Next, by Lemma 4.5 and Lemma B.2 in Appendix

$$||\widehat{V}_h^\star - \widehat{V}_{h,\{s,u_t^\star\}}^\star||_\infty = ||\widehat{V}_{h,\{s,\hat{u}_t^\star\}}^\star - \widehat{V}_{h,\{s,u_t^\star\}}^\star||_\infty \leq H \max_t |\hat{u}_t^\star - u_t^\star| \leq 2H \max_t |\widehat{V}_t^\star - V_t^\star|,$$

by a crude bound (Lemma J.10), $\max_t |\widehat{V}_t^\star - V_t^\star| \lesssim H^2 \sqrt{\frac{S}{n_{s,a}}}$ which makes $\sqrt{\frac{1}{n_{s,a}}}||\widehat{V}_h^\star - \widehat{V}_{h,\{s,u_t^\star\}}^\star||_\infty$ have order $1/n_{s,a}$. Finally, to reduce the horizon dependence we apply $\sum_{t=h}^H \Gamma_{h+1:t}^\pi \sqrt{\mathrm{Var}_{s,a}(V_{t+1}^\pi)} \leq \sqrt{(H-h)^3}$ for any $\pi$. This (informally) bounds $\widehat{Q}_h^{\widehat{\pi}^\star} - Q_h^{\widehat{\pi}^\star}$ by $||\widehat{Q}_h^{\widehat{\pi}^\star} - Q_h^{\widehat{\pi}^\star}||_\infty \lesssim \sqrt{\frac{H^3}{n_{s,a}}} + \frac{Poly(H,S)}{n_{s,a}}$. Lastly, use $\min_{s,a} n_{s,a} \gtrsim H \cdot d_m$ to finish the proof.

**Remark 4.7.** *We emphasize the appropriate selection of $M_{s,\{u_t^\star\}_{t=1}^H}$ ($\widehat{M}_{s,\{u_t^\star\}_{t=1}^H}$) is the key for achieving optimality. It guarantees two things: 1. $\widehat{V}_{h,\{s,u^\star\}}^\star$ approximates $\widehat{V}_h^\star$ with sufficient accuracy (has rate $\sqrt{1/n_{s,a}}$); 2. it avoids the fine-grid design with exponential union bound in the dominate term ($\sqrt{\frac{\mathrm{Var}_{s,a}(\widehat{V}_h^\star)\log(|U_{s,a}|/\delta)}{N}}$ with $|U_{s,a}|$ to be at least $H^H$ Cui and Yang [2020].)*

## 5 New settings: offline Task-agnostic and offline Reward-free learning

From Corollary 4.2, our model-based offline learning algorithm has two steps: 1. constructing offline empirical MDP $\widehat{M}$ using the offline dataset $\mathcal{D} = \{(s_t^{(i)}, a_t^{(i)}, r(s_t^{(i)}, a_t^{(i)}), s_{t+1}^{(i)})\}_{i\in[n]}^{t\in[H]}$; 2. performing any accurate black-box *planning* algorithm and returning $\widehat{\pi}^\star$(or $\widehat{\pi}$) as the final output. However, the only *effective* data (data that contains stochasticity) is $\mathcal{D}' = \{(s_t^{(i)}, a_t^{(i)})\}_{i\in[n]}^{t\in[H]}$. This indicates we are essentially using the state-action space exploration data $\mathcal{D}'$ to solve the task-specific problem with reward $r$. With this perspective in mind, it is natural to ask: given only the offline exploration data $\mathcal{D}'$, can we efficiently learn a set of potentially conflicting $K$ tasks ($K$ rewards) simultaneously? Even more, can we efficiently learn all tasks simultaneously? This brings up the following definitions.

**Definition 5.1** (Offline Task-agnostic Learning). *Given a offline exploration datatset $\mathcal{D}' = \{(s_t^{(i)}, a_t^{(i)})\}_{i\in[n]}^{t\in[H]}$ by $\mu$ with $n$ episodes. Given $K$ tasks with reward $\{r_k\}_{k=1}^K$ and the corresponding $K$ MDPs $M_k = (\mathcal{S}, \mathcal{A}, P, r_k, H, d_1)$. Can we use $\mathcal{D}'$ to output $\hat{\pi}_1, \ldots, \hat{\pi}_K$ such that $\mathbb{P}\left[\forall r_k, k \in [K], \left\|V_{1,M_k}^\star - V_{1,M_k}^{\hat{\pi}_k}\right\|_\infty \leq \epsilon\right] \geq 1 - \delta$?*

**Definition 5.2** (Offline Reward-free Learning). *Given a offline exploration datatset $\mathcal{D}' = \{(s_t^{(i)}, a_t^{(i)})\}_{i\in[n]}^{t\in[H]}$ by $\mu$ with $n$ episodes. For any reward $r$ and the corresponding MDP $M = (\mathcal{S}, \mathcal{A}, P, r, H, d_1)$. Can we use $\mathcal{D}'$ to output $\hat{\pi}$ such that $\mathbb{P}\left[\forall r, \left\|V_{1,M}^\star - V_{1,M}^{\hat{\pi}}\right\|_\infty \leq \epsilon\right] \geq 1 - \delta$?*

Definition 5.1 and Definition 5.2 are the offline counterparts of Zhang et al. [2020b] and Jin et al. [2020a] in online RL. Those settings are of practical interests in the offline regime as well since in practice reward functions are often iteratively engineered to encourage desired behavior via trial and error and using one shot of offline exploration data $\mathcal{D}'$ to tackle problems with different reward functions (different tasks) could help improve sample efficiency significantly.

Our singleton absorbing MDP technique adapts to those settings and we have the following two theorems. The proofs of Theorem 5.3, 5.4 can be found in Appendix E, F.

**Theorem 5.3** (optimal offline task-agnostic learning). *Given $\mathcal{D}' = \{(s_t^{(i)}, a_t^{(i)})\}_{i\in[n]}^{t\in[H]}$ by $\mu$. Given $K$ tasks with reward $\{r_k\}_{k=1}^K$ and the corresponding $K$ MDPs $M_k = (\mathcal{S}, \mathcal{A}, P, r_k, H, d_1)$. Denote $\iota = \log(HSA/\delta)$. Let $\widehat{\pi}_k^\star := \arg\max_\pi \widehat{V}_{1,M_k}^\pi \ \forall k \in [K]$, when $n > O(H \cdot [\iota + \log(K)]/d_m)$, then with probability $1-\delta$, $\left\|V_{1,M_k}^\star - V_{1,M_k}^{\widehat{\pi}_k^\star}\right\|_\infty \leq O\left[\sqrt{\frac{H^2(\iota+\log(K))}{nd_m}} + \frac{H^{2.5}S^{0.5}(\iota+\log(K))}{nd_m}\right] \ \forall k \in [K]$.*

**Theorem 5.4** (optimal offline reward-free learning). *Given $\mathcal{D}' = \{(s_t^{(i)}, a_t^{(i)})\}_{i\in[n]}^{t\in[H]}$ by $\mu$. For any reward $r$ denote the corresponding MDP $M = (\mathcal{S}, \mathcal{A}, P, r, H, d_1)$. Denote $\iota = \log(HSA/\delta)$. Let $\widehat{\pi}_M^\star := \arg\max_\pi \widehat{V}_{1,M}^\pi \ \forall r$, when $n > O(HS \cdot \iota/d_m)$, then with probability $1 - \delta$, $\left\|V_{1,M}^\star - V_{1,M}^{\widehat{\pi}_M^\star}\right\|_\infty \leq O\left[\sqrt{\frac{H^2 S \cdot \iota}{nd_m}} + \frac{H^2 S \cdot \iota}{nd_m}\right], \ \forall r, M$.*

By a direct translation of both theorems, we have sample complexity of order $\widetilde{O}(H^2 \log(K)/d_m\epsilon^2)$ and $\widetilde{O}(H^2S/d_m\epsilon^2)$. All the parameters have the optimal rates, see the lower bounds in Zhang et al. [2020b] and Jin et al. [2020a].[8] The higher order dependence in Theorem 5.4 is also tight comparing to Theorem 5.3. Such statistically optimal results reveal the model-based methods generalize well to those seemingly challenging problems in the offline regime. Changing to these harder problems would not affect the optimal statistical efficiency of the model-based approach.

---

[8]We add a discussion in Appendix G to explain more clearly why our rates are optimal for these problems.

# 6 Extension to linear MDP with anchor representations

The principle of our *Singleton absorbing MDP* technique (with model-based construction) in decoupling the dependence between $\widehat{P}_{s,a}$ and $\widehat{V}^\star$ is not confined to tabular MDPs and therefore it is natural to generalize such idea for the episodic stationary transition setting for other problems. As an example, we further present a sharp result for the setting of finite horizon linear MDP with anchor points. We narrate by assuming a generative oracle (that allows sampling from $s' \sim P(\cdot|s,a)$) for the ease of exposition.

**Definition 6.1** (Linear MDP with anchor points [Yang and Wang, 2019, Cui and Yang, 2020]). *Let $\mathcal{S}$ be the exponential large space and $\mathcal{A}$ be the infinite (or even continuous) spaces. Assume there is feature map $\phi : \mathcal{S} \times \mathcal{A} \to \mathbb{R}^K$ (where $K \ll |\mathcal{S}|$), i.e. $\phi(s,a) = [\phi_1(s,a), \ldots, \phi_K(s,a)]$. Transition $P$ admits a linear representation: $P(s'|s,a) = \sum_{k \in [K]} \phi_k(s,a)\psi_k(s')$ where $\psi_1(\cdot), \ldots, \psi_K(\cdot)$ are unknowns. We further assume there exists a set of anchor state-action pairs $\mathcal{K}$ such that any $(s,a)$ can be represented as a convex combination of the anchors $\{(s_k, a_k)|k \in \mathcal{K}\}$:*

$$\exists \{\lambda_k^{s,a}\} : \phi(s,a) = \sum_{k \in \mathcal{K}} \lambda_k^{s,a} \phi(s_k, a_k), \sum_{k \in \mathcal{K}} \lambda_k^{s,a} = 1, \lambda_k \geq 0, \forall k \in \mathcal{K}, (s,a) \in (\mathcal{S}, \mathcal{A}).$$

Under the definition, denote $N$ be the number of samples at each anchor pairs. Then we have the following (see Appendix H for the proof):

**Theorem 6.2** (Optimal sample complexity). *Under Definition 6.1, let $\widehat{\pi}^\star = \mathrm{argmax}_\pi \widehat{V}_1^\pi$. Then if $N \geq cH^2|\mathcal{S}|\log(KH/\delta)$, we have with probability $1 - \delta$, $||Q_1^\star - Q_1^{\widehat{\pi}^\star}||_\infty \leq \widetilde{O}(\sqrt{H^3/N})$.*

Comparing to Theorem 4 of Cui and Yang [2020], Theorem 6.2 removes the additional dependence $\min\{|S|, K, H\}$. In term of the total sample complexity, Theorem 6.2 gives $\tilde{O}(KH^3/\epsilon^2)$ while Cui and Yang [2020] has $\widetilde{O}(KH^4/\epsilon^2)$ (see their Section 7, first bullet point). Our result again reveals the model-based method is statistically optimal for the current setting.

**Remark 6.3.** *The rate $\tilde{O}(KH^3/\epsilon^2)$ with anchor point assumption has the linear dependence on $K$ and for the standard linear bandit [Lattimore and Szepesvári, 2020] $\Omega(\sqrt{d^2T})$ or the linear (mixture) MDP [Jin et al., 2020b, Zhou et al., 2020] $\Omega(\sqrt{d^2H^2T})$ the lower bound dependence on the feature dimension $d$ is quadratic. We believe one reason for this to happen is that anchor representations assumption is somewhat strong as it abstracts the whole state action space by only finite points (via convex combination).*

# 7 Conclusion and Future Works

This work studies the uniform convergence problems for offline policy evaluation (OPE) and provides complete answers for their optimality behaviors. We achieve the optimal sample complexity for stationary-transition case using a novel adaptation of the absorbing MDP trick, which is more generally applicable to the new offline task-agnostic and reward-free settings combining with the model-based approach and we hope it can be applied to a broader range of future problems. We end the section by two future directions.

**On the higher order error term.** Our main result (Theorem 4.1) has an additional $\sqrt{HS}$ dependence in the higher order error term and we cannot further remove it based on our current technique. Nevertheless, this is already among the best higher order results to our knowledge. In fact, most state-of-the-art works (*e.g.* Azar et al. [2017], Dann et al. [2019], Zhang et al. [2021b]) have additional $S$ dependence in the higher order and Jin et al. [2018] has only extra $\sqrt{S}$ in the higher order term but it also has additional $\sqrt{A}$ (see Table 1 of Zhang et al. [2021b] for a clear reference). How to obtain optimality not only for the main term but also for the higher order error terms remains elusive for the community.

**Uniform OPE and beyond.** The current study of uniform OPE derives results with expression using parameter dependence and deriving instance-dependent uniform convergence result will draw a clearer picture on the individual behaviors for each policy. Besides, this work concentrates on Tabular MDPs and generalizing uniform convergence to more practical settings like linear MDPs, game environments and multi-agent settings are promising future directions. Specifically, general complexity measure (mirroring VC-dimensions and Rademacher complexities for statistical learning problems) that precisely captures local and global uniform convergence would be of great interest.

**Acknowledgment**

The research is partially supported by NSF Awards #2007117 and #2003257. MY would like to thank Simon S. Du, Zihan Zhang for explaining a question in Zhang et al. [2020c]; and Tongzheng Ren for the helpful discussions.

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
