# Appendix

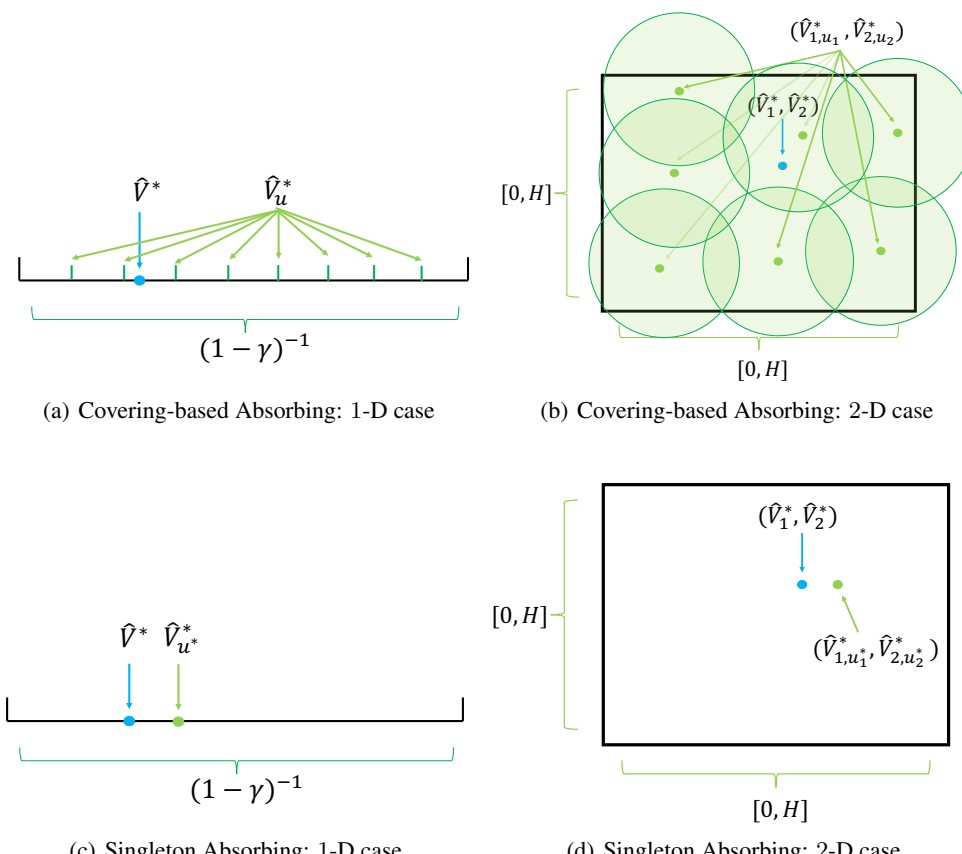

(a) Covering-based Absorbing: 1-D case

(b) Covering-based Absorbing: 2-D case

(c) Singleton Absorbing: 1-D case

(d) Singleton Absorbing: 2-D case

Figure 2: Visualization of singleton absorbing MDP technique

We provide a visualization (Figure 2) for understanding the *singleton absorbing MDP* technique at the beginning of Appendix. 2(a), 2(c) demonstrate the infinite horizon case and 2(b), 2(d) demonstrate the finite horizon case. In particular, it should be a $H$-dimensional hypercube $[0, H]^H$ (that contains $\widehat{V}_1^\star, \ldots, \widehat{V}_h^\star$) instead of only the square $[0, H] \times [0, H]$ ($\widehat{V}_1^\star, \widehat{V}_2^\star$). This is only for the ease of visualization.

The standard absorbing MDP technique Agarwal et al. [2020], Cui and Yang [2020] leverages a set of absorbing MDPs to cover the range of value functions (following the standard covering principle) to make sure $\widehat{V}^\star$ is close to one of the element (absorbing MDP) in the set (Figure 2(a),2(b)). The size of the covering set (*i.e.* the covering number) grows exponentially in $H$ 2(b) in the finite horizon setting and this is due to the fact that there are $\widehat{V}_1^\star, \widehat{V}_2^\star, \ldots, \widehat{V}_H^\star$ quantities to cover. This results in the metric entropy (the $\log$ of the covering number) to blow up by a factor of $H$ and incurs suboptimality. On the other hand, by the nifty chosen singleton absorbing MDP $\widehat{V}_{h,u^\star}^\star$ (Figure 2(c),2(d)), we completely get rid of the covering issue (covering the $H$-dimensional space requires exponential in $H$ size), maintain the independence and control the error propagation ($\left\|\widehat{V}^\star - \widehat{V}_{u^\star}^\star\right\|_\infty$ is sufficiently small). See Section B for all the technical details.

## A    Discussion on Related works

**Offline reinforcement learning.** Information-theoretical considerations for offline RL are first proposed for *infinite horizon discounted setting* via Fitted Q-Iteration (FQI) type function approximation

algorithms [Chen and Jiang, 2019, Le et al., 2019, Xie and Jiang, 2021, 2020] which can be traced back to [Munos, 2003, Szepesvári and Munos, 2005, Antos et al., 2008a,b]. Later, Xie and Jiang [2021] considers the offline RL under only the *realizability* assumption and Liu et al. [2020b] considers the offline RL *without good exploration*. Those are all challenging problems but with they only provide suboptimal polynomial complexity in terms of $(1-\gamma)^{-1}$.

For the finite horizon case, Yin et al. [2021a] first achieves $\tilde{O}(H^3/d_m\epsilon^2)$ complexity under non-stationary transition but their results cannot be further improved in the stationary setting. Concurrent to our work, a recently released work Yin et al. [2021b] designs the offline variance reduction algorithm for achieving the optimal $\tilde{O}(H^2/d_m\epsilon^2)$ rate. Their result is for a specific algorithm that uses data splitting while our results work for any algorithms that returns a nearly empirically optimal policy via a uniform convergence guarantee. Our results on the offline task-agnostic and the reward-free settings are entirely new. Another concurrent work Ren et al. [2021] considers the horizon-free setting but does not provide uniform convergence guarantee. Even more recently, Rashidinejad et al. [2021] considers the single concentrability coefficient $C^\star := \max_{s,a} \frac{d^{\pi^\star}(s,a)}{d^\mu(s,a)}$ and obtains the sample complexity $\tilde{O}[(1-\gamma)^{-5}SC^\star/\epsilon^2]$. Concurrently, Yin and Wang [2021] first derives the instance-dependent offline RL bounds, but their result is for time-inhomogeneous MDPs.

In the linear MDP case, Jin et al. [2020c] studies the pessimism-based algorithms for offline policy optimization under the weak compliance assumption and Wang et al. [2021], Zanette [2021] provide some negative results (exponential lower bound) for offline RL with linear MDP structure.

**Model-based approaches with minimaxity.** It is known model-based methods are minimax-optimal for online RL with regret $\tilde{O}(\sqrt{HSAT})$ (*e.g.* Azar et al. [2017], Efroni et al. [2019]). For linear MDP, In the generative model setting, Agarwal et al. [2020] shows model-based approach is still minimax optimal $\tilde{O}((1-\gamma)^{-3}SA/\epsilon^2)$ by using a $s$-absorbing MDP construction and this model-based technique is later reused for other more general settings (*e.g.* Markov games [Zhang et al., 2020a] and linear MDPs [Cui and Yang, 2020]) and also for improving the sample size barrier [Li et al., 2020]. In offline RL, Yu et al. [2020], Kidambi et al. [2020] use model-based approaches for continuous policy optimization and Yin et al. [2021a] uses the model-based methods to achieve $\tilde{O}(H^3/d_m\epsilon^2)$ complexity.

**Task-agnostic and Reward-free problems.** The reward-free problem is initiated in the online RL [Jin et al., 2020a] where the agent needs to efficiently explore an MDP environment *without* using any reward information. It requires high probability guarantee for learning optimal policy for *any* reward function, which is strictly stronger than the standard learning task that one only needs to learn to optimal policy for a fixed reward. Later, Kaufmann et al. [2020], Menard et al. [2020] establish the $\tilde{O}(H^3S^2A/\epsilon^2)$ complexity and Zhang et al. [2020c] further tightens the dependence to $\tilde{O}(H^2S^2A/\epsilon^2)$.[9] Recently, Zhang et al. [2020b] proposes the task-agnostic setting where one needs to use exploration data to simultaneously learn $K$ tasks and provides a upper bound with complexity $\tilde{O}(H^5SA\log(K)/\epsilon^2)$. For linear MDP setting, Wang et al. [2020] achieves the sample complexity $\tilde{O}(d^3H^6/\epsilon^2)$ and Liu et al. [2020a] considers such problem in the online two-player Markov game. However, although these settings remain critical in the offline regime, no statistical result has been formally derived so far.

# B   Proof of optimal local uniform convergence

## B.1   Model-based Offline Plug-in Estimator

Recall the model-based estimator uses empirical estimator $\widehat{P}$ for estimating $P$ and the estimator is calculated accordingly:

$$\widehat{Q}_h^\pi = r + \widehat{P}^{\pi_{h+1}}Q_{h+1}^\pi = r + \widehat{P}V_{H+1}^\pi,$$

where $\widehat{P}(s'|s,a)$ can be expressed as:

---

[9]We translate [Zhang et al., 2020c] their dimension-free result to $\tilde{O}(H^2S^2A/\epsilon^2)$ under the standard assumption $r \in [0,1]$.

$$\widehat{P}(s'|s,a) = \frac{\sum_{i=1}^{n}\sum_{h=1}^{H}\mathbf{1}[(s_{h+1}^{(i)}, a_h^{(i)}, s_h^{(i)}) = (s', s, a)]}{n_{s,a}}, \quad n_{s,a} = \sum_{h=1}^{H}\sum_{i=1}^{n}\mathbf{1}[(s_h^{(i)}, a_h^{(i)}) = (s, a)].$$

and $\widehat{P}(s'|s,a) = \frac{1}{S}$, if $n_{s,a} = 0$. The initial distribution is also constructed as $\widehat{d}_1^\pi(s) = n_s/n$.

First of all, we have by definition the Bellman optimality equation

$$V_t^\star(s) = \max_a \left\{ r(s,a) + \sum_{s'} P(s'|s,a)V_{t+1}^\star(s') \right\}, \quad \forall s \in \mathcal{S}. \tag{3}$$

and similarly the empirical version

$$\widehat{V}_t^\star(s) = \max_a \left\{ r(s,a) + \sum_{s'} \widehat{P}(s'|s,a)\widehat{V}_{t+1}^\star(s') \right\}, \quad \forall s \in \mathcal{S}.$$

The key difficulty in obtaining the optimal dependence in stationary setting is decoupling the dependence of $P - \widehat{P}$ and $\widehat{V}^\star$. This issue is not encountered in the non-stationary setting due to the possibility to estimate different transition at each time [Yin et al., 2021a], but it cannot further reduce the sample complexity on $H$. Moreover, the direct use of $s$-absorbing MDP in Agarwal et al. [2020] is not sharp for finite horizon stationary setting, as it requires $s$-absorbing MDPs with $H$-dimensional cover (which has size $\approx e^H$ and it is not optimal). We design the *singleton-absorbing MDP* to get rid of the issue.

## B.2 General absorbing MDP

The general absorbing MDP is defined as follows: for a fixed state $s$ and a sequence $\{u_t\}_{t=1}^H$, MDP $M_{s,\{u_t\}_{t=1}^H}$ is identical to $M$ for all states except $s$, and state $s$ is absorbing in the sense $P_{M_{s,\{u_t\}_{t=1}^H}}(s|s,a) = 1$ for all $a$, and the instantaneous reward at time $t$ is $r_t(s,a) = u_t$ for all $a \in \mathcal{A}$. Also, we use the shorthand notation $V_{\{s,u_t\}}^\pi$ for $V_{s,M_{s,\{u_t\}_{t=1}^H}}^\pi$ and similarly for $Q_{\{s,u_t\}}$ and transition $P_{\{s,u_t\}}$. Then the following properties hold:

**Lemma B.1.**

$$V_{h,\{s,u_t\}}^\star(s) = \sum_{t=h}^H u_t.$$

*Proof.* We prove by backward induction. For $h = H$, under $M_{s,\{u_t\}_{t=1}^H}$ state $s$ is absorbing (and by convention $V_{H+1,\{s,u_t\}}^\star = 0$) therefore

$$V_{H,\{s,u_t\}}^\star(s) = \max_a \left\{ r_{H,\{s,u_t\}}(s,a) + \sum_{s'} P_{\{s,u_t\}}(s'|s,a)V_{H+1,\{s,u_t\}}^\star(s') \right\} = \max_a \left\{ r_{H,\{s,u_t\}}(s,a) \right\} = u_H$$

for general $h$, note $\sum_{s'} P_{\{s,u_t\}}(s'|s,a)V_{h+1,\{s,u_t\}}^\star(s') = 1 \cdot V_{h+1,\{s,u_t\}}^\star(s)$, therefore using induction property $V_{h+1,\{s,u_t\}}^\star(s) = \sum_{t=h+1}^H u_t$ we can similarly obtain $V_{h,\{s,u_t\}}^\star(s) = \sum_{t=h}^H u_t$. ∎

**Lemma B.2.** *Fix state $s$. For two different sequences $\{u_t\}_{t=1}^H$ and $\{u_t'\}_{t=1}^H$, we have*

$$\max_h \left\| Q_{h,\{s,u_t\}}^\star - Q_{h,\{s,u_t'\}}^\star \right\|_\infty \le H \cdot \max_{t\in[H]} |u_t - u_t'|.$$

*Proof.* Let $\pi^\star_{\{s,u_t\}}$ be the optimal policy in $M_{\{s,u_t\}}$. Then (by convention $\prod_{a=h+1}^h P^{\pi_a} = I$)

$$Q^\star_{h,\{s,u_t\}} - Q^\star_{h,\{s,u'_t\}} = Q^\star_{h,\{s,u_t\}} - \max_\pi \sum_{i=h}^H \left( \prod_{a=h+1}^i P^{\pi_a}_{\{s,u'_t\}} \right) r_{i,\{s,u'_t\}}$$

$$\leq Q^\star_{h,\{s,u_t\}} - \sum_{i=h}^H \left( \prod_{a=h+1}^i P^{\pi^\star_{a,\{s,u_t\}}}_{\{s,u'_t\}} \right) r_{i,\{s,u'_t\}} = \sum_{i=h}^H \left( \prod_{a=h+1}^i P^{\pi^\star_{a,\{s,u_t\}}}_{\{s,u'_t\}} \right) \left( r_{i,\{s,u_t\}} - r_{i,\{s,u'_t\}} \right)$$

$$\leq \sum_{i=h}^H \max_{s,a} \left\| \left( \prod_{a=h+1}^i P^{\pi^\star_{a,\{s,u_t\}}}_{\{s,u'_t\}} \right)^{i-h} (\cdot|s,a) \right\|_1 \cdot \left\| r_{i,\{s,u_t\}} - r_{i,\{s,u'_t\}} \right\|_\infty \cdot \mathbf{1} = (H-h+1) \cdot \max_t |u_t - u'_t| \cdot \mathbf{1}$$

where the first equal sign uses the definition of $Q^\star$, the second equal sign uses $P_{\{s,u_t\}}$ only depends $s$ but not the specification of $u_t$'s and the last equal sign comes from $r_{i,\{s,u_t\}}(s,a) = u_i$ for any $a \in \mathcal{A}$ and $r_{i,\{s,u_t\}}(\tilde{s},a) = r_{i,\{s,u'_t\}}(\tilde{s},a)$ for any $\tilde{s} \neq s$. Lastly by symmetry we finish the proof. ∎

### B.3 Singleton-absorbing MDP

The direct of transfer of absorbing technique created in Agarwal et al. [2020] will require each $u_t$ to fill in the range of $[0, H]$ using evenly spaced elements. For finite horizon MDP there are $H$ layers, therefore the total number of $H$-tuples $(u_1, \ldots, u_H)$ has order $|U_s| = Poly(H)^H$, therefore when apply the union bound, it will incur the additional $H$ factor. We get rid of this issue by choosing one single point in $H$-dimensional space $[0, H]^H$. We first give the following two lemmas.

**Lemma B.3.** $V^\star_t(s) - V^\star_{t+1}(s) \geq 0$, for all state $s \in \mathcal{S}$ and all $t \in [H]$.

*Proof.* Let the optimal policy for $V^\star_{t+1}$ be $\pi^\star_{t+1:H}$, i.e. $V^\star_{t+1} = V^{\pi^\star_{t+1:H}}_{t+1}$, then artificially construct a policy $\pi_{t:H}$ such that $\pi_{t:H-1} = \pi^\star_{t+1:H}$ and $\pi_H$ is arbitrary, then by the definition of optimal value

$$V^\star_t(s) \geq V^{\pi_{t:H}}_t(s) = \mathbb{E}^{\pi_{t:H}} \left[ \sum_{i=t}^H r(s_i, a_i) \middle| s_t = s \right]$$

$$= \mathbb{E}^{\pi_{t:H-1}} \left[ \sum_{i=t}^{H-1} r(s_i, a_i) \middle| s_t = s \right] + \mathbb{E}^{\pi_{t:H}} \left[ r(s_H, a_H)|s_t = s \right]$$

$$= \mathbb{E}^{\pi^\star_{t+1:H}} \left[ \sum_{i=t+1}^H r(s_i, a_i) \middle| s_{t+1} = s \right] + \mathbb{E}^{\pi_{t:H}} \left[ r(s_H, a_H)|s_t = s \right]$$

$$\geq \mathbb{E}^{\pi^\star_{t+1:H}} \left[ \sum_{i=t+1}^H r(s_i, a_i) \middle| s_{t+1} = s \right] + 0 = V^\star_{t+1}(s),$$

where the third equal sign uses exactly that *P is a STATIONARY transition* and definition $\pi_{t:H-1} = \pi^\star_{t+1:H}$. The last inequality uses assumption that reward is always non-negative. ∎

**Remark B.4.** *Lemma B.3 leverages P is stationary and above may not be true in the non-stationary setting. This enables us to establish the following lemma, which is the key for singleton-absorbing MDP.*

**Lemma B.5.** *Fix a state $s$. If we choose $u^\star_t := V^\star_t(s) - V^\star_{t+1}(s) \, \forall t \in [H]$, then we have the following vector form equation*

$$V^\star_{h,\{s,u^\star_t\}} = V^\star_{h,M} \quad \forall h \in [H].$$

*Similarly, if we choose $\widehat{u}^\star_t := \widehat{V}^\star_t(s) - \widehat{V}^\star_{t+1}(s)$, then $\widehat{V}^\star_{h,\{s,\hat{u}^\star_t\}} = \widehat{V}^\star_{h,M}, \forall h \in [H].$*

*Proof.* We focus on the first claim. Note by Lemma B.3 the assignment of $u^\star_t(:= r_{t,\{s,u^\star_t\}})$ is well-defined. Next recall $V^\star_{h,M}$ is the optimal value under true MDP $M$ and $V^\star_{h,\{s,u^\star_t\}}$ is the optimal value under the assimilating MDP $M_{s,\{u^\star_t\}^H_{t=1}}$. We prove by backward induction.

For $h = H$, note by convention $V^\star_{H+1} = 0$, therefore $u^\star_H = V^\star_H(s) - V^\star_{H+1}(s) = V^\star_H(s) - 0 = V^\star_H(s)$ and Bellman optimality equation becomes

$$V^\star_H(\tilde{s}) = \max_a \{r(\tilde{s}, a)\}, \quad \forall \tilde{s} \in \mathcal{S}.$$

Under $M_{s, \{u^\star_t\}^H_{t=1}}$, for state $s$ by Lemma B.1 we have $V^\star_{H, \{s, u^\star_t\}}(s) = u^\star_H = V^\star_H(s)$, for other states $\tilde{s} \neq s$, reward in $M_{s, \{u^\star_t\}^H_{t=1}} = M$ so we also have $V^\star_{H, \{s, u^\star_t\}}(\tilde{s}) = V^\star_H(\tilde{s})$ for all $\tilde{s} \neq s$.

Now for general $h$, for state $s$ by Lemma B.1

$$V^\star_{h, \{s, u^\star_t\}}(s) = \sum_{t=h}^H u^\star_t = \sum_{t=h}^H \left(V^\star_t(s) - V^\star_{t+1}(s)\right) = V^\star_h(s),$$

for state $\tilde{s} \neq s$, by Bellman optimality equation

$$V^\star_{h, \{s, u^\star_t\}}(\tilde{s}) = \max_a \left\{ r_{\{s, u^\star_t\}}(\tilde{s}, a) + \sum_{s'} P_{\{s, u^\star_t\}}(s'|\tilde{s}, a) V^\star_{h+1, \{s, u^\star_t\}}(s') \right\}$$

$$= \max_a \left\{ r(\tilde{s}, a) + \sum_{s'} P(s'|\tilde{s}, a) V^\star_{h+1, \{s, u^\star_t\}}(s') \right\}$$

$$= \max_a \left\{ r(\tilde{s}, a) + \sum_{s'} P(s'|\tilde{s}, a) V^\star_{h+1}(s') \right\} = V^\star_h(\tilde{s}),$$

where the second equal sign uses when $\tilde{s} \neq s$, $M_{s, \{u^\star_t\}^H_{t=1}}$ is identical to $M$ and the third equal sign uses induction assumption that element-wisely $V^\star_{h+1, \{s, u^\star_t\}} = V^\star_{h+1}$. Similar result can be derived for $\hat{u}^\star$ version and this completes the proof. ∎

The singleton MDP we used is exactly $M_{s, \{u^\star_t\}^H_{t=1}}$ (or $\widehat{M}_{s, \{u^\star_t\}^H_{t=1}}$).

## B.4  Proof for local uniform convergence

Recall the local policy class

$$\Pi_l := \left\{ \pi : \text{ s.t. } \left\| \widehat{V}^\pi_h - \widehat{V}^{\widehat{\pi}^\star}_h \right\|_\infty \leq \epsilon_{\text{opt}}, \forall h \in [H] \right\}.$$

For ease of exposition, we denote $N := \min_{s,a} n_{s,a}$. Note $N$ itself is a random variable, therefore for the rest of proof we first conditional on $N$. Later we shall remove the conditional on $N$ (see Section B.7).

For any $\widehat{\pi} \in \Pi_l$, by (empirical) Bellman equation we have element-wisely:

$$\widehat{Q}^{\widehat{\pi}}_h - Q^{\widehat{\pi}}_h = r_h + \widehat{P}^{\widehat{\pi}_{h+1}} \widehat{Q}^{\widehat{\pi}}_{h+1} - r_h - P^{\widehat{\pi}_{h+1}} Q^{\widehat{\pi}}_{h+1}$$

$$= \left( \widehat{P}^{\widehat{\pi}_{h+1}} - P^{\widehat{\pi}_{h+1}} \right) \widehat{Q}^{\widehat{\pi}}_{h+1} + P^{\widehat{\pi}_{h+1}} \left( \widehat{Q}^{\widehat{\pi}}_{h+1} - Q^{\widehat{\pi}}_{h+1} \right)$$

$$= \left( \widehat{P} - P \right) \widehat{V}^{\widehat{\pi}}_{h+1} + P^{\widehat{\pi}_{h+1}} \left( \widehat{Q}^{\widehat{\pi}}_{h+1} - Q^{\widehat{\pi}}_{h+1} \right)$$

$$= \ldots = \sum_{t=h}^H \Gamma^{\widehat{\pi}}_{h+1:t} \left( \widehat{P} - P \right) \widehat{V}^{\widehat{\pi}}_{t+1}$$

$$\leq \underbrace{\sum_{t=h}^H \Gamma^{\widehat{\pi}}_{h+1:t} \left| \left( \widehat{P} - P \right) \widehat{V}^{\widehat{\pi}^\star}_{t+1} \right|}_{(\star)} + \underbrace{\sum_{t=h}^H \Gamma^{\widehat{\pi}}_{h+1:t} \left| \left( \widehat{P} - P \right) \left( \widehat{V}^{\widehat{\pi}}_{t+1} - \widehat{V}^{\widehat{\pi}^\star}_{t+1} \right) \right|}_{(\star\star)}$$

where $\Gamma^\pi_{h+1:t} = \prod_{i=h+1}^t P^{\pi_i}$ is multi-step state-action transition and $\Gamma_{h+1:h} := I$.

### B.5 Analyzing ($\star\star$)

Term ($\star\star$) can be readily bounded using the following lemma.

**Lemma B.6.** *Fix $N > 0$, we have with probability $1 - \delta$, for all $t = 1, ..., H - 1$*

$$\sum_{t=h}^{H} \Gamma_{h+1:t}^{\widehat{\pi}} \left| (\widehat{P} - P)(\widehat{V}_{h+1}^{\widehat{\pi}^\star} - \widehat{V}_{h+1}^{\widehat{\pi}}) \right| \leq C \epsilon_{\text{opt}} \cdot \sqrt{\frac{H^2 S \log(SA/\delta)}{N}} \cdot \mathbf{1}$$

*where $C$ absorb the higher order term and absolute constants.*

*Proof.* First, by vector induced matrix norm[10] we have

$$\left\| \sum_{t=h}^{H} \Gamma_{h+1:t}^{\widehat{\pi}} \cdot \left| (\widehat{P} - P)(\widehat{V}_{t+1}^{\widehat{\pi}^\star} - \widehat{V}_{t+1}^{\widehat{\pi}}) \right| \right\|_{\infty} \leq H \cdot \sup_{t} \left\| \Gamma_{h+1:t}^{\widehat{\pi}} \right\|_{\infty} \left\| |(\widehat{P} - P)(\widehat{V}_{t+1}^{\widehat{\pi}^\star} - \widehat{V}_{t+1}^{\widehat{\pi}})| \right\|_{\infty}$$

$$\leq H \cdot \sup_{t} \left\| |(\widehat{P} - P)(\widehat{V}_{t+1}^{\widehat{\pi}^\star} - \widehat{V}_{t+1}^{\widehat{\pi}})| \right\|_{\infty}$$

$$= H \cdot \sup_{t,s,a} \left| (\widehat{P} - P)(\cdot|s,a)(\widehat{V}_{t+1}^{\widehat{\pi}^\star} - \widehat{V}_{t+1}^{\widehat{\pi}}) \right|$$

$$\leq H \cdot \sup_{t,s,a} \left\| (\widehat{P} - P)(\cdot|s,a) \right\|_{1} \cdot \left\| \widehat{V}_{t+1}^{\widehat{\pi}^\star} - \widehat{V}_{t+1}^{\widehat{\pi}} \right\|_{\infty} \cdot \mathbf{1}$$

where the second inequality uses multi-step transition $\Gamma_{t+1:h-1}^{\pi}$ is row-stochastic. Note given $N$, therefore by Lemma J.9 and a union bound we have with probability $1 - \delta$,

$$\sup_{s,a} \left\| (\widehat{P} - P)(\cdot|s,a) \right\|_{1} \leq C \left( \sqrt{\frac{S \log(SA/\delta)}{N}} \right),$$

(where $C$ absorb the higher order term and absolute constants) and using definition of $\Pi_l$ we have $\sup_{t} \left\| \widehat{V}_{t}^{\widehat{\pi}^\star} - \widehat{V}_{t}^{\widehat{\pi}} \right\|_{\infty} \leq \epsilon_{\text{opt}}$. This indicates

$$\sup_{t,s,a} \left\| (\widehat{P} - P)(\cdot|s,a) \right\|_{1} \cdot \left\| \widehat{V}_{t+1}^{\widehat{\pi}^\star} - \widehat{V}_{t+1}^{\widehat{\pi}} \right\|_{\infty} \cdot \mathbf{1} \leq C \left( \epsilon_{\text{opt}} \sqrt{\frac{S \log(SA/\delta)}{N}} \cdot \mathbf{1} \right),$$

where $\mathbf{1} \in \mathbb{R}^S$ is all-one vector. Then multiple by $H$ to get the stated result.

∎

### B.6 Analyzing ($\star$)

**Concentration on** $\left( \widehat{P} - P \right) \widehat{V}_{h}^{\star}$**.**[11] Since $\widehat{P}$ aggregates all data from different step so that $\widehat{P}$ and $\widehat{V}_{h}^{\star}$ are on longer independent, Bernstein inequality cannot be directly applied. We use the singleton-absorbing MDP $M_{s,\{u_t^\star\}_{t=1}^{H}}$ to handle the case (recall $u_t^\star := V_t^\star(s) - V_{t+1}^\star(s) \ \forall t \in [H]$). Again, let us fix a state $s$ and $a \in \mathcal{A}$ be any action. Also, we use $P_{s,a}$ to denote row vector to avoid long

---

[10]For $A$ a matrix and $x$ a vector we have $\|Ax\|_{\infty} \leq \|A\|_{\infty} \|x\|_{\infty}$.

[11]Here we use $\widehat{V}_{h}^{\star}$ instead of $\widehat{V}_{t}^{\star}$ since we later have $\widehat{V}_{h,\{s,u_t^\star\}}^{\star}$. We avoid the same $t$ twice in the expression to prevent confusion.

expression. Then we have:

$$\left(\widehat{P}_{s,a} - P_{s,a}\right)\widehat{V}_h^\star = \left(\widehat{P}_{s,a} - P_{s,a}\right)\left(\widehat{V}_h^\star - \widehat{V}_{h,\{s,u_t^\star\}}^\star + \widehat{V}_{h,\{s,u_t^\star\}}^\star\right)$$

$$= \left(\widehat{P}_{s,a} - P_{s,a}\right)\left(\widehat{V}_h^\star - \widehat{V}_{h,\{s,u_t^\star\}}^\star\right) + \left(\widehat{P}_{s,a} - P_{s,a}\right)\widehat{V}_{h,\{s,u_t^\star\}}^\star$$

$$\leq \left\|\widehat{P}_{s,a} - P_{s,a}\right\|_1 \left\|\widehat{V}_h^\star - \widehat{V}_{h,\{s,u_t^\star\}}^\star\right\|_\infty + \sqrt{\frac{2\log(4/\delta)}{N}}\sqrt{\mathrm{Var}_{s,a}(\widehat{V}_{h,\{s,u_t^\star\}}^\star)} + \frac{2H\log(1/\delta)}{3N}$$

$$\leq \left\|\widehat{P}_{s,a} - P_{s,a}\right\|_1 \left\|\widehat{V}_h^\star - \widehat{V}_{h,\{s,u_t^\star\}}^\star\right\|_\infty + \sqrt{\frac{2\log(4/\delta)}{N}}\left(\sqrt{\mathrm{Var}_{s,a}(\widehat{V}_h^\star)} + \sqrt{\mathrm{Var}_{s,a}(\widehat{V}_{h,\{s,u_t^\star\}}^\star - \widehat{V}_h^\star)}\right) + \frac{2H\log(1/\delta)}{3N}$$

$$\leq \left\|\widehat{P}_{s,a} - P_{s,a}\right\|_1 \left\|\widehat{V}_h^\star - \widehat{V}_{h,\{s,u_t^\star\}}^\star\right\|_\infty + \sqrt{\frac{2\log(4/\delta)}{N}}\left(\sqrt{\mathrm{Var}_{s,a}(\widehat{V}_h^\star)} + \sqrt{\left\|\widehat{V}_{h,\{s,u_t^\star\}}^\star - \widehat{V}_h^\star\right\|_\infty^2}\right) + \frac{2H\log(1/\delta)}{3N}$$

$$= \left(\left\|\widehat{P}_{s,a} - P_{s,a}\right\|_1 + \sqrt{\frac{2\log(4/\delta)}{N}}\right)\left\|\widehat{V}_h^\star - \widehat{V}_{h,\{s,u_t^\star\}}^\star\right\|_\infty + \sqrt{\frac{2\log(4/\delta)}{N}}\sqrt{\mathrm{Var}_{s,a}(\widehat{V}_h^\star)} + \frac{2H\log(1/\delta)}{3N}$$

$$(4)$$

where the first inequality uses Bernstein inequality (Lemma J.3), the second inequality uses $\sqrt{\mathrm{Var}(\cdot)}$ is norm (norm triangle inequality). Now we treat $\left\|\widehat{P}_{s,a} - P_{s,a}\right\|_1$ and $\left\|\widehat{V}_h^\star - \widehat{V}_{h,\{s,u_t^\star\}}^\star\right\|_\infty$ separately.

**For** $\left\|\widehat{P}_{s,a} - P_{s,a}\right\|_1$. Indeed, by Lemma J.9 again $\left\|\widehat{P}_{s,a} - P_{s,a}\right\|_1 \leq \tilde{O}(\sqrt{\frac{S\log(S/\delta)}{N}})$ and by a union bound we obtain w.p., $1 - \delta$

$$\sup_{s,a}\left\|\widehat{P}_{s,a} - P_{s,a}\right\|_1 \leq C\sqrt{\frac{S\log(SA/\delta)}{N}}. \qquad (5)$$

where $C$ absorbs the higher order term and constants.

**For** $\left\|\widehat{V}_h^\star - \widehat{V}_{h,\{s,u_t^\star\}}^\star\right\|_\infty$. Note if we set $\widehat{u}_t^\star = \widehat{V}_t^\star(s) - \widehat{V}_{t+1}^\star(s)$, then by Lemma B.5

$$\widehat{V}_h^\star = \widehat{V}_{h,\{s,\hat{u}_t^\star\}}^\star$$

Next since $\widehat{V}_{h,\{s,\hat{u}_t^\star\}}^\star(\tilde{s}) = \max_a \widehat{Q}_{h,\{s,\hat{u}_t^\star\}}^\star(\tilde{s},a) \ \forall \tilde{s} \in \mathcal{S}$, by generic inequality $|\max f - \max g| \leq \max|f - g|$, we have $|\widehat{V}_{h,\{s,\hat{u}_t^\star\}}^\star(\tilde{s}) - \widehat{V}_{h,\{s,u_t^\star\}}^\star(\tilde{s})| \leq \max_a |\widehat{Q}_{h,\{s,\hat{u}_t^\star\}}^\star(\tilde{s},a) - \widehat{Q}_{h,\{s,u_t^\star\}}^\star(\tilde{s},a)|$, taking $\max_{\tilde{s}}$ on both sides, we obtain exactly

$$\left\|\widehat{V}_{h,\{s,\hat{u}_t^\star\}}^\star - \widehat{V}_{h,\{s,u_t^\star\}}^\star\right\|_\infty \leq \left\|\widehat{Q}_{h,\{s,\hat{u}_t^\star\}}^\star - \widehat{Q}_{h,\{s,u_t^\star\}}^\star\right\|_\infty$$

then by Lemma B.2,

$$\left\|\widehat{V}_h^\star - \widehat{V}_{h,\{s,u_t^\star\}}^\star\right\|_\infty \leq \left\|\widehat{Q}_{h,\{s,\hat{u}_t^\star\}}^\star - \widehat{Q}_{h,\{s,u_t^\star\}}^\star\right\|_\infty \leq H\max_t |\hat{u}_t^\star - u_t^\star|, \qquad (6)$$

Recall

$$\hat{u}_t^\star - u_t^\star = \widehat{V}_t^\star(s) - \widehat{V}_{t+1}^\star(s) - \left(V_t^\star(s) - V_{t+1}^\star(s)\right).$$

Now we denote

$$\Delta_s := \max_t |\hat{u}_t^\star - u_t^\star| = \max_t \left|\widehat{V}_t^\star(s) - \widehat{V}_{t+1}^\star(s) - \left(V_t^\star(s) - V_{t+1}^\star(s)\right)\right|,$$

then $\Delta_s$ itself is a scalar and a random variable.

To sum up, by (4), (5) and (6) and a union bound we have

**Lemma B.7.** *Fix $N > 0$. With probability $1 - \delta$, element-wisely, for all $h \in [H]$,*

$$\left|\left(\widehat{P} - P\right)\widehat{V}_h^\star\right| \leq C\sqrt{\frac{S\log(HSA/\delta)}{N}} \cdot H\max_s \Delta_s \cdot \mathbf{1} + \sqrt{\frac{2\log(4HSA/\delta)}{N}}\sqrt{\mathrm{Var}_P(\widehat{V}_h^\star)} + \frac{2H\log(HSA/\delta)}{3N} \cdot \mathbf{1}$$

Now plug Lemma B.7 back into $(\star)$ and combine Lemma B.6, we receive:

$$\left|\widehat{Q}_h^{\widehat{\pi}} - Q_h^{\widehat{\pi}}\right|$$

$$\leq \sum_{t=h}^{H} \Gamma_{h+1:t}^{\widehat{\pi}} \left( C\sqrt{\frac{S\log(HSA/\delta)}{N}} \cdot H \max_s \Delta_s \cdot \mathbf{1} + \sqrt{\frac{2\log(4HSA/\delta)}{N}}\sqrt{\mathrm{Var}_P(\widehat{V}_{t+1}^\star)} + \frac{2H\log(HSA/\delta)}{3N} \cdot \mathbf{1} \right)$$

$$+ C\epsilon_{\mathrm{opt}} \cdot \sqrt{\frac{H^2 S\log(SA/\delta)}{N}} \cdot \mathbf{1}$$

$$\leq \sum_{t=h}^{H} \Gamma_{h+1:t}^{\widehat{\pi}} \sqrt{\frac{2\log(4HSA/\delta)}{N}}\sqrt{\mathrm{Var}_P(\widehat{V}_{t+1}^\star)} + CH^2\sqrt{\frac{S\log(HSA/\delta)}{N}} \cdot \max_s \Delta_s \cdot \mathbf{1} + \frac{2H^2\log(HSA/\delta)}{3N} \cdot \mathbf{1}$$

$$+ C\epsilon_{\mathrm{opt}} \cdot \sqrt{\frac{H^2 S\log(SA/\delta)}{N}} \cdot \mathbf{1}$$

Next note

$$\sqrt{\mathrm{Var}_P(\widehat{V}_h^\star)} := \sqrt{\mathrm{Var}_P\left(\widehat{V}_h^{\widehat{\pi}\star}\right)} = \sqrt{\mathrm{Var}_P\left(\widehat{V}_h^{\widehat{\pi}\star} - \widehat{V}_h^{\widehat{\pi}} + \widehat{V}_h^{\widehat{\pi}}\right)}$$

$$\leq \sqrt{\mathrm{Var}_P\left(\widehat{V}_h^{\widehat{\pi}}\right)} + \sqrt{\mathrm{Var}_P\left(\widehat{V}_h^{\widehat{\pi}\star} - \widehat{V}_h^{\widehat{\pi}}\right)} \leq \sqrt{\mathrm{Var}_P\left(\widehat{V}_h^{\widehat{\pi}}\right)} + \left\|\widehat{V}_h^{\widehat{\pi}\star} - \widehat{V}_h^{\widehat{\pi}}\right\|_\infty \qquad (7)$$

$$\leq \sqrt{\mathrm{Var}_P\left(\widehat{V}_h^{\widehat{\pi}}\right)} + \epsilon_{\mathrm{opt}} \cdot \mathbf{1} \leq \sqrt{\mathrm{Var}_P\left(V_h^{\widehat{\pi}}\right)} + \sqrt{\mathrm{Var}_P\left(\widehat{V}_h^{\widehat{\pi}} - V_h^{\widehat{\pi}}\right)} + \epsilon_{\mathrm{opt}} \cdot \mathbf{1}$$

$$\leq \sqrt{\mathrm{Var}_P\left(V_h^{\widehat{\pi}}\right)} + \left\|\widehat{V}_h^{\widehat{\pi}} - V_h^{\widehat{\pi}}\right\|_\infty + \epsilon_{\mathrm{opt}} \cdot \mathbf{1} \leq \sqrt{\mathrm{Var}_P\left(V_h^{\widehat{\pi}}\right)} + \left\|\widehat{Q}_h^{\widehat{\pi}} - Q_h^{\widehat{\pi}}\right\|_\infty + \epsilon_{\mathrm{opt}} \cdot \mathbf{1}$$

Plug (7) back to above we obtain $\forall h \in [H]$,

$$\left|\widehat{Q}_h^{\widehat{\pi}} - Q_h^{\widehat{\pi}}\right| \leq \sqrt{\frac{2\log(4HSA/\delta)}{N}} \sum_{t=h}^{H} \Gamma_{h+1:t}^{\widehat{\pi}} \left( \sqrt{\mathrm{Var}_P\left(V_{t+1}^{\widehat{\pi}}\right)} + \left\|\widehat{Q}_{t+1}^{\widehat{\pi}} - Q_{t+1}^{\widehat{\pi}}\right\|_\infty + \epsilon_{\mathrm{opt}} \cdot \mathbf{1} \right)$$

$$+ CH^2\sqrt{\frac{S\log(HSA/\delta)}{N}} \cdot \max_s \Delta_s \cdot \mathbf{1} + \frac{2H^2\log(HSA/\delta)}{3N} \cdot \mathbf{1} + C\epsilon_{\mathrm{opt}} \cdot \sqrt{\frac{H^2 S\log(SA/\delta)}{N}} \cdot \mathbf{1}$$

$$\leq \sqrt{\frac{2\log(4HSA/\delta)}{N}} \sum_{t=h}^{H} \Gamma_{h+1:t}^{\widehat{\pi}} \sqrt{\mathrm{Var}_P\left(V_{t+1}^{\widehat{\pi}}\right)} + \sqrt{\frac{2\log(4HSA/\delta)}{N}} \sum_{t=h}^{H} \left\|\widehat{Q}_{t+1}^{\widehat{\pi}} - Q_{t+1}^{\widehat{\pi}}\right\|_\infty$$

$$+ CH^2\sqrt{\frac{S\log(HSA/\delta)}{N}} \cdot \max_s \Delta_s \cdot \mathbf{1} + \frac{2H^2\log(HSA/\delta)}{3N} \cdot \mathbf{1} + C_1\epsilon_{\mathrm{opt}} \cdot \sqrt{\frac{H^2 S\log(SA/\delta)}{N}} \cdot \mathbf{1}$$

$$\qquad (8)$$

Apply Lemma J.5 and the coarse uniform bound (Lemma J.10) we obtain the following lemma:

**Lemma B.8.** *Given $N > 0$ and $\epsilon_{opt} \leq \sqrt{H/S}$. With probability $1 - \delta$, for all $h \in [H]$,*

$$\left\|\widehat{Q}_h^{\widehat{\pi}} - Q_h^{\widehat{\pi}}\right\|_\infty \leq \sqrt{\frac{C_0 H^3 \log(4HSA/\delta)}{N}} + \sqrt{\frac{2\log(4HSA/\delta)}{N}} \sum_{t=h}^{H} \left\|\widehat{Q}_{t+1}^{\widehat{\pi}} - Q_{t+1}^{\widehat{\pi}}\right\|_\infty + C'H^4 \frac{S\log(HSA/\delta)}{N}$$

*Proof.* Since

$$\Delta_s := \max_t |\widehat{u}_t^\star - u_t^\star| = \max_t \left|\widehat{V}_t^\star(s) - \widehat{V}_{t+1}^\star(s) - \left(V_t^\star(s) - V_{t+1}^\star(s)\right)\right|$$

$$\leq 2 \cdot \max_t \left|\widehat{V}_t^\star(s) - V_t^\star(s)\right|$$

$$= 2 \cdot \max_t \left|\max_\pi \widehat{V}_t^\pi(s) - \max_\pi V_t^\pi(s)\right| \qquad (9)$$

$$\leq 2 \cdot \max_{\pi \in \Pi_g, t \in [H]} \left\|\widehat{V}_t^\pi - V_t^\pi\right\|_\infty \leq C \cdot H^2 \sqrt{\frac{S\log(HSA/\delta)}{N}}$$

where the last inequality uses Lemma J.10. Then apply union bound w.p. $1 - \delta/2$, we obtain $\max_s \Delta_s \leq C \cdot H^2 \sqrt{\frac{S \log(HSA/\delta)}{N}}$. Note (8) holds with probability $1 - \delta/2$, therefore plug above into (8) we obtain w.p. $1 - \delta$,

$$\left| \widehat{Q}_h^{\widehat{\pi}} - Q_h^{\widehat{\pi}} \right| \leq \sqrt{\frac{2 \log(4HSA/\delta)}{N}} \sum_{t=h}^{H} \Gamma_{h+1:t}^{\widehat{\pi}} \sqrt{\mathrm{Var}_P \left( V_{t+1}^{\widehat{\pi}} \right)} + \sqrt{\frac{2 \log(4HSA/\delta)}{N}} \sum_{t=h}^{H} \left\| \widehat{Q}_{t+1}^{\widehat{\pi}} - Q_{t+1}^{\widehat{\pi}} \right\|_\infty$$

$$+ C' H^4 \frac{S \log(HSA/\delta)}{N} \cdot \mathbf{1} + C_1 \epsilon_{\mathrm{opt}} \cdot \sqrt{\frac{H^2 S \log(SA/\delta)}{N}} \cdot \mathbf{1}$$

$$\leq \left[ \sqrt{\frac{C_0 H^3 \log(4HSA/\delta)}{N}} + \sqrt{\frac{2 \log(4HSA/\delta)}{N}} \sum_{t=h}^{H} \left\| \widehat{Q}_{t+1}^{\widehat{\pi}} - Q_{t+1}^{\widehat{\pi}} \right\|_\infty + C' H^4 \frac{S \log(HSA/\delta)}{N} \right] \cdot \mathbf{1},$$

where the last inequality uses Lemma J.5 and $\epsilon_{\mathrm{opt}} \leq \sqrt{H/S}$ and renames $C' = C' + C_1$. Take $\|\cdot\|_\infty$ then obtain the result. ∎

**Lemma B.9.** *Given $N > 0$. Define $C'' := 2 \cdot \max(\sqrt{C_0}, C')$ where $C'$ is the universal constant in Lemma B.8. When $N \geq 8H^2 \log(4HSA/\delta)$, then with probability $1 - \delta$, $\forall h \in [H]$,*

$$\left\| \widehat{Q}_h^{\widehat{\pi}} - Q_h^{\widehat{\pi}} \right\|_\infty \leq C'' \sqrt{\frac{H^3 \log(4HSA/\delta)}{N}} + C'' \frac{H^4 S \log(HSA/\delta)}{N}.$$
$$\left\| \widehat{Q}_h^{\pi^\star} - Q_h^{\pi^\star} \right\|_\infty \leq C'' \sqrt{\frac{H^3 \log(4HSA/\delta)}{N}} + C'' \frac{H^4 S \log(HSA/\delta)}{N}. \tag{10}$$

*Proof.* We prove by backward induction. For $h = H$, by Lemma B.8

$$\left\| \widehat{Q}_H^{\widehat{\pi}} - Q_H^{\widehat{\pi}} \right\|_\infty \leq \sqrt{\frac{C_0 H^3 \log(4HSA/\delta)}{N}} + \sqrt{\frac{2 \log(4HSA/\delta)}{N}} \left\| \widehat{Q}_{H+1}^{\widehat{\pi}} - Q_{H+1}^{\widehat{\pi}} \right\|_\infty + C' H^4 \frac{S \log(HSA/\delta)}{N}$$

$$= \sqrt{\frac{C_0 H^3 \log(4HSA/\delta)}{N}} + 0 + C' H^4 \frac{S \log(HSA/\delta)}{N}$$

$$\leq C'' \sqrt{\frac{H^3 \log(4HSA/\delta)}{N}} + C'' H^4 \frac{S \log(HSA/\delta)}{N},$$

for general $h$, by condition we have $H \sqrt{\frac{2 \log(4HSA/\delta)}{N}} \leq 1/2$, therefore by Lemma B.8

$$\left\| \widehat{Q}_h^{\widehat{\pi}} - Q_h^{\widehat{\pi}} \right\|_\infty \leq \sqrt{\frac{C_0 H^3 \log(4HSA/\delta)}{N}} + \sqrt{\frac{2 \log(4HSA/\delta)}{N}} \sum_{t=h}^{H} \left\| \widehat{Q}_{t+1}^{\widehat{\pi}} - Q_{t+1}^{\widehat{\pi}} \right\|_\infty + C' H^4 \frac{S \log(HSA/\delta)}{N}$$

$$\leq \sqrt{\frac{C_0 H^3 \log(4HSA/\delta)}{N}} + H \sqrt{\frac{2 \log(4HSA/\delta)}{N}} \max_{t+1} \left\| \widehat{Q}_{t+1}^{\widehat{\pi}} - Q_{t+1}^{\widehat{\pi}} \right\|_\infty + C' H^4 \frac{S \log(HSA/\delta)}{N}$$

$$\leq \sqrt{\frac{C_0 H^3 \log(4HSA/\delta)}{N}} + C' H^4 \frac{S \log(HSA/\delta)}{N}$$

$$+ \frac{1}{2} \left( C'' \sqrt{\frac{H^3 \log(4HSA/\delta)}{N}} + C'' \frac{H^4 S \log(HSA/\delta)}{N} \right)$$

$$\leq C'' \sqrt{\frac{H^3 \log(4HSA/\delta)}{N}} + C'' \frac{H^4 S \log(HSA/\delta)}{N}$$

The proof of the second claim is even easier since $\pi^\star$ is no longer a random policy and it is really just a non-uniform point-wise OPE. There are multiple ways to prove it and we leave it as an exercise to avoid redundancy: 1. Follow the same proving pipeline as $\left\| \widehat{Q}_h^{\widehat{\pi}} - Q_h^{\widehat{\pi}} \right\|_\infty$ used; 2. Mimic the procedure of point-wise OPE result in Lemma 3.4. in Yin et al. [2021a]. ∎

**Remark B.10.** *Note the higher order term has dependence $H^4 S$, which is somewhat unsatisfactory. We use the* recursion-back *trick to further reduce it to $H^{3.5} S^{0.5}$.*

**Lemma B.11.** *Given $N > 0$. There exists universal constants $C_1, C_2$ such that when $N \geq C_1 H^2 \log(HSA/\delta)$, then with probability $1 - \delta$, $\forall h \in [H]$,*

$$\left\| \widehat{Q}_h^{\widehat{\pi}} - Q_h^{\widehat{\pi}} \right\|_\infty \leq C_2 \sqrt{\frac{H^3 \log(HSA/\delta)}{N}} + C_2 \frac{H^3 \sqrt{HS} \log(HSA/\delta)}{N}. \tag{11}$$

*and*

$$\left\| \widehat{Q}_h^{\pi^\star} - Q_h^{\pi^\star} \right\|_\infty \leq C_2 \sqrt{\frac{H^3 \log(HSA/\delta)}{N}} + C_2 \frac{H^3 \sqrt{HS} \log(HSA/\delta)}{N}.$$

*Proof.* Note

$$\begin{aligned}
\widehat{V}_t^\star(s) - V_t^\star(s) &:= \widehat{V}_t^{\widehat{\pi}^\star}(s) - V_t^{\pi^\star}(s) \\
&= \widehat{V}_t^{\widehat{\pi}^\star}(s) - V_t^{\widehat{\pi}^\star}(s) + V_t^{\widehat{\pi}^\star}(s) - V_t^{\pi^\star}(s) \\
&\leq \widehat{V}_t^{\widehat{\pi}^\star}(s) - V_t^{\widehat{\pi}^\star}(s) \leq \left| \widehat{V}_t^{\widehat{\pi}^\star}(s) - V_t^{\widehat{\pi}^\star}(s) \right|
\end{aligned} \tag{12}$$

and similarly $V_t^\star(s) - \widehat{V}_t^\star(s) \leq \left| \widehat{V}_t^{\pi^\star}(s) - V_t^{\pi^\star}(s) \right|$, therefore by Lemma B.9 (and use $\|\widehat{V}_t^\pi - V_t^\pi\|_\infty \leq \|\widehat{Q}_t^\pi - Q_t^\pi\|_\infty$), with probability $1 - \delta$,

$$\Delta_s \leq 2 \cdot \sup_t \left\| V_t^\star - \widehat{V}_t^\star \right\| \leq 2 \max_{\widehat{\pi}^\star, \pi^\star} \sup_t \left\| \widehat{V}_t^\pi - V_t^\pi \right\|_\infty \leq C_2 \sqrt{\frac{H^3 \log(HSA/\delta)}{N}} + C_2 \frac{H^4 S \log(HSA/\delta)}{N},$$

where the second inequality uses (12). This replaces the crude bound of $O(\sqrt{H^4 S \log(HSA/\delta)/N})$ for $\max_s \Delta_s$ (recall (9)) by $O(\sqrt{H^3 \log(HSA/\delta)/N})$.

Plug this back to (8) and repeat the similar analysis we end up with (11). The second result is similarly proved.

∎

## B.7  Proof of Theorem 4.1

*Proof of Theorem 4.1.* Note $n_{s,a} = \sum_{i=1}^n \sum_{t=1}^H \mathbf{1}[s_t^{(i)} = s, a_t^{(i)} = a]$, which implies

$$\mathbb{E}[n_{s,a}] = \mathbb{E}\left[ \sum_{i=1}^n \sum_{t=1}^H \mathbf{1}[s_t^{(i)} = s, a_t^{(i)} = a] \right] = n \cdot \sum_{t=1}^H d_t^\mu(s, a).$$

Or equivalently, $n_{s,a}$ follows Binomial$(n, \sum_{t=1}^H d_t^\mu(s, a))$. Then apply the first result of Lemma J.1 by taking $\theta = 1/2$, we have when $n > 1/d_m \cdot \log(HSA/\delta)$[12], then with probability $1 - \delta$,

$$n_{s,a} \geq \frac{1}{2} n \cdot \sum_{t=1}^H d_t^\mu(s, a), \quad \forall s \in \mathcal{S}, a \in \mathcal{A}.$$

This further implies w.p. $1 - \delta$, $n_{s,a} \geq \frac{1}{2} n \cdot \sum_{t=1}^H d_t^\mu(s, a) = \frac{1}{2} n \cdot H \cdot d^\mu(s, a) \geq \frac{1}{2} n H \cdot d_m$ and further ensures

$$N := \min_{s,a} n_{s,a} \geq \frac{1}{2} n H \cdot d_m.$$

Finally, apply above to Lemma B.11, we can get over with the condition on $N$ and obtain the stated result. ∎

---

[12]The exact sufficient condition for applying Lemma J.1 is $n > 1/\sum_{t=1}^H d_t(s, a) \cdot \log(HSA/\delta)$ for all $s, a$. However, since $\sum_{t=1}^H d_t(s, a) \geq H d_m \geq d_m$, our condition $n > 1/d_m \cdot \log(HSA/\delta)$ used here is a much stronger version thus Lemma J.1 apply.

## C  Proof of minimax lower bound for model-based global uniform OPE

*Proof of Theorem 3.1.* In particular, we first focus on the case where $H = 2$ and extend the result of $H = 2$ to the general $H \geq 3$ at the end.

First of all, by Definition 2.1 let $\widehat{P}$ be the learned transition by certain model-based method. Since we assume $r_h$ is known and by convention $Q_{H+1}^\pi = 0$ for any $\pi$, then by Bellman equation

$$\widehat{Q}_h^\pi = r_h + \widehat{P}^{\pi_{h+1}}\widehat{Q}_{h+1}^\pi, \ \forall h \in [H].$$

In particular, $\widehat{Q}_{H+1}^\pi = Q_{H+1}^\pi = 0$, and this implies

$$\widehat{Q}_H^\pi = r_H + \widehat{P}^{\pi_{H+1}}\widehat{Q}_{H+1}^\pi = r_H; \quad Q_H^\pi = r_H + P^{\pi_{H+1}}Q_{H+1}^\pi = r_H + 0 = r_H$$

Now, again by definition of Bellman equation

$$\widehat{Q}_{H-1}^\pi = r_{H-1} + \widehat{P}^{\pi_H}\widehat{Q}_H^\pi = r_{H-1} + \widehat{P}^{\pi_H}r_H$$
$$Q_{H-1}^\pi = r_{H-1} + P^{\pi_H}Q_H^\pi = r_{H-1} + P^{\pi_H}r_H$$

Therefore

$$\sup_{\pi \in \Pi_g} \left\|\widehat{Q}_{H-1}^\pi - Q_{H-1}^\pi\right\|_\infty = \sup_{\pi \in \Pi_g} \left\|\left(\widehat{P}^{\pi_H} - P^{\pi_H}\right)r_H\right\|_\infty$$

$$= \sup_{\pi \in \Pi_g} \left\|\left(\widehat{P} - P\right)r_H^{\pi_H}\right\|_\infty = \sup_{\pi \in \Pi_g} \sup_{s,a} \left|\left(\widehat{P}(\cdot|s,a) - P(\cdot|s,a)\right)r_H^{\pi_H}\right|$$

$$= \sup_{s,a} \sup_{\pi \in \Pi_g} \left|\left(\widehat{P}(\cdot|s,a) - P(\cdot|s,a)\right)r_H^{\pi_H}\right|,$$

where $P^{\pi_H} \in \mathbb{R}^{S \cdot A \times S \cdot A}, r_H \in \mathbb{R}^{S \cdot A}, P \in \mathbb{R}^{S \cdot A \times S}$ and $r_H^{\pi_H} \in \mathbb{R}^S$. Note $A \geq 2$, so we can choose an instance of $r_H$ as (there are at least two actions since $A \geq 2$)

$$(r_H(s,a_1), r_H(s,a_2),...) := (1, 0, ...) \qquad \forall s \in \mathcal{S}.$$

Above implies: if $\pi_H(s) = a_1$, then $r_H^{\pi_H}(s) = 1$; if $\pi_H(s) = a_2$, then $r_H^{\pi_H}(s) = 0$; ...

Hence, if $\Pi_g$ is the global deterministic policy class, then $r_H^{\pi_H}$ can traverse all the $S$-dimensional vectors with either $0$ or $1$ in each coordinate, which is exactly

$$\left\{r_H^{\pi_H} \in \mathbb{R}^S : \pi_H \in \Pi_g\right\} \supset \{0, 1\}^S.$$

Now let us first consider fixed $s, a$. Then with this choice of $r$, above implies

$$\sup_{\pi \in \Pi_g} \left|\left(\widehat{P}(\cdot|s,a) - P(\cdot|s,a)\right)r_H^{\pi_H}\right| \geq \sup_{r \in \{0,1\}^S} \left|\left(\widehat{P}(\cdot|s,a) - P(\cdot|s,a)\right) \cdot r\right|$$

$$= \sup_{r \in \{0,1\}^S} \left|\sum_{i:r_i=1} \left(\widehat{P}(s_i|s,a) - P(s_i|s,a)\right)\right|$$

Let $I_+ := \{i \in [S] : s.t. \ \widehat{P}(s_i|s,a) - P(s_i|s,a) > 0\}$ be the set of indices where $\widehat{P}(s_i|s,a) - P(s_i|s,a)$ are positive and $I_- := \{i \in [S] : s.t. \ \widehat{P}(s_i|s,a) - P(s_i|s,a) < 0\}$ be the set of indices where $\widehat{P}(s_i|s,a) - P(s_i|s,a)$ are negative, then we further have

$$\sup_{r \in \{0,1\}^S} \left|\sum_{i:r_i=1} \left(\widehat{P}(s_i|s,a) - P(s_i|s,a)\right)\right|$$

$$\geq \max\left\{\left|\sum_{i \in I+} [\widehat{P}(s_i|s,a) - P(s_i|s,a)]\right|, \left|\sum_{i \in I-} [\widehat{P}(s_i|s,a) - P(s_i|s,a)]\right|\right\}$$

$$= \max\left\{\sum_{i \in I+} \left|\widehat{P}(s_i|s,a) - P(s_i|s,a)\right|, \sum_{i \in I-} \left|\widehat{P}(s_i|s,a) - P(s_i|s,a)\right|\right\}$$

On the other hand, we have

$$\sum_{i \in I+} \left| \widehat{P}(s_i|s,a) - P(s_i|s,a) \right| + \sum_{i \in I-} \left| \widehat{P}(s_i|s,a) - P(s_i|s,a) \right| = \left\| \widehat{P}(\cdot|s,a) - P(\cdot|s,a) \right\|_1$$

since $\widehat{P}(s_i|s,a) - P(s_i|s,a) = 0$ contributes nothing to the $l_1$ norm. Combine all the steps together, we obtain

$$\sup_{\pi \in \Pi_g} \left\| \widehat{Q}_{H-1}^\pi - Q_{H-1}^\pi \right\|_\infty \geq \sup_{s,a} \frac{1}{2} \left\| \widehat{P}(\cdot|s,a) - P(\cdot|s,a) \right\|_1 \underset{\underset{①}{\uparrow}}{\geq} c \cdot \sup_{s,a} \sqrt{\frac{S}{n_{s,a}}} \underset{\underset{②}{\uparrow}}{\geq} c'\sqrt{\frac{S}{nd_m}} \quad (13)$$

holds with constant probability $p$. Here $n_{s,a} = \sum_{h=1}^{H} \sum_{i=1}^{n} \mathbf{1}[s_h^{(i)} = s, a_h^{(i)} = a]$ is the number of data pieces visited $(s,a)$ in $n$ episodes. Now we explain how to obtain ① and ②. In particular, we first explain ②.

**Explain ②.** Recall we consider the case $H = 2$. Then

$$\mathbb{E}[n_{s,a}] = \mathbb{E}\left[ \sum_{h=1}^{H} \sum_{i=1}^{n} \mathbf{1}[s_h^{(i)} = s, a_h^{(i)} = a] \right] = n \sum_{i=1}^{2} \mathbb{E}\left[ \mathbf{1}[s_h^{(1)} = s, a_h^{(1)} = a] \right] = n \sum_{h=1}^{2} d_h^\mu(s,a)$$

*i.e.* $n_{s,a}$ is a Binomial random variable with parameter $n$ and $\sum_{h=1}^{2} d_h^\mu(s,a)$. Then by Lemma J.1, choose $\theta = \frac{1}{2}$, apply the second result, we obtain when $n > (1/2d_m) \cdot \log(SA/\delta)$[13], with probability $1 - \delta$

$$n_{s,a} \leq \frac{3}{2} n \cdot \sum_{h=1}^{2} d_h^\mu(s,a), \quad \forall s, a$$

Next, similar to the lower bound proof (Theorem G.2.) of Yin et al. [2021a], we can choose $\mu$ and $M$ (**near uniform** but not exact uniform) such that $d_h^\mu(s,a) \leq C \cdot d_m$, which further implies $n_{s,a} \leq C \cdot n \cdot d_m$, $\forall s, a$. Summarize above we end up with the following Lemma:

**Lemma C.1.** *Suppose $n \geq (1/2d_m) \cdot \log(SA/\delta)$, then*

$$\sup_{\mu, M} \mathbb{P}\left[ \sqrt{\frac{1}{n_{s,a}}} \geq C \cdot \sqrt{\frac{1}{n \cdot d_m}}, \ \forall s, a \right] \geq 1 - \delta$$

**Explain ①.** To make the explanation rigorous, we first fix a pair $(s,a)$ and conditional on $n_{s,a}$. Then by a direct translation of Lemma J.7, we have

$$\inf_{\widehat{P}} \sup_{P(\cdot|s,a) \in \mathcal{M}_S} \mathbb{P}\left[ \|\widehat{P}(\cdot|s,a) - P(\cdot|s,a)\|_1 \geq \frac{1}{8}\sqrt{\frac{eS}{2n_{s,a}}} - o(\cdot) \middle| n_{s,a} \geq \frac{e}{32}S \right] \geq p,$$

where $o(\cdot)$ is some exponentially small term in $S, n$. Now we consider everything under the condition $n \geq \frac{e}{32} \cdot S/d_m \log(SA/\delta)$. Next again take $\theta = 1/2$, then by the first result of Lemma J.1, with probability $1 - \delta$,

$$n_{s,a} \geq \frac{1}{2} n \cdot \sum_{h=1}^{2} d_h^\mu(s,a) \geq n \cdot d_m \geq \frac{e}{32} S \log(SA/\delta).$$

---

[13]By Lemma J.1,the inequality holds as long as $n \geq 1/\sum_{h=1}^{2} d_h^\mu(s,a) \log(SA/\delta)$, here $n > (1/2d_m) \cdot \log(SA/\delta)$ is a stronger sufficient condition.

where the last inequality uses the condition $n \geq \frac{e}{32} \cdot S/d_m \log(SA/\delta)$. Therefore this implies

$$\inf_{\widehat{P}} \sup_{P(\cdot|s,a)\in\mathcal{M}_S} \mathbb{P}\left[\|\widehat{P}(\cdot|s,a) - P(\cdot|s,a)\|_1 \geq \frac{1}{8}\sqrt{\frac{eS}{2n_{s,a}}} - o\left(\cdot\right)\right]$$

$$= \inf_{\widehat{P}} \sup_{P(\cdot|s,a)\in\mathcal{M}_S} \left(\mathbb{P}\left[\|\widehat{P}(\cdot|s,a) - P(\cdot|s,a)\|_1 \geq \frac{1}{8}\sqrt{\frac{eS}{2n_{s,a}}} - o\left(\cdot\right)\Big| n_{s,a} \geq \frac{e}{32}S\right] \cdot \mathbb{P}\left[n_{s,a} \geq \frac{e}{32}S\right]\right.$$

$$\left.+ \mathbb{P}\left[\|\widehat{P}(\cdot|s,a) - P(\cdot|s,a)\|_1 \geq \frac{1}{8}\sqrt{\frac{eS}{2n_{s,a}}} - o\left(\cdot\right)\Big| n_{s,a} \leq \frac{e}{32}S\right] \cdot \mathbb{P}\left[n_{s,a} \leq \frac{e}{32}S\right]\right)$$

$$\geq \inf_{\widehat{P}} \sup_{P(\cdot|s,a)\in\mathcal{M}_S} \mathbb{P}\left[\|\widehat{P}(\cdot|s,a) - P(\cdot|s,a)\|_1 \geq \frac{1}{8}\sqrt{\frac{eS}{2n_{s,a}}} - o\left(\cdot\right)\Big| n_{s,a} \geq \frac{e}{32}S\right] \cdot \mathbb{P}\left[n_{s,a} \geq \frac{e}{32}S\right]$$

$$\geq p \cdot (1 - \delta),$$

To sum up, we have the following lemma:

**Lemma C.2.** *Let $n \geq \frac{e}{32}S/d_m \cdot \log(SA/\delta)$, then there exists a $0 < p < 1$,*

$$\inf_{\widehat{P}} \sup_{P(\cdot|s,a)\in\mathcal{M}_S} \mathbb{P}\left[\|\widehat{P}(\cdot|s,a) - P(\cdot|s,a)\|_1 \geq \frac{1}{8}\sqrt{\frac{eS}{2n_{s,a}}} - o\left(\cdot\right)\right] \geq p \cdot (1 - \delta).$$

Now we finish the proof for the case where $H = 2$. First note by (13),

$$\sup_{\pi\in\Pi_g} \left\|\widehat{Q}^\pi_{H-1} - Q^\pi_{H-1}\right\|_\infty \geq \sup_{s,a} \frac{1}{2}\left\|\widehat{P}(\cdot|s,a) - P(\cdot|s,a)\right\|_1$$

with probability 1, therefore by (13), Lemma C.1, Lemma C.2 we have

$$\inf_{\widehat{P}} \sup_{P\in\mathcal{M}_S} \mathbb{P}\left[\sup_{\pi\in\Pi_g} \left\|\widehat{Q}^\pi_{H-1} - Q^\pi_{H-1}\right\|_\infty \geq C \cdot \sqrt{\frac{S}{nd_m}}\right] \geq p(1 - \delta) - \delta$$

when $n \geq c \cdot S/d_m \log(SA/\delta)$ for some $c \geq \frac{e}{32}$. Above holds for any $\delta$.

It is easy to check $\frac{3}{2}\frac{p}{1+p} \leq 1$, therefore, in particular we set $\delta = \frac{3}{2}\frac{p}{1+p}$, direct calculation shows

$$p(1 - \delta) - \delta = \frac{p}{2},$$

which completes the proof for $H = 2$.

**Extend to the general $H \geq 3$.**

**Step 1.** Similar to the decomposition in section B.4, we also have:

$$\widehat{Q}^\pi_t - Q^\pi_t = \sum_{h=t}^{H} \widehat{\Gamma}^\pi_{t+1:h}(\widehat{P} - P)V^\pi_{h+1}$$

**Step 2.** Now choosing rewards recursively from back (with $\|r_H\|_\infty = c$ sufficiently small) such that $1 \geq r_h \geq (\|r_{h+1}\|_\infty + \ldots + \|r_H\|_\infty)$ element-wisely $\forall h$, and $\max_{s,a} r_h(s,a) = 3 \min_{s,a} r_h(s,a)$. We denote $r_{h,max} := \max_{s,a} r_h(s,a)$ and $r_{h,min} := \min_{s,a} r_h(s,a)$. This choice guarantees:

$$r_{h,min} := \min_{s,a} r_h(s,a) > \|P^{\pi_{h+1}}r_{h+1} + .. + P^{\pi_{h+1:H}}r_H\|_\infty$$

since $P^{\pi_h}$ is row-stochastic.

**Step 3.** Next note $V^\pi_h = r_h + P^{\pi_{h+1}}r_{h+1} + .. + P^{\pi_{h+1:H}}r_H$, so set $(r_h(s,a_1), r_h(s,a_2), \ldots) := (\max_{s,a} r_h(s,a), \min_{s,a} r_h(s,a), \ldots)$, then choose $\pi_h$ similar to the $H = 2$ case and use **Step 1** and **Step 2** we have

$$|(\widehat{P}_{s,a} - P_{s,a})V^\pi_h| \geq \frac{1}{2}\|\widehat{P}_{s,a} - P_{s,a}\|_1 \cdot (r_{h,max} - r_{h,min} - (P^{\pi_{h+1}}r_{h+1} + .. + P^{\pi_{h+1:H}}r_H))$$

$$\geq \frac{1}{2}\|\widehat{P}_{s,a} - P_{s,a}\|_1 \cdot r_{h,min} \geq \frac{1}{2}\|\widehat{P}_{s,a} - P_{s,a}\|_1 \cdot c$$

where the reasoning of the first inequality is similar to the case of $H = 2$. Next use $\hat{\Gamma}^\pi_{t+1:h}$ is row-stochastic then from **Step 1** and take the sum we have

$$||\hat{Q}^\pi_1 - Q^\pi_1||_\infty \geq \frac{1}{2} c \cdot H \min_{s,a} ||\hat{P}_{s,a} - P_{s,a}||_1.$$

for such choice of rewards and $\pi$.

**Step 4.** However, in the above construction $c$ actually depends on $H$ due to the design $1 \geq r_h \geq (||r_{h+1}||_\infty + \ldots + ||r_H||_\infty)$. To get a universal constant $c$ we could use the bound $||\hat{Q}^\pi_1 - Q^\pi_1||_\infty \gtrsim r_{\frac{H}{2},min} \cdot \frac{H}{2} \min_{s,a} ||\hat{P}_{s,a} - P_{s,a}||_1$ instead, where $r_{\frac{H}{2},min}$ in Step 2 is universally lower bounded. Then we apply $||\hat{P}_{s,a} - P_{s,a}||_1 \gtrsim \Omega(\sqrt{S/nd_m})$ to obtain the lower bound $\Omega(\sqrt{H^2 S/nd_m})$.

■

**Remark C.3.** *We point out while our lower bound of $\Omega(H^2 S/d_m \epsilon^2)$ for uniform OPE appears to be qualitatively similar to the lower bound of $\Omega(H^2 S^2 A/\epsilon^2)$ derived for the online reward-free RL setting [Jin et al., 2020a], our result is not implied by theirs and cannot be proven by directly adapting their construction. Those two results are in principle different since: the result in [Jin et al., 2020a] is learning-oriented where they define the problem class on $O(S)$ states and forcing $\Omega(SA/\epsilon^2)$ episodes in each state and end up with $O(S^2 A/\epsilon^2)$ complexity; our result is evaluation-oriented where we need reduce the uniform evaluation problem to estimating probability distribution in $\ell_1$-error. The global uniform OPE and the reward-free setting are also different tasks (one cannot imply the other): the former deals with uniform convergence over all policies but with a fixed reward while the latter aims at learning simultaneously over all rewards.*

# D  Proof for optimal offline learning (Corollary 4.2)

*Proof.* This is a corollary of Theorem 4.1. Indeed, by taking $\hat{\pi} = \hat{\pi}^\star$, we first have

$$\left\|\widehat{V}^{\hat{\pi}^\star}_1 - V^{\hat{\pi}^\star}_1\right\|_\infty \leq \left\|\widehat{Q}^{\hat{\pi}^\star}_1 - Q^{\hat{\pi}^\star}_1\right\|_\infty \leq C\left[\sqrt{\frac{H^2 \iota}{nd_m}} + \frac{H^{2.5} S^{0.5} \iota}{nd_m}\right].$$

Similar to the second result in Lemma B.11, we also have

$$\left\|\widehat{V}^{\pi^\star}_1 - V^{\pi^\star}_1\right\|_\infty \leq \left\|\widehat{Q}^{\pi^\star}_1 - Q^{\pi^\star}_1\right\|_\infty \leq C\left[\sqrt{\frac{H^2 \iota}{nd_m}} + \frac{H^{2.5} S^{0.5} \iota}{nd_m}\right].$$

Next, recall the definition of $\hat{\pi} \in \Pi_l$ that

$$\left\|\widehat{V}^{\hat{\pi}^\star}_1 - \widehat{V}^{\hat{\pi}}_1\right\|_\infty \leq \epsilon_{\text{opt}},$$

and Theorem 4.1 again that

$$\left\|\widehat{V}^{\hat{\pi}}_1 - V^{\hat{\pi}}_1\right\|_\infty \leq \left\|\widehat{Q}^{\hat{\pi}}_1 - Q^{\hat{\pi}}_1\right\|_\infty \leq C\left[\sqrt{\frac{H^2 \iota}{nd_m}} + \frac{H^{2.5} S^{0.5} \iota}{nd_m}\right].$$

Therefore

$$V_1^{\pi^\star} - V_1^{\widehat{\pi}} = V_1^{\pi^\star} - V_1^{\widehat{\pi}^\star} + V_1^{\widehat{\pi}^\star} - V_1^{\widehat{\pi}}$$

$$\leq \max_{\widehat{\pi}^\star, \pi^\star} \left\| \widehat{V}_1^\pi - V_1^\pi \right\|_\infty + V_1^{\widehat{\pi}^\star} - V_1^{\widehat{\pi}}$$

$$= \max_{\widehat{\pi}^\star, \pi^\star} \left\| \widehat{V}_1^\pi - V_1^\pi \right\|_\infty + \left( V_1^{\widehat{\pi}^\star} - \widehat{V}_1^{\widehat{\pi}^\star} \right) + \left( \widehat{V}_1^{\widehat{\pi}^\star} - \widehat{V}_1^{\widehat{\pi}} \right) + \left( \widehat{V}_1^{\widehat{\pi}} - V_1^{\widehat{\pi}} \right)$$

$$\leq 3C \left[ \sqrt{\frac{H^2 \iota}{nd_m}} + \frac{H^{2.5} S^{0.5} \iota}{nd_m} \right] + \left\| \widehat{V}_1^{\widehat{\pi}^\star} - \widehat{V}_1^{\widehat{\pi}} \right\|_\infty \cdot \mathbf{1}$$

$$\leq 3C \left[ \sqrt{\frac{H^2 \iota}{nd_m}} + \frac{H^{2.5} S^{0.5} \iota}{nd_m} \right] + \epsilon_{\text{opt}} \cdot \mathbf{1}.$$

This completes the proof.

■

# E    Proof for optimal offline Task-agnostic learning (Theorem 5.3)

*Proof.* Recall the definition of offline task-agnostic setting, where $K$ tasks corresponds to $K$ MDPs $M_k = (\mathcal{S}, \mathcal{A}, P, r_k, H, d_1)$ with different mean reward functions $r_k$'s. Since the incremental number of rewards do not incur randomness, therefore by Corollary 4.2, choose $\widehat{\pi}_k = \widehat{\pi}_k^\star$ and apply a union bound we obtain with probability $1 - \delta$,

$$\sup_{k \in [K]} \| V_{1,M_k}^\star - V_{1,M_k}^{\widehat{\pi}_k^\star} \|_\infty \leq O \left[ \sqrt{\frac{H^2 \log(HSAK/\delta)}{nd_m}} + \frac{H^{2.5} S^{0.5} \log(HSAK/\delta)}{nd_m} \right]$$

$$= O \left[ \sqrt{\frac{H^2 (\iota + \log(K))}{nd_m}} + \frac{H^{2.5} S^{0.5} (\iota + \log(K))}{nd_m} \right],$$

which completes the proof.

■

**Remark E.1.** *We stress that Section 3 of Zhang et al. [2020b] claims the definition of task-agnostic RL setting embraces one challenge that $r_k^{(i)}$'s are the observed random realizations and the need to accurately estimate mean rewards $r_k$'s causes the additional $\log(K)$ dependence. However, for offline case, this is not essential since, by straightforward calculation, estimating $r_k^{(i)}$'s accurately only requires $\tilde{O}(\log(K)/d_m \epsilon^2)$ samples, which is of lower order comparing to $\tilde{O}(H^2 \log(K)/d_m \epsilon^2)$ learning bound. Therefore, in Definition 5.1 we do not incorporate the random version statement for reward $r_k$.*

## E.1    Offline Learning in the Constrained MDPs (CMDP)

Recently, there is a line of studies in the Constrained Markov Decision Processes (CMDP) (*e.g.* ?), where the MDP $M = (\mathcal{S}, \mathcal{A}, P, H, d_1)$. When the reward is set to be $r$, it defines the objective function $V_r^\pi$ and there is another utility function $g$ that defines the constraint. To be concrete, the objective formualted as:

$$\underset{\pi \in \Delta(\mathcal{A}|\mathcal{S}, H)}{\text{maximize}} \, V_{r,1}^\pi (x_1) \text{ subject to } V_{g,1}^\pi (x_1) \geq b \tag{14}$$

where $b \in (0, H]$ is some constraint threshold. In addition, the formulation needs a Slater condition that: there exists $\gamma > 0$ and $\bar{\pi} \in \Delta(\mathcal{A}|\mathcal{S}, H)$ such that $V_{g,1}^{\bar{\pi}}(x_1) \geq b + \gamma$.

Let $\pi^\star$ be the optimal solution that is compatible with the programming (14) (note this is **different** from the optimal policy that maximizes $V_{r,1}^\pi$ only), then by feasibility it satisfies $V_{g,1}^{\pi^\star} \geq b$.

Now let $\hat{\pi}^\star$ be the solution of the empirical program:

$$\underset{\pi \in \Delta(\mathcal{A}|\mathcal{S},H)}{\text{maximize}} \widehat{V}_{r,1}^\pi (x_1) \text{ subject to } \widehat{V}_{g,1}^\pi (x_1) \geq b \tag{15}$$

then we can show $\hat{\pi}^\star$ is a near-optimal solution for (14) via the local uniform convergence guarantee (Theorem 4.1).

Indeed, define a surrogate program:

$$\underset{\pi \in \Delta(\mathcal{A}|\mathcal{S},H)}{\text{maximize}} \widehat{V}_{r,1}^\pi (x_1) \text{ subject to } V_{g,1}^\pi (x_1) \geq b \tag{16}$$

and let $\bar{\pi}^\star$ be the solution for (16). Then apparently $\bar{\pi}^\star$ satisfies $V_{g,1}^{\bar{\pi}^\star} (x_1) \geq b$. Moreover, we have

$$\begin{aligned}
V_{r,1}^{\pi^\star} - V_{r,1}^{\bar{\pi}^\star} &= V_{r,1}^{\pi^\star} - \widehat{V}_{r,1}^{\pi^\star} + \widehat{V}_{r,1}^{\pi^\star} - \widehat{V}_{r,1}^{\bar{\pi}^\star} + \widehat{V}_{r,1}^{\bar{\pi}^\star} - V_{r,1}^{\bar{\pi}^\star} \\
&\leq V_{r,1}^{\pi^\star} - \widehat{V}_{r,1}^{\pi^\star} + 0 + \widehat{V}_{r,1}^{\bar{\pi}^\star} - V_{r,1}^{\bar{\pi}^\star} \\
&\leq 2 \sup_\pi |V_{r,1}^\pi - \widehat{V}_{r,1}^\pi|
\end{aligned}$$

On the other hand, by local uniform convergence guarantee, $|V_{g,1}^\pi - \widehat{V}_{g,1}^\pi| \leq \tilde{O}(\sqrt{H^2/nd_m})$ for all $\pi$ in the $\sqrt{H/S}$-neighborhood of $\hat{\pi}^\star$ (w.r.t $g$). This implies

$$V_{r,1}^{\pi^\star} - V_{r,1}^{\hat{\pi}^\star} \leq 2 \sup_\pi |V_{r,1}^\pi - \widehat{V}_{r,1}^\pi| + \tilde{O}(\sqrt{H^2/nd_m})$$

and the violation of the constraint is bounded by $\tilde{O}(\sqrt{H^2/nd_m})$. This means any approach that solves (15) is near-optimal for the original constrained MDP task given the uniform convergence guarantee.

## F  Proof for optimal offline Reward-free learning (Theorem 5.4)

Similar to before, recall $n_{s,a} = \sum_{h=1}^H \sum_{i=1}^n \mathbf{1}[s_h^{(i)} = s, a_h^{(i)} = a]$. We first prove two lemmas which essentially provide a version of *"Maximal Bernstein inequality"*. We first fix a pair $(s,a)$ and then conditional on $n_{s,a}$.

**Lemma F.1.** *We define $\epsilon_1 = \sqrt{\frac{1}{HS^2}}$. Let $\mathcal{G} = \{[i_1\epsilon_1, i_2\epsilon_1, \ldots, i_S\epsilon_1]^\top | i_1, i_2, \ldots, i_S \in \mathbb{Z}\} \cap [0,H]^S$ be the $S$-dimensional grid. Next define $\iota_1 = \log[(\sqrt{H^3S^2})^S/\delta]$. Then with probability $1 - \delta$,*

$$\left| (P_{s,a} - \widehat{P}_{s,a})w \right| \leq \sqrt{\frac{2\mathrm{Var}_{s,a}(w)\iota_1}{n_{s,a}}} + \frac{2H\iota_1}{3n_{s,a}}, \quad \forall w \in \mathcal{G}.$$

This is by the direct application of Bernstein inequality with a union bound, where the cardinality of $\mathcal{G}$ is

$$\left(\frac{H}{\epsilon_1}\right)^S = \left(\sqrt{H^3S^2}\right)^S.$$

**Lemma F.2.** *Let the $S$-dimensional grid be $\mathcal{G} = \{[i_1\epsilon_1, i_2\epsilon_1, \ldots, i_S\epsilon_1]^\top | i_1, i_2, \ldots, i_S \in \mathbb{Z}\} \cap [0,H]^S$ and define $\iota_1 = \log[(\sqrt{H^3S^2})^S/\delta]$. It holds with probability $1 - \delta$,*

$$\left| (P_{s,a} - \widehat{P}_{s,a})v \right| \leq \sqrt{\frac{2\mathrm{Var}_{s,a}(v)\iota_1}{n_{s,a}}} + C\sqrt{\frac{\iota_1}{n_{s,a}HS}} + \frac{2H\iota_1}{3n_{s,a}}, \quad \forall\, v \in [0,H]^S.$$

*Proof.* Let $z := \mathrm{Proj}_{\mathcal{G}}(v)$. Then by design of $\mathcal{G}$ we have

$$\|z - v\|_\infty \leq \epsilon_1 = \sqrt{\frac{1}{HS^2}}.$$

Therefore we obtain $\forall v \in [0, H]^S$,

$$
\begin{aligned}
\left| (P_{s,a} - \widehat{P}_{s,a}) v \right| &\le \left| (P_{s,a} - \widehat{P}_{s,a})(v - z) \right| + \left| (P_{s,a} - \widehat{P}_{s,a}) z \right| \\
&\le \left\| P_{s,a} - \widehat{P}_{s,a} \right\|_1 \| z - v \|_\infty + \left| (P_{s,a} - \widehat{P}_{s,a}) z \right| \\
&\le c \sqrt{\frac{S}{n_{s,a}}} \| z - v \|_\infty + \sqrt{\frac{2 \mathrm{Var}_{s,a}(z) \iota_1}{n_{s,a}}} + \frac{2H \iota_1}{3 n_{s,a}} \\
&\le c \sqrt{\frac{S}{n_{s,a}}} \| z - v \|_\infty + \sqrt{\frac{2 \| z - v \|_\infty^2 \iota_1}{n_{s,a}}} + \sqrt{\frac{2 \mathrm{Var}_{s,a}(v) \iota_1}{n_{s,a}}} + \frac{2H \iota_1}{3 n_{s,a}} \\
&\le C \sqrt{\frac{S \iota_1}{n_{s,a}}} \| z - v \|_\infty + \sqrt{\frac{2 \mathrm{Var}_{s,a}(v) \iota_1}{n_{s,a}}} + \frac{2H \iota_1}{3 n_{s,a}} \\
&\le C \sqrt{\frac{\iota_1}{n_{s,a} H S}} + \sqrt{\frac{2 \mathrm{Var}_{s,a}(v) \iota_1}{n_{s,a}}} + \frac{2H \iota_1}{3 n_{s,a}}.
\end{aligned}
$$

where the third inequality uses Lemma F.1 and Lemma J.9. $\blacksquare$

Then recall $N := \min_{s,a} n_{s,a}$, by Lemma F.2 and a union bound we obtain with probability $1 - \delta$, element-wisely,

$$
\left| (P - \widehat{P}) v \right| \le C \cdot \left( \sqrt{\frac{2 \mathrm{Var}_{s,a}(v) \iota_2}{N}} + 2 \sqrt{\frac{\iota_2}{N \cdot HS}} + \frac{2H \iota_2}{3N} \right) \cdot \mathbf{1}, \quad \forall\, v \in [0, H]^S, \qquad (17)
$$

where $\iota_2 = S \log(HSA/\delta)$.

**Remark F.3.** *Equation 17 is a form of maximal Bernstein inequality as it keeps validity for all $v \in [0, H]^S$. The price for this stronger result is the extra $S$ factor (coming from $\iota_2$) in the dominate term.*

Now, for *any* reward $r$, by (empirical) Bellman equation we have element-wisely:

$$
\begin{aligned}
\widehat{Q}_h^{\widehat{\pi}^\star} - Q_h^{\widehat{\pi}^\star} &= r_h + \widehat{P}^{\widehat{\pi}_{h+1}^\star} \widehat{Q}_{h+1}^{\widehat{\pi}^\star} - r_h - P^{\widehat{\pi}_{h+1}^\star} Q_{h+1}^{\widehat{\pi}^\star} \\
&= \left( \widehat{P}^{\widehat{\pi}_{h+1}^\star} - P^{\widehat{\pi}_{h+1}^\star} \right) \widehat{Q}_{h+1}^{\widehat{\pi}^\star} + P^{\widehat{\pi}_{h+1}^\star} \left( \widehat{Q}_{h+1}^{\widehat{\pi}^\star} - Q_{h+1}^{\widehat{\pi}^\star} \right) \\
&= \left( \widehat{P} - P \right) \widehat{V}_{h+1}^{\widehat{\pi}^\star} + P^{\widehat{\pi}_{h+1}^\star} \left( \widehat{Q}_{h+1}^{\widehat{\pi}^\star} - Q_{h+1}^{\widehat{\pi}^\star} \right) \\
&= \ldots = \sum_{t=h}^{H} \Gamma_{h+1:t}^{\widehat{\pi}^\star} \left( \widehat{P} - P \right) \widehat{V}_{t+1}^{\widehat{\pi}^\star}
\end{aligned}
$$

where $\Gamma_{h+1:t}^{\pi} = \prod_{i=h+1}^{t} P^{\pi_i}$ is multi-step state-action transition and $\Gamma_{h+1:h} := I$.

**Concentration on** $\left(\widehat{P} - P\right)\widehat{V}_h^\star$**.** Now by (17), we have the following:

$$
\left(\widehat{P}_{s,a} - P_{s,a}\right)\widehat{V}_h^\star
$$

$$
\leq C \cdot \left(\sqrt{\frac{2\mathrm{Var}_{s,a}(\widehat{V}_h^\star)\iota_2}{N}} + 2\sqrt{\frac{\iota_2}{N \cdot HS}} + \frac{2H\iota_2}{3N}\right)
$$

$$
\leq C \cdot \left(\sqrt{\frac{2\mathrm{Var}_{s,a}(V_h^{\widehat{\pi}^\star})\iota_2}{N}} + 2\sqrt{\frac{\iota_2}{N \cdot HS}} + \sqrt{\frac{2\iota_2}{N}} \cdot \left\|\widehat{V}_h^{\widehat{\pi}^\star} - V_h^{\widehat{\pi}^\star}\right\|_\infty + \frac{2H\iota_2}{3N}\right) \qquad (18)
$$

$$
\leq C \cdot \left(\sqrt{\frac{2\mathrm{Var}_{s,a}(V_h^{\widehat{\pi}^\star})\iota_2}{N}} + 2\sqrt{\frac{\iota_2}{N \cdot HS}} + \sqrt{\frac{2\iota_2}{N}} \cdot H^2\sqrt{\frac{S}{N}} + \frac{2H\iota_2}{3N}\right)
$$

$$
\leq C' \cdot \left(\sqrt{\frac{2\mathrm{Var}_{s,a}(V_h^{\widehat{\pi}^\star})\iota_2}{N}} + 2\sqrt{\frac{\iota_2}{N \cdot HS}} + \frac{2H^2 S\log(HSA/\delta)}{N}\right),
$$

where the third inequality uses Lemma J.10[14]. Then above implies

$$
\widehat{Q}_h^{\widehat{\pi}^\star} - Q_h^{\widehat{\pi}^\star}
$$

$$
\leq C' \sum_{t=h}^{H} \Gamma_{h+1:t}^{\widehat{\pi}^\star} \cdot \left(\sqrt{\frac{2\mathrm{Var}_{s,a}(V_h^{\widehat{\pi}^\star})\iota_2}{N}} + 2\sqrt{\frac{\iota_2}{N \cdot HS}} + \frac{2H^2 S\log(HSA/\delta)}{N}\right)
$$

$$
\leq C' \left[\sum_{t=h}^{H} \Gamma_{h+1:t}^{\widehat{\pi}^\star} \cdot \sqrt{\frac{2\mathrm{Var}_{s,a}(V_h^{\widehat{\pi}^\star})\iota_2}{N}} + 2\sqrt{\frac{H\log(HSA/\delta)}{N}} + \frac{2H^3 S\log(HSA/\delta)}{N}\right]
$$

$$
\leq C' \left[\sqrt{\frac{2H^3 S\log(HSA/\delta)}{N}} + 2\sqrt{\frac{H\log(HSA/\delta)}{N}} + \frac{2H^3 S\log(HSA/\delta)}{N}\right]
$$

$$
\leq C'' \left[\sqrt{\frac{H^3 S\log(HSA/\delta)}{N}} + \frac{H^3 S\log(HSA/\delta)}{N}\right]
$$

$$
\leq O \left[\sqrt{\frac{H^2 S\log(HSA/\delta)}{nd_m}} + \frac{H^2 S\log(HSA/\delta)}{nd_m}\right],
$$

where the third inequality uses Lemma J.5 and the last one uses $N \geq \frac{1}{2}nd_m$ with high probability. Similar result holds for $\widehat{Q}_h^{\pi^\star} - Q_h^{\pi^\star}$. Combing those results we have reward-free bound (for any reward simultaneously)

$$
O \left[\sqrt{\frac{H^2 S\log(HSA/\delta)}{nd_m}} + \frac{H^2 S\log(HSA/\delta)}{nd_m}\right],
$$

which finishes the proof of Theorem 5.4.

**Remark F.4.** *Note above result is tight in both the dominate term AND the higher order term. Therefore this result cannot be further improved even in the higher order term.*

---

[14]Note the use of Lemma J.10 also works for any rewards since the only high probability result they used is for $||P - \hat{P}||_1$. Therefore conditional on the concentration for $||P - \hat{P}||_1$, the argument follows for any arbitrary reward as well.

# G Discussion of Section 5

In this section we explain why Theorem 5.3 and Theorem 5.4 are optimal in the offline RL.

We begin with the offline task-agnostic setting. For the exquisite readers who check the proof of Theorem 5 of Zhang et al. [2020b], the proving procedure of their lower bound follows the standard reduction to best-arm identification in multi-armed bandit problems. More specifically, to incorporate the dependence of $\log(K)$, they rely on the Theorem 10 of Zhang et al. [2020b] (which is originated from Mannor and Tsitsiklis [2004]) to show in order to be $(\epsilon, \delta)$-correct for a problem with $A$ arms and with $K$ tasks, it need at least $\Omega(\frac{A}{\epsilon^2} \log(\frac{K}{\delta}))$ samples. Such a result updates the Lemma G.1. in Yin et al. [2021b] by the extra factor $\log(K)$ for the bandit problem with $K$ tasks. With no modification, the rest of the proof in Section E of Yin et al. [2021b] follows though and one can end up with the lower bound $\Omega(H^2 \log(K)/d_m \epsilon^2)$ over the problem class $\mathcal{M}_{d_m} := \{(\mu, M) \mid \min_{t,s_t,a_t} d_t^\mu(s_t, a_t) \geq d_m\}$. The case for the offline reward-free setting is also similar. Indeed, the $\Omega(SA/\epsilon^2)$ trajectories in Lemma 4.2 in Jin et al. [2020a] could be replaced by $\Omega(1/d_m \epsilon^2)$ by choosing some hard *near-uniform* behavior policy instances (see Section E.2 in Yin et al. [2021b]) and the rest follows since by forcing $S$ such instances (Section 4.2 of Jin et al. [2020a]) to obtain $\Omega(S/d_m \epsilon^2)$ and create a chain of $\Omega(H)$ rewards for $\Omega(H^2 S/d_m \epsilon^2)$.

# H Proof of the linear MDP with anchor representations (Section 6)

Recall that we assume a generative oracle here. Sometimes we abuse the notation $\mathcal{K}$ for either anchor point set or the anchor point indices set. The meaning should be clear in each context.

## H.1 Model-based Plug-in Estimator for Anchor Representations

**Step 1:** For each $(s_k, a_k)$ where index $k \in \mathcal{K}$, collect $N$ samples from $P(\cdot|s_k, a_k)$; compute

$$\widehat{P}_\mathcal{K}(s'|s_k, a_k) = \frac{count(s, a, s')}{N};$$

**Step 2:** Compute the linear combination coefficients $\lambda_k^{s,a}$ satisfies $\phi(s,a) = \sum_{k \in \mathcal{K}} \lambda_k^{s,a} \phi(s_k, a_k)$;

**Step 3:** Estimate transition distribution

$$\widehat{P}(s'|s, a) = \sum_{k \in \mathcal{K}} \lambda_k^{s,a} \cdot \widehat{P}_\mathcal{K}(s'|s_k, a_k).$$

We need to check such $\widehat{P}(s'|s, a)$ is a valid distribution. This is due to:

$$\sum_{k \in \mathcal{K}} \lambda_k^{s,a} = \sum_{k \in \mathcal{K}} \sum_{s'} \lambda_k^{s,a} P(s'|s_k, a_k) = \sum_{s'} \sum_{k \in \mathcal{K}} \lambda_k^{s,a} P(s'|s_k, a_k)$$
$$= \sum_{s'} \sum_{k \in \mathcal{K}} \lambda_k^{s,a} \langle \phi(s_k, a_k), \psi(s') \rangle = \sum_{s'} \langle \phi(s, a), \psi(s') \rangle = \sum_{s'} P(s'|s, a) = 1$$

and

$$\sum_{s'} \widehat{P}(s'|s, a) = \sum_{s'} \sum_{k \in \mathcal{K}} \lambda_k^{s,a} \widehat{P}_\mathcal{K}(s' \mid s_k, a_k) = \sum_{k \in \mathcal{K}} \sum_{s'} \lambda_k^{s,a} \widehat{P}_\mathcal{K}(s' \mid s_k, a_k)$$
$$= \sum_{k \in \mathcal{K}} \lambda_k^{s,a} \frac{N}{N} = 1.$$

**Step 4:** construct empirical model $\widehat{M} = (\mathcal{S}, \mathcal{A}, \widehat{P}, r, H)$ and output $\widehat{\pi}^\star = \operatorname{argmax}_\pi \widehat{V}_1^\pi$.

Similarly, Bellman (optimality) equations hold[15]

$$V_t^\star(s) = \max_a \left\{ r(s,a) + \int_{s'} V_{t+1}^\star(s') dP(s'|s,a) \right\}, \quad \forall s \in \mathcal{S}.$$

$$\widehat{V}_t^\star(s) = \max_a \left\{ r(s,a) + \int_{s'} \widehat{V}_{t+1}^\star(s') d\widehat{P}(s'|s,a) \right\}, \quad \forall s \in \mathcal{S}.$$

## H.2 General absorbing MDP

The definition of the general absorbing MDP remains the same: *i.e.* for a fixed state $s$ and a sequence $\{u_t\}_{t=1}^H$, MDP $M_{s,\{u_t\}_{t=1}^H}$ is identical to $M$ for all states except $s$, and state $s$ is absorbing in the sense $P_{M_{s,\{u_t\}_{t=1}^H}}(s|s,a) = 1$ for all $a$, and the instantaneous reward at time $t$ is $r_t(s,a) = u_t$ for all $a \in \mathcal{A}$. Also, we use the shorthand notation $V_{\{s,u_t\}}^\pi$ for $V_{s,M_{s,\{u_t\}_{t=1}^H}}^\pi$ and similarly for $Q_{\{s,u_t\}}$ and transition $P_{\{s,u_t\}}$. Then the following properties mirroring the Lemma B.1 and Lemma B.2 with nearly identical proof but for the integral version (which we skip):

**Lemma H.1.**
$$V_{h,\{s,u_t\}}^\star(s) = \sum_{t=h}^H u_t.$$

**Lemma H.2.** *Fix state $s$. For two different sequences $\{u_t\}_{t=1}^H$ and $\{u_t'\}_{t=1}^H$, we have*
$$\max_h \left\| Q_{h,\{s,u_t\}}^\star - Q_{h,\{s,u_t'\}}^\star \right\|_\infty \leq H \cdot \max_{t \in [H]} |u_t - u_t'|.$$

## H.3 Singleton-absorbing MDP

The well-definedness of singleton-absorbing MDP for linear MDP with anchor points depends on the following two lemmas whose proofs are still nearly identical to Lemma B.3 and Lemma B.5 which we skip.

**Lemma H.3.** $V_t^\star(s) - V_{t+1}^\star(s) \geq 0$, *for all state $s \in \mathcal{S}$ and all $t \in [H]$.*

**Lemma H.4.** *Fix a state $s$. If we choose $u_t^\star := V_t^\star(s) - V_{t+1}^\star(s) \ \forall t \in [H]$, then we have the following vector form equation*
$$V_{h,\{s,u_t^\star\}}^\star = V_{h,M}^\star \quad \forall h \in [H].$$

*Similarly, if we choose $\widehat{u}_t^\star := \widehat{V}_t^\star(s) - \widehat{V}_{t+1}^\star(s)$, then $\widehat{V}_{h,\{s,\widehat{u}_t^\star\}}^\star = \widehat{V}_{h,M}^\star$, $\forall h \in [H]$.*

The singleton MDP we used is exactly $M_{s,\{u_t^\star\}_{t=1}^H}$ (or $\widehat{M}_{s,\{u_t^\star\}_{t=1}^H}$).

## H.4 Proof for the optimal sample complexity

For $\widehat{\pi}^\star$, by (empirical) Bellman equation we have element-wisely:

$$\begin{aligned}
\widehat{Q}_h^{\widehat{\pi}^\star} - Q_h^{\widehat{\pi}^\star} &= r_h + \widehat{P}^{\widehat{\pi}_{h+1}^\star} \widehat{Q}_{h+1}^{\widehat{\pi}^\star} - r_h - P^{\widehat{\pi}_{h+1}^\star} Q_{h+1}^{\widehat{\pi}^\star} \\
&= \left( \widehat{P}^{\widehat{\pi}_{h+1}^\star} - P^{\widehat{\pi}_{h+1}^\star} \right) \widehat{Q}_{h+1}^{\widehat{\pi}^\star} + P^{\widehat{\pi}_{h+1}^\star} \left( \widehat{Q}_{h+1}^{\widehat{\pi}^\star} - Q_{h+1}^{\widehat{\pi}^\star} \right) \\
&= \left( \widehat{P} - P \right) \widehat{V}_{h+1}^{\widehat{\pi}^\star} + P^{\widehat{\pi}_{h+1}^\star} \left( \widehat{Q}_{h+1}^{\widehat{\pi}^\star} - Q_{h+1}^{\widehat{\pi}^\star} \right) \\
&= \ldots = \sum_{t=h}^H \Gamma_{h+1:t}^{\widehat{\pi}^\star} \left( \widehat{P} - P \right) \widehat{V}_{t+1}^{\widehat{\pi}^\star} \leq \underbrace{\sum_{t=h}^H \Gamma_{h+1:t}^{\widehat{\pi}^\star} \left| \left( \widehat{P} - P \right) \widehat{V}_{t+1}^{\widehat{\pi}^\star} \right|}_{(\star)}
\end{aligned}$$

where $\Gamma_{h+1:t}^{\pi^\star} = \prod_{i=h+1}^t P^{\pi_i^\star}$ is multi-step state-action transition and $\Gamma_{h+1:h} := I$.

---

[15] We use the integral only to denote $\mathcal{S}$ could be exponentially large.

## H.5 Analyzing (⋆)

**Concentration on $\left(\widehat{P} - P\right) \widehat{V}_h^\star$.** Since $\widehat{P}$ aggregates all data from different step so that $\widehat{P}$ and $\widehat{V}_h^\star$ are on longer independent. We use the singleton-absorbing MDP $M_{s, \{u_t^\star\}_{t=1}^H}$ to handle the case (recall $u_t^\star := V_t^\star(s) - V_{t+1}^\star(s) \,\forall t \in [H]$). **Here, we fix the state action $(s, a) \in \mathcal{K}$.** Then we have:

$$
\begin{aligned}
&\left(\widehat{P}_{s,a} - P_{s,a}\right) \widehat{V}_h^\star = \left(\widehat{P}_{s,a} - P_{s,a}\right) \left(\widehat{V}_h^\star - \widehat{V}_{h,\{s,u_t^\star\}}^\star + \widehat{V}_{h,\{s,u_t^\star\}}^\star\right) \\
&= \left(\widehat{P}_{s,a} - P_{s,a}\right) \left(\widehat{V}_h^\star - \widehat{V}_{h,\{s,u_t^\star\}}^\star\right) + \left(\widehat{P}_{s,a} - P_{s,a}\right) \widehat{V}_{h,\{s,u_t^\star\}}^\star \\
&\leq \left\|\widehat{P}_{s,a} - P_{s,a}\right\|_1 \left\|\widehat{V}_h^\star - \widehat{V}_{h,\{s,u_t^\star\}}^\star\right\|_\infty + \sqrt{\frac{2\log(4/\delta)}{N}} \sqrt{\mathrm{Var}_{s,a}(\widehat{V}_{h,\{s,u_t^\star\}}^\star)} + \frac{2H\log(1/\delta)}{3N} \\
&\leq \left\|\widehat{P}_{s,a} - P_{s,a}\right\|_1 \left\|\widehat{V}_h^\star - \widehat{V}_{h,\{s,u_t^\star\}}^\star\right\|_\infty + \sqrt{\frac{2\log(4/\delta)}{N}} \left(\sqrt{\mathrm{Var}_{s,a}(\widehat{V}_h^\star)} + \sqrt{\mathrm{Var}_{s,a}(\widehat{V}_{h,\{s,u_t^\star\}}^\star - \widehat{V}_h^\star)}\right) + \frac{2H\log(1/\delta)}{3N} \\
&\leq \left\|\widehat{P}_{s,a} - P_{s,a}\right\|_1 \left\|\widehat{V}_h^\star - \widehat{V}_{h,\{s,u_t^\star\}}^\star\right\|_\infty + \sqrt{\frac{2\log(4/\delta)}{N}} \left(\sqrt{\mathrm{Var}_{s,a}(\widehat{V}_h^\star)} + \sqrt{\left\|\widehat{V}_{h,\{s,u_t^\star\}}^\star - \widehat{V}_h^\star\right\|_\infty^2}\right) + \frac{2H\log(1/\delta)}{3N} \\
&= \left(\left\|\widehat{P}_{s,a} - P_{s,a}\right\|_1 + \sqrt{\frac{2\log(4/\delta)}{N}}\right) \left\|\widehat{V}_h^\star - \widehat{V}_{h,\{s,u_t^\star\}}^\star\right\|_\infty + \sqrt{\frac{2\log(4/\delta)}{N}} \sqrt{\mathrm{Var}_{s,a}(\widehat{V}_h^\star)} + \frac{2H\log(1/\delta)}{3N}
\end{aligned}
$$

$$(19)$$

where the first inequality uses Bernstein inequality (Lemma J.3) (**note here $P_{s,a}V = \int_{s'} V(s')dP(s'|s,a)$ since $\mathcal{S}$ could be continuous space, but this does not affect the availability of Bernstein inequality!**), the second inequality uses $\sqrt{\mathrm{Var}(\cdot)}$ is norm (norm triangle inequality). Now we treat $\left\|\widehat{P}_{s,a} - P_{s,a}\right\|_1$ and $\left\|\widehat{V}_h^\star - \widehat{V}_{h,\{s,u_t^\star\}}^\star\right\|_\infty$ separately.

**For $\left\|\widehat{P}_{s,a} - P_{s,a}\right\|_1$.** Recall here $(s,a) \in \mathcal{K}$. By Lemma J.9 we obtain w.p. $1 - \delta$

$$
\left\|\widehat{P}_{s,a} - P_{s,a}\right\|_1 \leq C\sqrt{\frac{|\mathcal{S}|\log(1/\delta)}{N}}. \tag{20}
$$

where $C$ absorbs the higher order term and constants.

**For $\left\|\widehat{V}_h^\star - \widehat{V}_{h,\{s,u_t^\star\}}^\star\right\|_\infty$.** Note if we set $\widehat{u}_t^\star = \widehat{V}_t^\star(s) - \widehat{V}_{t+1}^\star(s)$, then by Lemma H.4

$$
\widehat{V}_h^\star = \widehat{V}_{h,\{s,\hat{u}_t^\star\}}^\star
$$

Next since $\widehat{V}_{h,\{s,\hat{u}_t^\star\}}^\star(\tilde{s}) = \max_a \widehat{Q}_{h,\{s,\hat{u}_t^\star\}}^\star(\tilde{s}, a) \,\forall \tilde{s} \in \mathcal{S}$, by generic inequality $|\max f - \max g| \leq \max|f - g|$, we have $|\widehat{V}_{h,\{s,\hat{u}_t^\star\}}^\star(\tilde{s}) - \widehat{V}_{h,\{s,u_t^\star\}}^\star(\tilde{s})| \leq \max_a |\widehat{Q}_{h,\{s,\hat{u}_t^\star\}}^\star(\tilde{s}, a) - \widehat{Q}_{h,\{s,u_t^\star\}}^\star(\tilde{s}, a)|$, taking $\max_{\tilde{s}}$ on both sides, we obtain exactly

$$
\left\|\widehat{V}_{h,\{s,\hat{u}_t^\star\}}^\star - \widehat{V}_{h,\{s,u_t^\star\}}^\star\right\|_\infty \leq \left\|\widehat{Q}_{h,\{s,\hat{u}_t^\star\}}^\star - \widehat{Q}_{h,\{s,u_t^\star\}}^\star\right\|_\infty
$$

then by Lemma H.2,

$$
\left\|\widehat{V}_h^\star - \widehat{V}_{h,\{s,u_t^\star\}}^\star\right\|_\infty \leq \left\|\widehat{Q}_{h,\{s,\hat{u}_t^\star\}}^\star - \widehat{Q}_{h,\{s,u_t^\star\}}^\star\right\|_\infty \leq H \max_t |\hat{u}_t^\star - u_t^\star|, \tag{21}
$$

Recall

$$
\hat{u}_t^\star - u_t^\star = \widehat{V}_t^\star(s) - \widehat{V}_{t+1}^\star(s) - \left(V_t^\star(s) - V_{t+1}^\star(s)\right).
$$

Now we denote

$$
\Delta_s := \max_t |\hat{u}_t^\star - u_t^\star| = \max_t \left|\widehat{V}_t^\star(s) - \widehat{V}_{t+1}^\star(s) - \left(V_t^\star(s) - V_{t+1}^\star(s)\right)\right|,
$$

then $\Delta_s$ itself is a scalar and a random variable.

To sum up, by (19), (5) and (21) and a union bound over all $(s, a) \in \mathcal{K}$ we have

**Lemma H.5.** *Fix $N > 0$. With probability $1-\delta$, element-wisely, for all $h \in [H]$ and all $(s_k, a_k) \in \mathcal{K}$,*

$$\left| \left( \widehat{P}_{s_k, a_k} - P_{s_k, a_k} \right) \widehat{V}_h^\star \right| \leq C \sqrt{\frac{|\mathcal{S}| \log(HK/\delta)}{N}} \cdot H \max_{s_k} \Delta_{s_k}$$
$$+ \sqrt{\frac{2 \log(4HK/\delta)}{N}} \sqrt{\mathrm{Var}_{P_{s_k, a_k}}(\widehat{V}_h^\star)} + \frac{2H \log(HK/\delta)}{3N}$$

Now we extend Lemma H.5 to any arbitrary $(s, a)$ by proving the following lemma:

**Lemma H.6** (recover lemma). *For any function $V$ and any state action $(s, a)$, we have*

$$\sum_{k \in \mathcal{K}} \lambda_k^{s,a} \sqrt{\mathrm{Var}_{P_{s_k, a_k}}(V)} \leq \sqrt{\mathrm{Var}_{P_{s,a}}(V)}$$

*Proof of Lemma H.6.* Since $\lambda_k^{s,a}$ are probability distributions, by Jensen's inequality twice

$$\sum_{k \in \mathcal{K}} \lambda_k^{s,a} \sqrt{\mathrm{Var}_{P_{s_k, a_k}}(V)} \leq \sqrt{\sum_{k \in \mathcal{K}} \lambda_k^{s,a} \mathrm{Var}_{P_{s_k, a_k}}(V)}$$

$$= \sqrt{\sum_{k \in \mathcal{K}} \lambda_k^{s,a} \mathrm{Var}_{P_{s_k, a_k}}(V)} = \sqrt{\sum_{k \in \mathcal{K}} \lambda_k^{s,a} (P_{s_k, a_k} V^2 - (P_{s_k, a_k} V)^2)}$$

$$\leq \sqrt{\sum_{k \in \mathcal{K}} \lambda_k^{s,a} \cdot P_{s_k, a_k} V^2 - \left( \sum_{k \in \mathcal{K}} \lambda_k^{s,a} P_{s_k, a_k} V \right)^2}$$

$$= \sqrt{P_{s,a} V^2 - (P_{s,a} V)^2} = \sqrt{\mathrm{Var}_{P_{s,a}}(V)},$$

where we use $P_{s,a} = \sum_{k \in \mathcal{K}} \lambda_k^{s,a} P_{s_k, a_k}$. ∎

Therefore for all $(s, a)$, using Lemma H.5 and Lemma H.6 we obtain w.p. $1 - \delta$,

$$\left| \left( \widehat{P}_{s,a} - P_{s,a} \right) \widehat{V}_h^\star \right| \leq \sum_{k \in \mathcal{K}} \lambda_k^{s,a} \left| \left( \widehat{P}_{s_k, a_k} - P_{s_k, a_k} \right) \widehat{V}_h^\star \right|$$

$$\leq C \sum_{k \in \mathcal{K}} \lambda_k^{s,a} \sqrt{\frac{S \log(HK/\delta)}{N}} \cdot H \max_{s_k} \Delta_{s_k} + \sum_{k \in \mathcal{K}} \lambda_k^{s,a} \sqrt{\frac{2 \log(4HK/\delta)}{N}} \sqrt{\mathrm{Var}_{P_{s_k, a_k}}(\widehat{V}_h^\star)}$$

$$+ \sum_{k \in \mathcal{K}} \lambda_k^{s,a} \frac{2H \log(HK/\delta)}{3N}$$

$$= C \sqrt{\frac{S \log(HK/\delta)}{N}} \cdot H \max_{s_k} \Delta_{s_k} + \sum_{k \in \mathcal{K}} \lambda_k^{s,a} \sqrt{\frac{2 \log(4HK/\delta)}{N}} \sqrt{\mathrm{Var}_{P_{s_k, a_k}}(\widehat{V}_h^\star)}$$

$$+ \frac{2H \log(HK/\delta)}{3N}$$

$$\leq C \sqrt{\frac{S \log(HK/\delta)}{N}} \cdot H \max_{s_k} \Delta_{s_k} + \sqrt{\frac{2 \log(4HK/\delta)}{N}} \sqrt{\mathrm{Var}_{P_{s,a}}(\widehat{V}_h^\star)} + \frac{2H \log(HK/\delta)}{3N}$$

Now plug above back into $(\star)$, we receive:

$$\left| \widehat{Q}_h^{\widehat{\pi}^\star} - Q_h^{\widehat{\pi}^\star} \right|$$

$$\leq \sum_{t=h}^{H} \Gamma_{h+1:t}^{\widehat{\pi}^\star} \left( C \sqrt{\frac{S \log(HK/\delta)}{N}} \cdot H \max_{s_k} \Delta_{s_k} \cdot \mathbf{1} + \sqrt{\frac{2 \log(4HK/\delta)}{N}} \sqrt{\mathrm{Var}_P(\widehat{V}_{t+1}^\star)} + \frac{2H \log(HK/\delta)}{3N} \cdot \mathbf{1} \right)$$

$$\leq \sum_{t=h}^{H} \Gamma_{h+1:t}^{\widehat{\pi}} \sqrt{\frac{2 \log(4HK/\delta)}{N}} \sqrt{\mathrm{Var}_P(\widehat{V}_{t+1}^\star)} + C H^2 \sqrt{\frac{S \log(HK/\delta)}{N}} \cdot \max_s \Delta_s \cdot \mathbf{1} + \frac{2H^2 \log(HK/\delta)}{3N} \cdot \mathbf{1}$$

Similar to before, we get

$$\sqrt{\operatorname{Var}_P(\widehat{V}_h^\star)} := \sqrt{\operatorname{Var}_P\left(\widehat{V}_h^{\widehat{\pi}^\star}\right)} \le \sqrt{\operatorname{Var}_P\left(V_h^{\widehat{\pi}^\star}\right)} + \left\|\widehat{Q}_h^{\widehat{\pi}^\star} - Q_h^{\widehat{\pi}^\star}\right\|_\infty \tag{22}$$

Plug (22) back to above we obtain $\forall h \in [H]$,

$$
\begin{aligned}
\left|\widehat{Q}_h^{\widehat{\pi}^\star} - Q_h^{\widehat{\pi}^\star}\right| &\le \sqrt{\frac{2\log(4HK/\delta)}{N}} \sum_{t=h}^H \Gamma_{h+1:t}^{\widehat{\pi}^\star}\left(\sqrt{\operatorname{Var}_P\left(V_{t+1}^{\widehat{\pi}^\star}\right)} + \left\|\widehat{Q}_{t+1}^{\widehat{\pi}^\star} - Q_{t+1}^{\widehat{\pi}^\star}\right\|_\infty\right) \\
&\quad + CH^2\sqrt{\frac{S\log(HK/\delta)}{N}} \cdot \max_{s_k}\Delta_{s_k} \cdot \mathbf{1} + \frac{2H^2\log(HK/\delta)}{3N}\cdot\mathbf{1} \\
&\le \sqrt{\frac{2\log(4HK/\delta)}{N}} \sum_{t=h}^H \Gamma_{h+1:t}^{\widehat{\pi}^\star}\sqrt{\operatorname{Var}_P\left(V_{t+1}^{\widehat{\pi}^\star}\right)} + \sqrt{\frac{2\log(4HK/\delta)}{N}} \sum_{t=h}^H \left\|\widehat{Q}_{t+1}^{\widehat{\pi}^\star} - Q_{t+1}^{\widehat{\pi}^\star}\right\|_\infty \\
&\quad + CH^2\sqrt{\frac{S\log(HK/\delta)}{N}} \cdot \max_{s_k}\Delta_{s_k} \cdot \mathbf{1} + \frac{2H^2\log(HK/\delta)}{3N}\cdot\mathbf{1}
\end{aligned}
\tag{23}
$$

Apply Lemma J.5 and the (anchor version using recover lemma H.6) coarse uniform bound (Lemma J.10) we obtain the following lemma:

**Lemma H.7.** *With probability $1 - \delta$, for all $h \in [H]$,*

$$\left\|\widehat{Q}_h^{\widehat{\pi}^\star} - Q_h^{\widehat{\pi}^\star}\right\|_\infty \le \sqrt{\frac{C_0 H^3 \log(4HK/\delta)}{N}} + \sqrt{\frac{2\log(4HK/\delta)}{N}} \sum_{t=h}^H \left\|\widehat{Q}_{t+1}^{\widehat{\pi}^\star} - Q_{t+1}^{\widehat{\pi}^\star}\right\|_\infty + C'H^4\frac{S\log(HK/\delta)}{N}$$

*Proof.* Since

$$
\begin{aligned}
\Delta_{s_k} := \max_t |\hat{u}_t^\star - u_t^\star| &= \max_t \left|\widehat{V}_t^\star(s_k) - \widehat{V}_{t+1}^\star(s_k) - \left(V_t^\star(s_k) - V_{t+1}^\star(s_k)\right)\right| \\
&\le 2\cdot\max_t\left|\widehat{V}_t^\star(s_k) - V_t^\star(s_k)\right| \\
&= 2\cdot\max_t\left|\max_\pi\widehat{V}_t^\pi(s_k) - \max_\pi V_t^\pi(s_k)\right| \\
&\le 2\cdot\max_{\pi\in\Pi_g,t\in[H]}\left\|\widehat{V}_t^\pi - V_t^\pi\right\|_\infty \le C\cdot H^2\sqrt{\frac{|\mathcal{S}|\log(HK/\delta)}{N}}
\end{aligned}
\tag{24}
$$

where the last inequality uses (the anchor version) of Lemma J.10.[16] Then apply union bound w.p. $1 - \delta/2$, we obtain $\max_{s_k}\Delta_{s_k} \le C\cdot H^2\sqrt{\frac{|\mathcal{S}|\log(HK^2/\delta)}{N}}$. Note (23) holds with probability $1 - \delta/2$, therefore plug above into (23) and uses Lemma J.5 and take $\|\cdot\|_\infty$ we obtain w.p. $1 - \delta$, the result holds. ∎

**Lemma H.8.** *Given $N > 0$. Define $C'' := 2\cdot\max(\sqrt{C_0}, C')$ where $C'$ is the universal constant in Lemma H.7. When $N \ge 8H^2|\mathcal{S}|\log(4HK/\delta)$, then with probability $1 - \delta$, $\forall h \in [H]$,*

$$
\begin{aligned}
\left\|\widehat{Q}_h^{\widehat{\pi}^\star} - Q_h^{\widehat{\pi}^\star}\right\|_\infty &\le C''\sqrt{\frac{H^3\log(4HK/\delta)}{N}} + C''\frac{H^4 S\log(HK/\delta)}{N}. \\
\left\|\widehat{Q}_h^{\pi^\star} - Q_h^{\pi^\star}\right\|_\infty &\le C''\sqrt{\frac{H^3\log(4HK/\delta)}{N}} + C''\frac{H^4 S\log(HK/\delta)}{N}.
\end{aligned}
\tag{25}
$$

*Proof.* The proof is the same as Lemma B.9. ∎

**Remark H.9.** *Note the higher order term has dependence $H^4 S$. Use the same* self-bounding *trick, we can reduce it to $H^{3.5}S^{0.5}$.*

---

[16]Here the anchor version means for any $(s,a)$ we can apply $\|\widehat{P}_{s,a} - P_{s,a}\|_1 = \|\sum_k \lambda_k^{s,a}(\widehat{P}_{s,a} - P_{s,a})\|_1 \le \sum_k \lambda_k^{s,a}\|\widehat{P}_{s,a} - P_{s,a}\|_1$.

**Lemma H.10.** *Given $N > 0$. There exists universal constants $C_1, C_2$ such that when $N \geq C_1 H^2 |\mathcal{S}| \log(HK/\delta)$, then with probability $1 - \delta$, $\forall h \in [H]$,*

$$\left\| \widehat{Q}_h^{\widehat{\pi}^\star} - Q_h^{\widehat{\pi}^\star} \right\|_\infty \leq C_2 \sqrt{\frac{H^3 \log(HK/\delta)}{N}} + C_2 \frac{H^3 \sqrt{HS} \log(HK/\delta)}{N}. \tag{26}$$

*and*

$$\left\| \widehat{Q}_h^{\pi^\star} - Q_h^{\pi^\star} \right\|_\infty \leq C_2 \sqrt{\frac{H^3 \log(HK/\delta)}{N}} + C_2 \frac{H^3 \sqrt{HS} \log(HK/\delta)}{N}.$$

*Proof.* The proof is similar to Lemma B.11. ∎

### H.6 Proof of Theorem 6.2

*Proof.* By the direct computing of the suboptimality,

$$Q_1^\star - Q_1^{\widehat{\pi}^\star} = Q_1^\star - \widehat{Q}_1^{\pi^\star} + \widehat{Q}_1^{\pi^\star} - \widehat{Q}_1^{\widehat{\pi}^\star} + \widehat{Q}_1^{\widehat{\pi}^\star} - Q_1^{\widehat{\pi}^\star} \leq |Q_1^\star - \widehat{Q}_1^{\pi^\star}| + |\widehat{Q}_1^{\widehat{\pi}^\star} - Q_1^{\widehat{\pi}^\star}|,$$

then by Lemma H.10 we can finish the proof. ∎

### H.7 Take-away in the linear MDP with anchor setting.

Under the setting $S$ could be exponential large, $\mathcal{A}$ could be infinite (or even continuous space), with anchor representations ($K \ll |\mathcal{S}|$), our Theorem 6.2 has order $\widetilde{O}(\sqrt{H^3/N})$ when $N$ is sufficiently large. This translate to $N = \widetilde{O}(H^3/\epsilon^2)$ and the total sample used is $KN = \widetilde{O}(KH^3/\epsilon^2)$. This improves the total complexity $\widetilde{O}(KH^4/\epsilon^2)$ in Cui and Yang [2020] and is optimal.

## I The computational efficiency for the model-based offline plug-in estimators

For completeness, we discuss the computational and storage aspect of our model-based method. Its computational cost is $\widetilde{O}(H^4/d_m\epsilon^2)$ for computing $\widehat{P}$, the same as its sample complexity in steps ($H$ steps is an episode), and running value iteration causes $O(HS^2A)$ time (here we assume the bit complexity $L(P, r, H) = 1$, see Agarwal et al. [2019] Section 1.3). The total computational complexity is $\widetilde{O}(H^4/d_m\epsilon^2) + O(HS^2A)$. The memory cost is $O(HS^2A)$.

## J Assisting lemmas

**Lemma J.1** (Multiplicative Chernoff bound Chernoff et al. [1952]). *Let $X$ be a Binomial random variable with parameter $p, n$. For any $1 \geq \theta > 0$, we have that*

$$\mathbb{P}[X < (1 - \theta)pn] < e^{-\frac{\theta^2 pn}{2}}. \quad \text{and} \quad \mathbb{P}[X \geq (1 + \theta)pn] < e^{-\frac{\theta^2 pn}{3}}$$

**Lemma J.2** (Hoeffding's Inequality Sridharan [2002]). *Let $x_1, ..., x_n$ be independent bounded random variables such that $\mathbb{E}[x_i] = 0$ and $|x_i| \leq \xi_i$ with probability 1. Then for any $\epsilon > 0$ we have*

$$\mathbb{P}\left( \frac{1}{n} \sum_{i=1}^n x_i \geq \epsilon \right) \leq e^{-\frac{2n^2\epsilon^2}{\Sigma_{i=1}^n \xi_i^2}}.$$

**Lemma J.3** (Bernstein's Inequality). *Let $x_1, ..., x_n$ be independent bounded random variables such that $\mathbb{E}[x_i] = 0$ and $|x_i| \leq \xi$ with probability 1. Let $\sigma^2 = \frac{1}{n} \sum_{i=1}^n \mathrm{Var}[x_i]$, then with probability $1 - \delta$ we have*

$$\frac{1}{n} \sum_{i=1}^n x_i \leq \sqrt{\frac{2\sigma^2 \cdot \log(1/\delta)}{n}} + \frac{2\xi}{3n} \log(1/\delta)$$

**Lemma J.4** (Freedman's inequality Tropp et al. [2011]). *Let $X$ be the martingale associated with a filter $\mathcal{F}$ (i.e. $X_i = \mathbb{E}[X|\mathcal{F}_i]$) satisfying $|X_i - X_{i-1}| \leq M$ for $i = 1,...,n$. Denote $W := \sum_{i=1}^{n} \mathrm{Var}(X_i|\mathcal{F}_{i-1})$ then we have*

$$\mathbb{P}(|X - \mathbb{E}[X]| \geq \epsilon, W \leq \sigma^2) \leq 2e^{-\frac{\epsilon^2}{2(\sigma^2 + M\epsilon/3)}}.$$

*Or in other words, with probability $1 - \delta$,*

$$|X - \mathbb{E}[X]| \leq \sqrt{8\sigma^2 \cdot \log(1/\delta)} + \frac{2M}{3} \cdot \log(1/\delta), \quad Or \quad W \geq \sigma^2.$$

**Lemma J.5** (Sum of expectation of conditional variance of value; Lemma F.3 of Yin et al. [2021a]).

$$\mathrm{Var}_\pi \left[ \sum_{t=h}^{H} r_t^{(1)} \mid s_h^{(1)} = s_h, a_h^{(1)} = a_h \right]$$

$$= \sum_{t=h}^{H} \left( \mathbb{E}_\pi \left[ \mathrm{Var} \left[ r_t^{(1)} + V_{t+1}^\pi \left( s_{t+1}^{(1)} \right) \mid s_t^{(1)}, a_t^{(1)} \right] \mid s_h^{(1)} = s_h, a_h^{(1)} = a_h \right] \right.$$

$$\left. + \mathbb{E}_\pi \left[ \mathrm{Var} \left[ \mathbb{E} \left[ r_t^{(1)} + V_{t+1}^\pi \left( s_{t+1}^{(1)} \right) \mid s_t^{(1)}, a_t^{(1)} \right] \mid s_t^{(1)} \right] \mid s_h^{(1)} = s_h, a_h^{(1)} = a_h \right] \right)$$

*By apply above, one can show*

$$\sum_{t=h}^{H} \Gamma_{h+1:t}^\pi \sqrt{\mathrm{Var}_P \left( V_{t+1}^\pi \right)} \leq \sqrt{(H-h)^3} \cdot \mathbf{1}.$$

**Remark J.6.** *The infinite horizon discounted setting counterpart result is $(I - \gamma P^\pi)^{-1} \sigma_{V^\pi} \leq (1 - \gamma)^{-3/2}$.*

## J.1 Minimax rate of discrete distributions under $l_1$ loss.

This Section provides the minimax rate for $\left\| \widehat{P} - P \right\|_1$ for any model-based algorithms and is based on Han et al. [2015]. Let $P$ be $S$ dimensional distribution.

**Lemma J.7** (Minimax lower bound for $\left\| \widehat{P} - P \right\|_1$). *Let $n$ be the number of data-points sampled from $P$. If $n > \frac{e}{32} S$, then there exists a constant $p > 0$, such that*

$$\inf_{\widehat{P}} \sup_{P \in \mathcal{M}_S} \mathbb{P} \left[ \left\| \widehat{P} - P \right\|_1 \geq \frac{1}{8} \sqrt{\frac{eS}{2n}} - o(e^{-n}) - o(e^{-S}) \right] \geq p,$$

*where $\mathcal{M}_S$ denotes the set of distributions with support size $S$ and the infimum is taken over **ALL** estimators.*

**Remark J.8.** *Note the $\widehat{P}$ in above carries over all estimators but not just empirical estimator. This provides the minimax result.*

*Proof.* The proof comes from Theorem 2 of Han et al. [2015], where we pick $\zeta = 1$. Note they establish the minimax result for $\mathbb{E}_P \|\hat{P} - P\|_1$. However, by a simple contradiction we can get the above. Indeed, suppose

$$\inf_{\widehat{P}} \sup_{P \in \mathcal{M}_S} \mathbb{P} \left[ \left\| \widehat{P} - P \right\|_1 < \frac{1}{8} \sqrt{\frac{eS}{2n}} - o(e^{-n}) - o(e^{-S}) \right] = 1,$$

then this implies $\inf_{\widehat{P}} \sup_{P \in \mathcal{M}_S} \mathbb{E}_P \|\hat{P} - P\|_1 < \frac{1}{8} \sqrt{\frac{eS}{2n}} - o(e^{-n}) - o(e^{-S})$ which contradicts Theorem 2 of Han et al. [2015]. ∎

**Lemma J.9** (Upper bound for $\left\|\widehat{P} - P\right\|_1$). *Let $n$ be the number of data-points sampled from $P$. Then with probability $1 - \delta$*

$$\left\|\widehat{P} - P\right\|_1 \leq C\left(\sqrt{\frac{S\log(S/\delta)}{n}} + \frac{S\log(S/\delta)}{n}\right)$$

*for any $P \in \mathcal{M}_S$. Here $\widehat{P}$ is the empirical (MLE) estimator.*

*Proof.* First fix a state $s$. Let $X_i = \mathbf{1}[s_i = s]$, then $X_i \sim Bern(p_s(1 - p_s))$ and $X_s = \sum_{i=1}^{n} X_i \sim Binomial(n, p_i)$. By Bernstein inequality,

$$\left|\frac{X_s}{n} - P_s\right| \leq \sqrt{\frac{2p_s(1 - p_s)\log(1/\delta)}{n}} + \frac{3}{n}\log(1/\delta)$$

Apply a union bound we obtain w.p. $1 - \delta$

$$\left|\frac{X_s}{n} - P_s\right| \leq \sqrt{\frac{2p_s(1 - p_s)\log(S/\delta)}{n}} + \frac{3}{n}\log(S/\delta) \quad \forall s \in \mathcal{S}$$

which implies

$$
\begin{aligned}
\left\|\widehat{P} - P\right\|_1 &= \sum_{s \in \mathcal{S}} \left|\frac{X_s}{n} - P_s\right| \\
&\leq \sum_{s \in \mathcal{S}} \sqrt{\frac{2p_s(1 - p_s)\log(S/\delta)}{n}} + \frac{3S}{n}\log(S/\delta) \\
&= \sqrt{\frac{1}{n}} \sum_{s \in \mathcal{S}} \frac{1}{S} \cdot \sqrt{2S^2 p_s(1 - p_s)\log(S/\delta)} + \frac{3S}{n}\log(S/\delta) \\
&\leq \sqrt{\frac{1}{n}} \sqrt{2S^2 \cdot \frac{\sum_{s \in \mathcal{S}} p_s}{S}\left(1 - \frac{\sum_{s \in \mathcal{S}} p_s}{S}\right)\log(S/\delta)} + \frac{3S}{n}\log(S/\delta) \\
&= \sqrt{\frac{2(S - 1)\log(S/\delta)}{n}} + \frac{3S}{n}\log(S/\delta).
\end{aligned}
$$

where the last inequality uses the concavity of $\sqrt{x(1 - x)}$.

Finally, we can absorb the higher order term using the mild condition $n > c \cdot S\log(S/\delta)$. ∎

## J.2 A crude uniform convergence bound

Here we provide a crude bound for $\sup_{\pi \in \Pi_g} \left\|\widehat{V}_1^\pi - V_1^\pi\right\|_\infty$, which is the finite horizon counterpart of Section 2.2 of Jiang [2018] and is a form of simulation lemma.

**Lemma J.10** (Crude bound by Simulation Lemma). *Fix $N > 0$ to be number of samples for each coordinates. Recall $\Pi_g$ is the global policy class. Then w.p. $1 - \delta$,*

$$\sup_{\pi \in \Pi_g, h \in [H]} \left\|\widehat{Q}_h^\pi - Q_h^\pi\right\|_\infty \leq C \cdot H^2 \sqrt{\frac{S\log(SA/\delta)}{N}},$$

*which further implies*

$$\sup_{\pi \in \Pi_g, h \in [H]} \left\|\widehat{V}_h^\pi - V_h^\pi\right\|_\infty \leq C \cdot H^2 \sqrt{\frac{S\log(SA/\delta)}{N}},$$

*Proof.*

$$
\begin{aligned}
\widehat{Q}_h^\pi - Q_h^\pi &= r_h + \widehat{P}^{\pi_{h+1}}\widehat{Q}_{h+1}^\pi - r_h - P^{\pi_{h+1}}Q_{h+1}^\pi \\
&= \left(\widehat{P}^{\pi_{h+1}} - P^{\pi_{h+1}}\right)\widehat{Q}_{h+1}^\pi + P^{\pi_{h+1}}\left(Q_{h+1}^\pi - Q_{h+1}^\pi\right) \\
&= \left(\widehat{P} - P\right)\widehat{V}_{h+1}^\pi + P^{\pi_{h+1}}\left(\widehat{Q}_{h+1}^\pi - Q_{h+1}^\pi\right) \\
&= \ldots = \sum_{t=h}^{H}\Gamma_{h+1:t}^\pi\left(\widehat{P} - P\right)\widehat{V}_{t+1}^\pi \\
&\leq \sum_{t=h}^{H}\Gamma_{h+1:t}^\pi\left|\left(\widehat{P} - P\right)\widehat{V}_{t+1}^\pi\right| \\
&\leq \sum_{t=h}^{H} 1\cdot\max_{s,a}\left\|(\widehat{P} - P)(\cdot|s,a)\right\|_1\cdot\left\|\widehat{V}_{t+1}^\pi\right\|_\infty\cdot\mathbf{1} \\
&\leq H^2\cdot\max_{s,a}\left\|(\widehat{P} - P)(\cdot|s,a)\right\|_1\cdot\mathbf{1} \leq C\cdot H^2\sqrt{\frac{S\log(SA/\delta)}{N}}\mathbf{1}
\end{aligned}
$$

with probability $1 - \delta$, where the last inequality is by Lemma J.9. By symmetry and taking the $\|\cdot\|_\infty$, we obtain w.p. $1 - \delta$

$$
\sup_{\pi\in\Pi_g, h\in[H]}\left\|\widehat{Q}_h^\pi - Q_h^\pi\right\|_\infty \leq C\cdot H^2\sqrt{\frac{S\log(SA/\delta)}{N}}.
$$

The above holds for $\forall\pi\in\Pi_g$ since Lemma J.9 acts on $\left\|\widehat{P} - P\right\|_1$ and is irrelevant to $\pi$. ∎