# OpenReview forum: "Optimal Uniform OPE and Model-based Offline Reinforcement Learning in Time-Homogeneous, Reward-Free and Task-Agnostic Settings "
_NeurIPS.cc/2021/Conference — NeurIPS 2021 Poster_

### Official Review · Reviewer_rLZC · 2021-07-13

**Rating:** 4
**Confidence:** 5

**Summary:**

This work studies uniform convergence for offline policy evaluation (OPE) with model-based methods for episodic MDP. Particularly, uniform convergence for all near-empirically optimal policies is established.

**Limitations And Societal Impact:**

Yes

**Main Review:**

The motivation for considering uniform OPE is insufficient, since it is not suitable to studying offline learning. The bound established based on (local) uniform OPE is too loose compared with existing results for offline learning (both model based approach and model free approach, also known as asynchronous Q-learning). Hence, I think this work is of limited importance. In addition, the proof is based on Agarwal et al. 2020. It is not evident to me what new challenges are in the proof.

A. Agarwal, S. Kakade, and L. F. Yang. Model-based reinforcement learning with a generative model is minimax optimal. Conference on Learning Theory, 2020.

**Time Spent Reviewing:**

2

---

> ### Author Response · Authors · 2021-08-09
> **Response for the reviewer rLZC**
>
> We thank and respect the reviewer for providing the valuable feedbacks. We would appreciate it if the reviewer could check our following responses since we believe there are some misunderstandings in the current review.
>
> ---------------- "The motivation ... not suitable to studying offline learning." ----------------------------
>
> In the standard statistical learning theory, the learnability of supervised learning is proved via the framework of uniform convergence with empirical risk minimizer (ERM) ([Vapnik,2013],[Percy Liang,2016]). Therefore, empirical optimal policy suffices as a natural extension of ERM for offline/batch RL and uniform OPE will be the uniform convergence counterpart of supervised learning. We state this in the Introduction section and especially in Line32-51. In addition, this is not a new concept and has been well-motivated by [Yin et al. 2021b] and in https://simons.berkeley.edu/talks/tbd-243.
>
> Most importantly, the main reason for studying the uniform OPE is that it is actually suitable for offline learning (see Line 43 eqn (1)), since with $\epsilon$-optimal local / global uniform ope guarantee we have $\hat{\pi}^\star$ is a $2\epsilon$-optimal policy. Such a uniform convergence OPE idea has been leveraged concurrently for the linear function approximation task as well since it enables offline learning ([Yichun,Nathan,Masatoshi] page 9).
>
> [1] [Yichun,Nathan,Masatoshi] Fast Rates for the Regret of Offline Reinforcement Learning, COLT 2021.
>
> [2] [Percy Liang] Statistical Learning Theory Stanford231 (Winter2016)
>
> We are not sure why the reviewer state it is not suitable for offline learning and there is no explanation from the reviewer. We are happy to receive further feedback and would love to answer in the later rolling discussions.
>
>
> ---- "The bound ... (local) uniform OPE is too loose compared with existing offline learning results" -----
>
> We study the sample complexity from the standard statistical perspective, i.e. in the minimax sense. For the global uniform OPE, our lower bound (Theorem 3.1) already shows any (model-based) algorithm needs $\frac{H^2S}{d_m\epsilon^2}$ complexity. This required extra $S$ factor further motivates us to consider the local uniform OPE (for more motivation please refer to Line216-226). For the local uniform OPE, our Theorem 4.1 achieves the sample complexity with rate $\widetilde{O}(H^2/d_m\epsilon^2)$, which implies sample-optimal offline learning with the same rate (Corollary 4.2). It is indeed rate-optimal since it matches the previous minimax lower bound (Theorem 4.2 of [Yin et al. 2021a]) up to the logarithmic factor.
>
> To sum up, our local uniform OPE upper bound is indeed tight and the global uniform OPE is an intrinsic harder problem by our lower bound result. Yet again, we have no idea why the reviewer commented our bound is too loose in the existing offline learning (like in what sense?) and there is no explanation from the reviewer.
>
> ----- "In addition, the proof ... Agarwal et al. 2020 ... what new challenges are in the proof." -------
>
> Thank you for the question. [Agarwal et al. 2020] propose the covering-based absorbing MDP technique and we design the new method, named singleton-absorbing MDP. The challenge is that the old technique is unable to answer whether the plug-in method is optimal for the finite horizon MDPs and our new technique provides a positive answer for this open question.
>
> The covering-based absorbing MDP in [Agarwal et al. 2020] is specifically designed for the infinite-horizon discounted setting. The covering argument works in finite horizon discounted setting because $V^\star(s)$ (or $\widehat{V}(s)$) is an one-dimensional quantity varies within the interval $[0,(1-\gamma)^{-1}]$. When borrowing such a technique to the finite horizon setting, the covering argument needs to cover the $H$-dimensional hypercube $[0,H]^H$ and hence the covering number is of the order $\exp(H)$ (which results in the metric entropy to have the form $H\cdot\log(\ldots)$) and causes the suboptimal analysis due to the extra horizon dependence. This phenomenon is well-studied by [Qiwen Cui, Lin Yang, 2020] and they made a pessimistic conjecture that absorbing MDP technique is not well-suited for finite horizon MDP (please check their Section 7, first bullet point for the detailed discussion).
>
> [3] [Qiwen C., Lin Y.] Is Plug-in Solver Sample-Efficient for Feature-based Reinforcement Learning? NeurIPS, 2020.
>
> Our singleton absorbing MDP technique answers this open question affirmatively. Indeed, due to the horizon curse by the covering argument, we abandon the evenly fine-grid designs and use only one single (random) absorbing MDP $\widehat{V}^\star_{s,u^\star}$ for each state. Such a design resolves the exponential covering issue and also approximates the true $\widehat{V}^\star(s)$ accurately at the same time (with our careful choice of $u^\star$ Lemma 4.3, Definition 4.4). Figure 2 visualizes the differences in the 2-D case. Our technique is a general sharp tool as it adapts to other offline settings as discussed in the paper. All the discussions can be found in Line99-110 and Line264-310 in the main text (Also see the Question&Clarity comments from the reviewer LF7M).
>
> We kindly request the reviewer to check our paper in a more careful manner and are happy to answer any essential questions in the future discussions :)

---

> > ### Comment · Reviewer_rLZC · 2021-08-18
> > **The authors misunderstand all the comments**
> >
> > The authors misunderstand all the comments.
> >
> > I said "uniform OPE is not suitable for offline learning" since the bound established by uniform OPE is much worse than the common results in offline learning. Specifically, the minimax optimal offline learning has been shown in a broader epsilon range in [2]. More importantly, offline learning often has access to the expert data, which can imitate the behavior of optimal policy, i.e., d^{\pi^{\star}}(s, a)/d^{\mu}(s, a) is small. However, uniform OPE requires that d^{\pi}(s, a)/d^{\mu}(s, a) is small for all policies including even the worst policy, which is far away from the expert behavior. In addition, the same arguments apply for offline learning with function approximation.
> >
> > The above two points are also the reason why I said "The bound established based on (local) uniform OPE is too loose compared with existing results for offline learning".
> >
> > [1] Paria Rashidinejad, Banghua Zhu, Cong Ma, Jiantao Jiao, Stuart Russell, Bridging offline reinforcement learning and imitation learning: A tale of pessimism
> >
> > [2] Tengyang Xie, Nan Jiang, Huan Wang, Caiming Xiong, Yu Bai, Policy Finetuning: Bridging Sample-Efficient Offline and Online Reinforcement Learning
> >
> > [3] Masatoshi Uehara, Wen Sun, Pessimistic Model-based Offline RL: PAC Bounds and Posterior Sampling under Partial Coverage
> >
> > As for "Improve the dependence on H for finite horizon MDP", the authors mislead the information given in [4]. They consider the RL with function approximation but not the tabular case. Actually, the (local) minimax bound for the tabular case has already been established in [5] in a more complicated setting. The decoupling method is based on the simple decomposition |(\hat{P} - P)\hat{V}| \le |(\hat{P} - P)V^{\star}| + |(\hat{P} - P)(\hat{V}-V^{\star})| given the small epsilon range. I think this basic technique can also be used for offline learning.
> >
> > [4] Qiwen Cui, Lin F. Yang, Is Plug-in Solver Sample-Efficient for Feature-based Reinforcement Learning?
> >
> > [5] Mohammad Gheshlaghi Azar, Ian Osband, and Remi Munos, Minimax Regret Bounds for Reinforcement Learning
> >
> > I'm quite familiar with this topic and the related techniques. I kindly request the authors to answer the comments in a more positive view to avoid misleading.

---

> > > ### Author Response · Authors · 2021-08-20
> > > **Further response to reviewer rLZC**
> > >
> > > We thank the reviewer for providing detailed explanations.
> > >
> > > First of all, we would like to reiterate the uniform OPE setting brings the important uniform convergence notion from statistical supervised learning theory into offline RL and this is a meaningful setting that has its own scientific merit. The following are our detailed responses.
> > >
> > > ---- "Specifically, the minimax optimal offline learning has been shown in a broader epsilon range in [2]" ------
> > >
> > > Thank you, but in general this seems not true. In [2], it states (in their Theorem 2, page 7 last two lines) their minimax optimal offline learning result requires $\epsilon\leq H^{-2.5}$ (we do point out they also have a Theorem 4, but that algorithm includes an online procedure which can obtain more information than the fixed offline data by online exploration and is beyond the scope of offline learning).
> > >
> > > In our Corollary 4.2, if we choose $\widehat{\pi}=\widehat{\pi}^\star$, then the suboptimality gap becomes (of order) $\sqrt{\frac{H^2}{nd_m}}+\frac{H^{2.5}S^{0.5}}{nd_m}$ and this can be translated into sample complexity (of order)
> > > $
> > > n=\frac{H^2}{d_m\epsilon^2}+ \frac{H^{2.5}S^{0.5}}{d_m\epsilon}
> > > $
> > > and it is minimax optimal if $\epsilon<1/\sqrt{HS}$.
> > >
> > > In general, $H^{-2.5}$ and $1/\sqrt{HS}$ are not well comparable since when $H$ is relatively large to $S$ then $H^{-2.5}$ can by much smaller than $1/\sqrt{HS}$, therefore we are not sure why [2] provides a broader epsilon range. Such a $\epsilon$-range dependence is due to the looseness in the higher order term and we are unable to discuss more due the space limit (see our Line 376-377).
> > >
> > >
> > >  Meanwhile, we thank the reviewer for pointing this sample barrier behavior in [6]. We are happy to include the above discussion on [2] in the paper too, but we would like to point out that [2] is released on arxiv after the NeurIPS deadline and it seems a little unfair to expect us to discuss about it in the initial submission.
> > >
> > >
> > > [2] Tengyang Xie, Nan Jiang, Huan Wang, Caiming Xiong, Yu Bai, Policy Finetuning: Bridging Sample-Efficient Offline and Online Reinforcement Learning
> > >
> > > [6] [Gen Li et al.] Breaking the sample size barrier in model-based reinforcement learning with a generative model
> > >
> > >
> > > ----- "More importantly, offline learning often has access to the expert data, which can imitate the behavior of optimal policy, i.e., $d^{\pi^{\star}}(s, a)/d^{\mu}(s, a)$ is small. However, uniform OPE requires that $d^{\pi}(s, a)/d^{\mu}(s, a)$ is small for all policies including even the worst policy" ----
> > >
> > > Thank you! Uniform coverage assumption is indeed stronger than partial coverage (e.g. [2]) but it is required since we are aiming for a stronger guarantee: uniform OPE. We have already mentioned in the main text the uniform coverage can be relaxed to partial coverage for pure learning task (see Line 179-181 and also Line 583-585).
> > >
> > > In addition, we agree with the reviewer that the partial coverage can provide better result in the imitation learning regime. However, there are also cases where $d_m$-based bound can provide better sample complexity result than the $C^\star$ based one. For example: consider a near-uniform behavior policy $\mu$, then there exists an MDP model s.t. $d_m\approx c\cdot 1/SA$ (i.e. reduces to the generative model setting). On the other hand, for some state-action $(s_0,a_0)$ with constant probability $c'$ under $\pi^\star$, then $SC^\star\geq S\cdot \frac{d^\star(s_0,a_0)}{d^\mu(s_0,a_0)}\geq S\frac{c'}{1/SA}=c' S^2A$ which is higher than $1/d_m \approx c SA$ complexity (For the concrete example of such an MDP, please see the hard instance construction in Section G of [Yin et al. 2021a], in particular state $s_{h+1,i}$). Intuitively, this phenomenon
> > > can be explained from the technical perspective: when plug out the ratio $d^\star/d^\mu$, it will only be helpful if it is a constant (i.e. the imitation learning regime), and it could be lossy if the maximum ratio is large (e.g. $SA$) since when further apply the Cauchy inequality, the bound has to suffer the additional $\sqrt{S}$ based on the $C^\star$ assumption; on the other hand, plug out $1/d_m$ can be bad if $\mu$ imitates some deterministic optimal policy but it is near-optimal if $\mu$ is near-uniform.
> > >
> > >
> > >
> > > In this sense, the partial coverage does not contradict with our $d_m$ assumption for uniform OPE problem nor the sample complexity bound in [2] can imply ours.
> > >
> > > ---- "As for "Improve the dependence on $H$ for finite horizon MDP", the authors mislead the information given in [4]. They consider the RL with function approximation but not the tabular case." -------
> > >
> > > This is a misunderstanding. Indeed, in [Cui and Yang] [4] Assumption 1, they assume the anchor assumption for the linear MDP structure. Our Definition 5.5 is the finite horizon counterpart of their setting. Our theorem 5.6 improves over their Theorem 4 and the total sample complexity saving by a factor of $H$. This is due to our new singleton absorbing MDP technique (see our discussion Line 366-369).
> > >
> > > ----- "Actually, the (local) minimax bound for the tabular case has already been established in [5] in a more complicated setting. The decoupling method is based on ... the small epsilon range. " -------
> > >
> > > First, the online algorithm in [5] does not imply our optimal local uniform OPE results. Second, technically, we never claim our new analysis is the only technique that can achieve minimax optimality in our paper. What we stated in the paper are the two aspects (Line99-110): 1. the singleton-absorbing is a new sharp analysis tool that improves over the covering-absorbing [Agarwal et al. (2020)] technique; 2. we provide a positive answer for the seemingly negative conjecture in [Cui and Yang] [4] that absorbing MDP technique can work in the finite horizon case. We hope this answers your concern.
> > >
> > > We thank the reviewer again and are looking forward to any further discussions from you.

---

> > > > ### Author Response · Authors · 2021-09-02
> > > > **Re: to reviewer rLZC**
> > > >
> > > > Hello reviewer rLZC, we are certain our detailed responses have addressed your questions from the scientific perspective. Could you please at least let us know if your concerns are resolved? We appreciate your time.

---

> > > > > ### Author Response · Authors · 2021-09-05
> > > > > **Re: Re: to reviewer rLZC**
> > > > >
> > > > > Hello reviewer rLZC, since we didn’t hear any response from you, as the last message we want to mention our results are tight in the minimax sense. Our Thm 3.1 is tight since it has quadratic dependence on $S$ (Note for Lower bound, the **larger** lower bound means **tighter** result). Our local result 4.1/4.2 is also tight as it matches the rate-optimal offline learning result under $d_m$ (Figure 1). Thm 3.1 and 4.1 characterize the statistical gap between the local and the global case for uniform OPE problems.

---

### Official Review · Reviewer_smXo · 2021-07-16

**Rating:** 7
**Confidence:** 5

**Summary:**

This paper considers uniform convergence in model-based off-policy evaluation and learning in time-homogeneous setting, and provide sharp error bound in different scenarios.

**Limitations And Societal Impact:**

The authors have discussed them properly.

**Main Review:**

The authors considers uniform convergence in model-based offline reinforcement learning, and generalize the results to two slightly different setting, termed as task-agnostic and reward-free reinforcement learning. Central to the analysis is an absorbing MDP technique to decouple the statistical dependency on the empirical and the terms obtained from the empirical model, which is a generalization from the infinite horizon case and eventually lead to a sharp result on the leading term. The results are clearly presented. One minor issue is that, the higher order term are not very satisfactory, especially the results in Theorem 5.6, that need samples proportion to the number of states in linear MDP with anchor points, which should be avoided in the setting with function approximation (as we want to deal with the large state space with function approximation).

**Time Spent Reviewing:**

2

---

> ### Author Response · Authors · 2021-08-09
> **Response for the reviewer smXo**
>
> We appreciate the reviewer for the valuable comments and the precise understanding of our paper.
>
> ----- "the higher order term ... especially the results in Theorem 5.6, that need samples proportion to the number of states in linear MDP with anchor points" -----
>
> Thank you for the insightful question! Yes, as we mentioned in Section 6 (Line376-377), our higher order term is indeed suboptimal by a factor of $\sqrt{HS}$ (in terms of the sample size $n$) and we believe tightening both the dominate term and the higher order term is a long lasting question for the RL theoretical community. To the best our knowledge, the work with best higher order dependence in online RL is [Zihan Z., Xiangyang J., Simon S. D., 2021] where they have the ``translated'' suboptimality gap (regret divided by $n$) to be
> $\sqrt{\frac{SA}{n}}+\frac{S^2A}{n}$ and with best higher order dependence in offline RL is [Ren et al. 2021] with
> $\sqrt{\frac{1}{nd_m}}+\frac{S}{nd_m}$. In this sense, tightening the higher order term remains an open question.
>
>
> [1] [Zihan Z., Xiangyang J., Simon S. D.] Is reinforcement learning more difficult than bandits? a near-optimal algorithm escaping the curse of horizon. COLT, 2021.
>
>
> Regrading the $S$ in the linear MDPs (with anchor points) example, it happens since the current singleton absorbing technique in this paper is designed for discrete MDPs.
> For the general linear MDP of the form
> $$
> r_{h}(s, a)=\theta_{h} \cdot \phi(s, a), \quad P(\cdot \mid s, a)=\mu\cdot \phi(s, a), \forall h\in[H]
> $$
> the singleton absorbing MDP idea might still work since we can mirror model-based estimator via the least square value iteration:
> $$
> \widehat{P}^{n}(\cdot \mid s, a) \cdot V:=\left(\widehat{\mu}^{n} \phi(s, a)\right) \cdot V=\phi(s, a)^{\top} \sum_{i=0}^{n-1}\left(\Lambda^{n}\right)^{-1} \phi\left(s^{i}, a^{i}\right) V\left(s^{\prime i}\right)
> $$
> where the above is the time-homogenuous version of the last equation of page 63 in the [AJKS] book (Note in the above expression it is valid for $\mu$ to be an infinite-dimensional vector, unlike (0.3) of page 63 in the book). Since the above holds for all $V$, for the quantity of interest $\widehat{P}^{n}(\cdot \mid s,a)\cdot \widehat{V}$, we may design a similar singleton absorbing MDP $\widehat{V}_{s,h,u}$ for decoupling the statistical dependence. Despite all that, to be mathematical rigorous we point out that such an idea needs careful treatments (e.g. one needs to make sure $\widehat{P}^n$ is a valid probability distribution) and we leave this as the future works.
>
> [2] [Agarwal, Jiang, Kakade, Sun] Reinforcement Learning: Theory and Algorithms, 2021.
>
> Lastly, going back to the singleton MDP technique, if we ignore the higher order dependence, then even in infinite-horizon discounted setting our technique remains optimal (this should be clear by comparing Figure 2 (a) and (c)). In this sense, single-absorbing is a more general technique as it applies to a wider range of problems.
>
> We will add the discussions for all of those in our revision.

---

### Official Review · Reviewer_Rbai · 2021-07-20

**Rating:** 6
**Confidence:** 4

**Summary:**

This paper studies uniform off-policy evaluation in the episodic setting. In particular, the authors consider a stationary episodic setting. The goal is to evaluate the value function corresponding to a given policy, up to some given accuracy, uniformly among all the policies. The idea is that a uniform off-policy evaluation can be used for off-policy learning directly, due to the equation (1).
The authors first characterize a minimax lower bound for global uniform OPE, and they show that this lower bound is proportional to the cardinality of the state space. Next they characterize a local uniform upper bound for plug-in estimator for OPE.
Next, the authors consider offline task-agnostic and offline reward-free learning. They provide the sample complexity of model based methods for these tasks, and they show that these bounds are optimal.

**Limitations And Societal Impact:**

Lack of experimental results.

**Main Review:**

- My main concern about this paper is on the relation between OPE and off-policy learning. The authors claim that global OPE can be directly translated to off-policy learning by an exhaustive search over space of all the policies. However, Search over space of all the policies is exponentially expensive in the cardinality of state and action space. Am I missing something here?
- Typo: Line 284: supports our design

**Time Spent Reviewing:**

10

---

> ### Author Response · Authors · 2021-08-09
> **Response for the reviewer Rbai**
>
> We appreciate the reviewer for the valuable comments and the precise understanding of our paper.
>
> ------ "the relation between OPE and off-policy learning ... However, Search over space ... exponentially expensive in the cardinality of state and action space." ------
>
> Thanks! What we mean is simply that offline learning can be achieved by finding the empirical optimal policy $\widehat{\pi}^\star$ when the uniform OPE is guaranteed. Here "offline learning can be achieved" means one can learn a policy that is near optimal to best policy within the selected policy class. The followings are the detailed explanations.
>
> Even though we analyze the two representative policy classes (the global class $\Pi_g$ and the local class $\Pi_l$), the concept of uniform OPE $\sup_{\pi\in\Pi}|\widehat{v}^\pi-v^\pi|$ can be readily applied to any policy class $\Pi$ of interests.
>
> 1. If the cardinality of $|\Pi|=k$ is small, then given the uniform OPE guarantee $\sup_{\pi\in\Pi}|\widehat{v}^\pi-v^\pi|<\epsilon$ we can select the best empirical optimal policy  $\widehat{\pi}^\star:=\mathrm{argmax}_{\pi\in\{\pi_1,\ldots,\pi_k\}}\widehat{v}^\pi$ and it is a $2\epsilon$-optimal policy (comparing to the best policy in $\Pi$). Such a procedure can be done by enumerating (since $k$ is small) and when the set of policies are small this task is usually referred as policy selection [Shayan D., Philip T., Emma B., 2017];
>
> [1] [Shayan D., Philip T., Emma B.] Importance Sampling for Fair Policy Selection, UAI best paper, 2017.
>
> 2. If the policy class $|\Pi|$ is exponentially large (say the global policy class $\Pi_g$), then we could run value iteration (VI) / policy iteration (PI) to efficiently obtain the empirical optimal policy $\widehat{\pi}^\star=\mathrm{argmax}_{\pi\in\Pi}\widehat{v}^\pi$. Even though the policy class is exponentially large or even infinite (if we include all the stochastic policies), for finite horizon discrete MDPs it is known that VI is computationally efficient, since it only requires the Bellman update and the max procedure over $A$ actions for all $S$ states at each time $h$ and the total complexity is polynomial in $H,S,A$ (instead of exponential computations), which exactly reflects the dynamic programming principle [Puterman, 1994]. The related discussion can be found in Appendix I.
>
> [2] [Puterman] Markov Decision Processes: Discrete Stochastic Dynamic Programming, 1994.
>
> Nevertheless, in practice it suffices to first build the model $\widehat{P}$ and plan over $\widehat{P}$ to find $\widehat{\pi}^\star$ for the purpose of offline learning (unless you have safety concern, e.g. for constraint MDP, but that is beyond the scope of this paper [also see our footnote 5]). We will discuss this in a clear way in our revision and are happy to take further questions.

---

> > ### Author Response · Authors · 2021-08-11
> > **Response to the Further question of Reviewer Rbai**
> >
> > The VI / PI we mentioned in the rebuttal refers to the standard value iteration or policy iteration algorithm which only requires an known transition $P$ [Sutton and Barto, 2018] Chapter 4.3, 4.4. Therefore, no additional samples are needed on top of our local uniform OPE result since samples are used only for constructing $\widehat{P}$. The VI/PI are running over the transition $\widehat{P}$ (Corollary 4.2) hence will not have any additional sample cost but do have computational cost. For the sample complexity comparisons, it is already listed in Figure 1 (model-based plug-in is ours). For the computational complexity, please check the appendix I and the computational complexity comparison between different methods can be vague/ill-defined since it depends on the definition of the specific machine precision/bit complexity (see [AJKS (2019)] Section 1.3 for a discussion in the infinite horizon discounted setting). Lastly, to help the reviewer distinguish the differences, we mention value iteration is a standard procedure for the purely planning purpose and this is different from VI-based algorithms (e.g. UCB-VI for online RL [Azar et al. 2017] and the LCB-VI for offline RL [Rashidinejad et al.]) that aims for both learning and planning (where the latter needs to use the samples).
> >
> > [1] [Sutton and Barto] Reinforcement learning: An Introduction, 2018.
> >
> > [2] [AJKS] Reinforcement learning: Theory and algorithms, 2019.
> >
> > [3] [Azar et al.] Minimax regret bounds for reinforcement learning, 2017.
> >
> > [4] [Rashidinejad et al.] Bridging offline reinforcement learning and imitation learning: A tale of pessimism, 2021.
> >
> > Thank you for asking and we are happy to answer any further questions.

---

### Official Review · Reviewer_LF7M · 2021-07-20

**Rating:** 7
**Confidence:** 2

**Summary:**

In this paper the authors provide uniform convergence results for offline policy evaluation, a minimax lower bound for global uniform OPE and an upper bound for local uniform OPE, with a new design analysis tool called singleton absorbing MDP. Besides, the authors generalize the results to new settings, including offline task-agnostic and offline reward-free rl.

**Ethics Review Area:**

["I don’t know"]

**Limitations And Societal Impact:**

I have two questions regarding the offline new settings presented in this paper:

1. For the offline task-agnostic learning settings, if we using a simple union bound over the K tasks, can we get similar results?

2. For the Offline Reward-free Learning, why this is an interesting settings and what we can benefit if we consider this setting?

**Main Review:**

Originality: The results presented in the paper is new and tighter than the prior works (Yin et.al 2021), and the proving idea of using singleton-absorbing MDP technique is novel.  Also, the new settings proposed in this paper (offline task-agnostic and reward-free rl ) are interesting.

Quality & Clarity: Overall the paper is well-written and results are well-presented. The visualizations of singleton absorbing mdp in the appendix are really nice and provide intuitions for understanding.

Significance: To me this is a nice theoretical paper and the paper makes contributions to the theoretical offline rl community.

**Time Spent Reviewing:**

5

---

> ### Author Response · Authors · 2021-08-09
> **Response for the reviewer LF7M**
>
> We appreciate the reviewer for the valuable comments and the precise understanding of our paper.
>
> ------- "For the offline task-agnostic learning settings, ... union bound over the $K$ tasks ... similar results?" -------
>
> Good observation and yes! Actually, this is exactly the message we want to convey (Line832-838), which is the union bound. This reveals two things: 1. from the statistical perspective, as long as we can learn $P$ via the model-based $\widehat{P}$, then doing the offline task-agnostic setting will only sacrifice the additional dependence of $\log(K)$; 2. in terms of the proving technique, as long as we can overcome the technical hurdle for the optimality of the specific reward (via singleton absorbing MDP), the offline task-agnostic problem can be proved only via an extra union bound (since $\log(K)$ is the essential quantity and can be achieved by union bound. More detailed discussion can be found in Appendix G and the original paper [Zhang et al. 2020b] that addresses the *online version* of the problem).
>
> ------ "For the Offline Reward-free Learning, why this is an interesting settings and what we can benefit if we consider this setting?" ------
>
> Thanks for the good question! We provide our explanations from two aspects.
>
> First of all, Reward-free learning is naturally suitable for batch RL since it contains the exploration phase (where the offline data are collected / provided) and the planning phase with any arbitrary (even data-dependent) reward. Frequently, the historical offline data could have missing reward information (e.g. due to the artificial negligence) and designers could look at the offline data to decide on an appropriate reward to use, e.g., for a new purpose. Those scenarios match the context of offline reward-free learning. Besides, the statement that Reward-free learning is suitable for the offline RL framework is well-demonstrated by previous literatures (e.g. [Ruosong W., Simon Du., Lin F., Ruslan S. 2020], [Zihan Z., Simon Du., Xiangyang J. 2021] check their first two sentences of the abstracts).
>
> [1] [Ruosong W., Simon Du., Lin F., Ruslan S.] On Reward-Free Reinforcement Learning with Linear Function Approximation, NeurIPS 2020
>
> [2] [Zihan Z., Simon Du., Xiangyang J.] Nearly Optimal Reward-Free Reinforcement Learning, ICML 2021
>
> Second, the practical reason for the emergence of reward-free RL is: in various applications, it is necessary to re-design the reward function to incentivize the agent to learn new desired behavior. A concrete example is: in the logistics system, the transportation of the goods are optimized according to the rewards (e.g. the value of destination of the goods). During the epidemic, the supply-and-demand relationship drastically changes and the rewards are re-designed (the destination with higher value before the epidemic may have lower value nowadays). In this scenario, the optimization of the transportation of the goods can be abstracted as an offline reward-free learning problem (since only historical data are available with no exploration information for this newly designed reward).
>
> Since the offline reward-free learning is not our main focus, we didn't include above discussions in the main paper due to the limited space constraint. We will add an discussion in the appendix.

---

### Decision · Program_Chairs · 2021-09-27

**Decision:**

Accept (Poster)

**Comment:**

The paper studies the uniform UPE problem in the offline RL setting with both upper and lower bound. Most of the reviewers believe the paper is well-written. There is one major concern about the epsilon range. The current results give the bound for eps<1/sqrt{S}. The paper would be much stronger if results about eps>1/sqrt{S} is also provided.